



# Radiative closure and cloud effects on the radiation budget based on satellite and ship-borne observations during the Arctic summer research cruise PS106

Carola Barrientos-Velasco, Hartwig Deneke, Anja Hünerbein, Hannes J. Griesche, Patric Seifert, and Andreas Macke

Leibniz Institute for Tropospheric Research, Leipzig, Germany

**Correspondence:** Carola Barrientos-Velasco (barrientos@tropos.de)

**Abstract.** For understanding Arctic climate change, it is critical to quantify and address uncertainties in climate data records on clouds and radiative fluxes derived from long-term passive satellite observations. A unique set of observations collected during the research vessel *Polarstern* PS106 expedition (28 May to 16 July 2017) by the OCEANET facility is exploited here for this purpose and compared with the CERES SYN1deg Ed. 4.1 satellite remote sensing products. Mean cloud fraction

(CF) of 86.7 % for CERES and 76.1 % for OCEANET were found for the entire cruise. The difference of CF between both data sets is due to different spatial resolution and momentary data gaps due to technical limitations of the set of ship-borne instruments. A comparison of radiative fluxes during clear-sky conditions enables radiative closure for CERES products by means of independent radiative transfer simulations. Several challenges were encountered to accurately represent clouds in radiative transfer under cloudy conditions, especially for ice-containing clouds and low-level stratus (LLS) clouds. During

LLS conditions, the OCEANET retrievals were in particular compromised by the altitude detection limit of 155 m of the cloud radar. Radiative fluxes from CERES show a good agreement with ship observations, having a bias (standard deviation) of -6.0 (14.6) $\mathrm{W\,m^{-2}}$ and 23.1 (59.3) $\mathrm{W\,m^{-2}}$ for the downward longwave (LW) and shortwave (SW) fluxes, respectively. Based on CERES products, mean values of the radiation budget and the cloud radiative effect (CRE) were determined for the PS106 cruise track and the central Arctic region (70°-90°N). For the period of study, the results indicate a strong influence of the SW

flux in the radiation budget, which is reduced by clouds leading to a net surface CRE of -8.8 $\mathrm{W\,m^{-2}}$ and -9.3 $\mathrm{W\,m^{-2}}$ along the PS106 cruise and for the entire Arctic, respectively. The similarity of local and regional CRE supports that the PS106 cloud observations can be considered to be representative of Arctic cloudiness during early summer.

## 1 Introduction

Arctic warming is a robust feature of climate change (Meredith et al., 2019). The Arctic air temperature increases at more than twice the rate of the global mean air temperature (Ballinger et al., 2020). This phenomenon, named Arctic amplification, has



been predicted by models and confirmed by measurements (Serreze and Barry, 2011; Winton, 2006; Johannessen et al., 2004). Clouds strongly influence the atmospheric energy budget and are a primary source of uncertainty in the Arctic climate system (Huang et al., 2017; Tan and Storelvmo, 2019; Zib et al., 2012).

Arctic clouds are complex due to their complicated structure, complex interactions with various physical processes and feedbacks (Morrison et al., 2012; Kay et al., 2016). They influence the shortwave (SW) and longwave (LW) fluxes, thereby modulating the radiation and heat budgets. In the summer Arctic, the highly reflective surface and low sun angles enhance cloud-surface reflections (Curry et al., 1996; Barrientos Velasco et al., 2020). Therefore, an effective combination of observations and modelling is needed to better understand Arctic clouds and climate (Kay et al., 2016).

A key advantage of satellite instruments for investigating changes in the climate system is the provision of consistent observations covering relatively long time periods with near global spatial coverage (Stubenrauch et al., 2013; Christensen et al., 2016; Huang et al., 2017). For example, Hartmann and Ceppi (2014) find large changes in the Arctic radiation budget at the top of the atmosphere (TOA) based on CERES (Clouds and the Earth's Radiant Energy System) observations, with trends of -5 W m$^{-2}$ and 3 W m$^{-2}$ per decade for the SW and LW net fluxes, respectively. In contrast, Devasthale et al. (2016) report
large discrepancies and no significant trends in cloud data records based on the Advanced Very-High Resolution Radiometer (AVHRR). Hence, these observations require critical evaluation to identify their shortcomings before they can be used to diagnose processes responsible for Arctic climate change.

Due to the limited availability of ground-based observations in the Arctic, only few studies have compared ground- and satellite-based observations of clouds and radiative fluxes. Dong et al. (2016) present a radiative closure (RC) study comparing
ground-based and satellite retrievals of microphysical and radiative properties of single-layer clouds at the Atmospheric Radiation Measurement North Slope of Alaska (ARM NSA) site at Utqiaġvik, Alaska. Considering overpasses of the Terra and Aqua satellites from 2000 to 2006, and the CERES Synoptic 1-degree daily flux (SYN1deg, ed.2 and ed.4) products (Loeb et al., 2009; Rutan et al., 2015; Minnis et al., 2020), this study reports good agreement of retrievals of liquid water path ($Q_L$) and cloud optical depth ($\tau$) under both snow-covered and snow-free surface conditions. Using ARM and CERES cloud retrievals
as input for a radiative transfer simulations, the modelled downward SW fluxes (SWD) agree with the corresponding ARM observations and CERES products within 10 W m$^{-2}$.

The investigation by Riihelä et al. (2017) presents an intercomparison between ground-based observations and several satellite products of surface radiative fluxes. Downward and upward LW and SW radiative flux observations from the Tara drifting ice camp and long-term observations on the Greenland Ice Sheet are compared to the CERES SYN1deg ed.3A, FluxNet, and
Satellite Application Facility on Climate Monitoring cLoud, Albedo and RAdiation (CLARA) data sets (Karlsson et al., 2017). This study concludes that CERES SYN1deg has the smallest root-mean-square error (RMSE) compared against in-situ fluxes. This study recommends to further investigate differences in the surface and cloud properties that lead to discrepancies in flux retrievals.

Alongside RC studies with satellite products, ground-based long-term observations can provide independent estimates of
the cloud radiative effect (CRE) and thus show how clouds influence the radiation budget. For example, Ebell et al. (2020) simulated radiative fluxes based on more than 2 years (June 2016 to September 2018) of cloud properties retrieved from remote





sensing observations with the Cloudnet algorithms at the atmospheric observatory of the Arctic French–German research station AWIPEV (Alfred Wegener Institute for Polar and Marine Research and French Polar Institute Paul Emile Victor) at Ny-Ålesund, Norway (Nomokonova et al., 2019). Simulated and observed surface broadband LW and SW fluxes were compared

for all-sky (AS), clear-sky (CS), and cloudy conditions. For all-sky conditions, a mean difference of 3.1 $\mathrm{W\,m^{-2}}$ and 0.2 $\mathrm{W\,m^{-2}}$ was found for the SWD flux and the downward LW (LWD) flux, respectively, confirming RC for the study period. Based on the simulations, estimates of the CRE at the SFC ($\mathrm{CRE}_{SFC}$), TOA ($\mathrm{CRE}_{TOA}$), and within the atmosphere ($\mathrm{CRE}_{ATM}$) were derived. For the location of Ny-Ålesund, the annual averages revealed a surface warming by clouds of about 11.1 $\mathrm{W\,m^{-2}}$ and -16.1 $\mathrm{W\,m^{-2}}$ at the TOA, with significant variability observed in particular during summer.

In contrast to large-scale observations, field campaigns provide higher degree of detail to investigate relevant physical processes, and to study remote locations with previously poor observational coverage. The Surface Heat Budget of the Arctic Ocean (SHEBA) program investigated the physical processes governing the surface energy budget and sea-ice mass balance, covering the Beaufort Sea and a complete annual cycle from October 1997 to October 1998 (Uttal et al., 2002). Based on SHEBA data, Shupe and Intrieri (2004) determined the $\mathrm{CRE}_{SFC}$, considering in detail the influence of surface albedo and

cloud properties. This study found that low-level liquid clouds had the largest radiative effect. This study also recommended the use of polarization-sensitive lidars to improve the differentiation of ice crystals and cloud droplets and to generally improve retrieval algorithms for estimating liquid water content ($q_L$) towards a better understanding of the radiative effect of Arctic Clouds.

With the aim to better understand the physical mechanisms underlying Arctic Amplification, the project (AC)[3] (Arctic Am-

plification: Climate Relevant Atmospheric and SurfaCe Processes and Feedback Mechanisms) held two major field campaigns in the early summer of 2017 North-West of Svalbard (Wendisch et al., 2019). Both campaigns performed in-situ and remote sensing observations over the Arctic Ocean. While ACLOUD (Arctic Cloud Observations Using Airborne Measurements during Polar Day) was an aircraft campaign, the PS106 ship-borne campaign took place aboard the German Research Vessel (R/V) *Polarstern*. PS106 consisted of two legs, PASCAL (Physical Feedbacks of Arctic Boundary Layer, Sea Ice, Cloud, and

Aerosol; PS106/1) from 24 May to 21 June 2017, and with some observations continuing during SiPCA (Survival of Polar Cod in a Changing Arctic Ocean; PS106/2), from 23 June to 20 July (Macke and Flores, 2018). The mobile remote sensing platform OCEANET was set up on board *Polarstern* (Macke, 2009; Kalisch and Macke, 2012; Hanschmann et al., 2012; Kanitz et al., 2013; Griesche et al., 2020). The atmospheric observations collected with OCEANET were subsequently utilized to derive macro and microphysics properties of clouds based on the Cloudnet algorithm (Illingworth et al., 2007; Griesche et al., 2020).

The present paper aims to investigate the radiative effects of clouds and their influence on the radiation budget during the PS106 expedition. Radiative fluxes are simulated based on reanalysis profiles and observed cloud properties. They are compared to the CERES Synoptic 1-degree Ed. 4.1 products (from here on referred to as CERES), as well as ship-borne flux observations as a basis for a radiative closure assessment. From these data, an estimate of the CRE and the radiation budget during PS106 is given. Our investigation specifically attempts to answer the following questions:

1. How consistent are clouds properties derived from the ship-based observations and the CERES cloud products?





2. How closely do our radiative transfer simulations agree with the CERES surface fluxes and *Polarstern* flux observations under clear-sky conditions?

3. Under which circumstances is it possible to confirm radiative closure between flux observations and our radiative transfer simulations and the CERES products?

4. What was the radiation budget during PS106, how was it influenced by clouds as quantified by the CRE, and how representative were the local conditions for the entire Arctic?

The paper is structured as follows. First, Section 2 presents a description of the observations and methods. Results and Discussion follow this description in Section 3. Its first part summarizes the atmospheric conditions during PS106, followed by a sensitivity study to determine the uncertainties of the radiative transfer simulations. The differences between simulations and
100 observations are quantified for specific case studies, and the entire PS106 expedition is Sect. 3.3. and 3.4, respectively. Based on these results, Sect. 3.5 presents the estimated radiation budget and CRE for PS106 and the entire Arctic. Finally, the paper closes with conclusions and an outlook to future research in Section 4.

## 2   Observations and methods

This section describes the ship-borne observations and resulting cloud products, as well as the satellite-based cloud and radia-
105 tion products used as basis for this study. Furthermore, a description of ancillary data used for the radiative transfer simulations and to describe the surface and atmospheric conditions are presented alongside the method used for the radiative transfer simulations. In addition, a classification of sky conditions based on the ship-borne data sets and the setup used for the broadband radiative transfer simulations are introduced.

### 2.1 Ship-borne data set

**2.1.1 Ship-borne instrumentation**

The OCEANET atmosphere facility (hereafter denoted as OCEANET) was established to provide continuous atmospheric observations aboard research vessels such as the German icebreaker *Polarstern*. It has been operated since 2009 during transfer cruises between the hemispheres, crossing the Atlantic Ocean from Bremerhaven in Germany to Punta Arena in Chile and Stellenbosch or Cape Town in South Africa (Macke, 2009; Kalisch and Macke, 2012; Hanschmann et al., 2012; Kanitz et al.,
2013). In 2017, OCEANET was operated for the first time in the Arctic Ocean during two legs of the PS106 expedition named PASCAL and SiPCA, respectively (see Fig. 1; Macke and Flores (2018); Griesche et al. (2020)).

The instrumentation of OCEANET consists of a multi-wavelength Raman polarization lidar Polly$^{XT}$, a 14-channel microwave radiometer (MWR) HATPRO (Humidity And Temperature PROfiler), and a fish-eye sky camera. Complementing the standard OCEANET observations, a Doppler cloud radar of type Mira-35 was installed about 10 meters above the container
during PS106 cruise. The cloud radar can profile optically thick clouds and measure Doppler spectra produced by the verti-





cal cloud motion. These remote sensing instrumentation are combined with observations by an Optical Disdrometer of type OM470, and Vaisala RS92-SGP radiosondes launched every 6 hours as input to the Cloudnet processing (see Section 2.1.2; Griesche et al. (2020)).

On the roof of the measurement container, a CMP21 broadband pyranometer (0.285-2.8 $\mu$m) and a CGR4 broadband pyr-geometer (4.5-42 $\mu$m) both manufactured by Kipp & Zonen were also installed. For surface radiation measurements in polar regions, the accuracy of these instruments is expected to be within $\pm 10$ W m$^{-2}$ and $\pm 20$ W m$^{-2}$ for pyrgeometers and pyra-nometers, respectively (Lanconelli et al., 2011). In the case of the pyrgeometer, additional uncertainties due to variation in the vertically integrated water vapour (IWV) column need to be considered. At values below 10 mm of IWV, the atmospheric window produces spectral inhomogeneities, which distort the pyrgeometer measurements (Gröbner et al., 2014). Gröbner and

Wacker (2015) compared the performance of a calibrated World Infrared Standard Group (WISG) and a CGR4 pyrgeometer, and found that for specific periods, deviations can reach up to $\pm 3$ W m$^{-2}$. Thus, in this study, the total instrumental uncertainty for the pyrgeometer is assumed to be $\pm 13$ W m$^{-2}$. Operating these instruments aboard *Polarstern* might introduce additional uncertainties due to harsh environmental conditions like the exhaust plume of the ship, and the superstructures of the ship interfering with the observations (see Fig. A1).

In addition, a meteorological station measuring atmospheric pressure, temperature, and relative humidity with a sensor of type Series EE33 from E+E Elektronik was installed at about 10 meters above sea level (m.a.s.l) next to the OCEANET container during PS106. The measured values of this near-surface temperature are presented in Fig. 2b.

### 2.1.2 Cloudnet products and description of sky conditions

The Cloudnet project targets a systematic evaluation of the representation of clouds in the weather forecast and climate models

(Illingworth et al., 2007). Within its scope, a robust suite of algorithms has been developed for retrieving vertical profiles of macro and microphysical properties of clouds from a synergistic combination of cloud radar, lidar, and microwave radiometer observations. A complete description is given in Illingworth et al. (2007), while details about its adaptation for PS106 including a new approach for the continuous determination of the ice effective radius is presented in Griesche et al. (2020). This section provides a summary of the Cloudnet products used as the basis of the present study, i.e., the location of the cloud boundaries,

as well as vertical profiles of liquid water content ($q_L$), liquid effective radius ($r_{E,L}$), ice water content ($q_I$) and ice effective radius ($r_{E,I}$).

  As a first step, the measurements are averaged onto a common pixel grid with a vertical and temporal resolution of 31.18 m and 30 s, respectively (Griesche et al., 2020). Then, each pixel is categorized into seven distinct classes, following a bitwise diagnosis described by Hogan and Connor (2004). The final target classification provides 11 different pixel categories: clear

sky, cloud droplets only, drizzle or rain, drizzle or rain and cloud droplets, ice, ice and super-cooled droplets, melting ice, melting ice and cloud droplets, aerosols, insects, and aerosols and insects.

  The determination of the cloud thermodynamic phase is implemented following the methodology introduced in Griesche et al. (2020), which is based on the lidar and radar measurements (see Table 1). Liquid phase is assigned to pixels when the lidar backscatter signal exceeds a threshold of 2e−5 Mm−1sr−1, and is reduced at least by a factor of 10 within 250 m due





to the strong attenuation by liquid clouds. When the radar Doppler signal indicates falling particles at a dew point temperature
below 0 °C, the pixel is considered as ice. If both criteria are fulfilled simultaneously, the pixel is classified as mixed-phase.

The liquid water path ($Q_L$) is retrieved based on the HATPRO MWR measurements using the retrieval method developed
in Löhnert and Crewell (2003). This method relies on a long-term radiosonde training data set, which in this case is based
on Ny-Ålesund, NO (78.9°N, 11.85°E, WMO Code 6260). Once $Q_L$ is known, the liquid water content ($q_L$) and $r_{E,L}$ are
160 determined. The retrieval of $q_L$ is obtained by distributing $Q_L$ among the identified liquid and mixed-phase cloud pixels
identified by the Cloudnet algorithm. This method assumes a log-normal cloud-droplet distribution, which is constant with
height. The uncertainties of $q_L$ are calculated by error propagation assuming a typical uncertainty of 20-25 gm$^{-2}$ in $Q_L$
(Löhnert and Crewell, 2003).

The ice water content ($q_I$) is obtained based on the measurements from the cloud radar for pixels flagged as ice or mixed-
165 phase cloud (Hogan et al., 2006). This parameter depends on temperature (T; °C) and cloud radar reflectivity (Ze; dBZ). Based
on the values of $q_L$ and $q_I$, vertical profiles of cloud fraction are calculated. The $r_{e,I}$ is derived based on empirical relationships
between the visible extinction coefficient, cloud radar reflectivity and model temperature, as it is further described in Griesche
et al. (2020). Additionally, Cloudnet provides the cloud fraction averaged to hourly values for 67 equidistant height layers with
a thickness of 300 m, ranging from 150 meters up to 19.95 km (see Fig. 4a).

The Cloudnet products also contain quality flags that indicate when the pixels have a reliable retrieval, contain a mixture
of liquid droplets and ice crystals, and when large ice crystals, drizzle, or rain might bias the radar reflectivity. Precipitation
conditions compromise the retrieval accuracy of $Q_L$ retrievals from the MWR.

Three independent types of flagging categories are used to complement the analysis. These flags are based on Cloudnet target
classification, and the identification of low-level stratus clouds (LLS) from Griesche et al. (2020). Quality flags determine the
175 atmospheric conditions directly affecting the lidar beam or cloud radar signal. This characterization identifies "optimum condi-
tions", moments with "LLS clouds", "precipitation", and "precipitation and LLS". A second classification describes structural
flags by identifying "clear-sky", "single-layer", and "multiple-layer" clouds. Moreover, the last flag classification focuses on
the cloud phase. We identify 'clear-sky moments, clouds with "precipitation", "ice", "liquid", "mixed-phase clouds type 1"
on which ice and liquid droplets are separately identified, and "mixed-phase clouds type 2" on which Cloudnet distinguishes
a mixed layer of ice and super-cooled droplets in the same layer of cloud. Complex cloud systems like multiple layer mixed-
phase clouds can also be identified by the simultaneous use of structural and phase flags; however, they are outside the analysis
of the time series classification (Fig. 5).

The three classification types only focus on clear or cloudy pixels. Thus, as a first step, any Cloudnet pixel type of "aerosols",
"insects", and "aerosols and insects" is removed by changing its assigned value to zero, the value of a clear-sky pixel. Once
this step is done, a set of iterative conditional assignments is applied independently to each vertical column. For instance, if no
cloudy pixel is present in the atmospheric column, that column is flagged as clear-sky. If cloudy pixels are found in a single
column, and no clear-sky pixels are identified in between, then the algorithm classifies the column as a single layer. If one or
more clear pixels are identified between cloudy pixels, the flag of multiple levels is assigned. A similar method is followed,
but with different conditions for quality and phase flags. Lastly, the three types of flags are linearly interpolated to match the





temporal resolution of the simulations and stored separately in the output files of the radiative transfer simulations. Section 3.1 provides a broader description of the atmospheric conditions and analysis of the flagging system during PS106.

**2.2 Satellite data set**

The CERES product provides global SW and LW radiative fluxes at the TOA, at four pressure levels, and at the SFC interpolated to an hourly resolution. While TOA fluxes are directly based on observations by the CERES instruments, in-atmosphere
and surface fluxes are calculated based on the Fu-Liou radiative transfer model (Fu and Liou, 1992), and are adjusted for consistency with the TOA observations (Gupta et al., 2010; Rose et al., 2013; Rutan et al., 2015; Kato et al., 2018; Minnis et al., 2020). CERES uses cloud properties from geostationary and polar satellites observations, in particular from the Moderate Resolution Imaging Spectroradiometer (MODIS). As ancillary input, CERES relies on reanalysis data from the Global Modeling Assimilation Office Global Earth Observing System (GEOS-5) in version 5.4. In addition to radiative fluxes, various surface,
cloud and aerosol parameters are included in the product to enable an exploration of the relationships among clouds, aerosols, and radiation (Minnis et al., 2020).

Specifically, CERES provides several relevant cloud properties. The parameters considered in this study are the cloud fraction (CF), $Q_L$, $Q_I$, $r_{E,L}$, $r_{E,I}$, cloud base ($P_B$) and top pressure ($P_T$). Note that cloud properties mentioned are retrieved based on MODIS retrievals of cloud emissivity, cloud effective temperature, cloud particle effective radius, and cloud optical thickness.
A description of the retrievals is presented in Minnis et al. (2020). A summary of the cloud parameters used in this study is presented in Table 1.

Amongst the surface parameters included in CERES, the surface albedo ($\alpha$), skin temperature $Ts$, and snow/ice coverage are relevant for our analysis. In contrast to previous versions, the surface albedo is determined considering the 1.24 $\mu$m channel instead of the 2.13 $\mu$m channel over snow surfaces. This change has the advantage of increasing the range of retrievable
cloud optical depths over snow-cover areas (Sun-Mack et al., 2006). However, this modification also increases the uncertainty of surface albedo due to higher variability in snow albedo and bidirectional reflectance (Minnis et al., 2020). All CERES parameters are provided on a spatial grid with a resolution of 1° latitude by 1° longitude and at a temporal resolution of 1 hour.

**2.3 Radiative transfer simulations**

The TROPOS Cloud and Aerosol Radiative effect Simulator (henceforward T-CARS) is a Python-based framework to carry
out radiative transfer simulations with a particular focus on the investigation of the radiative effects of aerosols and clouds. Parts of this framework have already been applied and described in Barlakas et al. (2020) and Witthuhn et al. (2021). T-CARS enables the use of various sources for input data such as atmospheric profiles of trace gases, temperature, humidity, properties of clouds, aerosols, and surface parameters. The present study employs the widely used rapid radiative transfer model (RRTM) for GCM applications (RRTMG; Mlawer et al. (1997); Barker et al. (2003); Clough et al. (2005)).
In this study, the daily T-CARS output files have a standard grid that consists of 197 atmospheric levels ranging from the surface up to 20 km height and with 1-minute temporal resolution. The first 10 km of the atmosphere is divided into 160 levels





with a geometric layer thickness of 62.5 m. The following 5 km of the atmosphere have a layer thickness of 250 m, while the last 5 km of the atmosphere a layer thickness of 193.8 m.

Hourly pressure level profiles of temperature, pressure, ozone mass mixing ratio and specific humidity from the European
Centre for Medium-Range Weather Forecasts (ECMWF) Re-Analysis (ERA5) data set are used as input parameters for the simulations. ERA5 uses 4D-Var assimilation using polar and geostationary satellites, surface, radiosonde, dropsonde, and aircraft measurements (Hersbach et al., 2020). In the case of PS106, the Vaisala Radiosonde RS92-SGP launched from *Polarstern* every 6 hours were also assimilated.

The atmospheric temperature and pressure measured at 10 m.a.s.l aboard *Polarstern* are used in T-CARS as skin temperature
and surface pressure, respectively. We opted for this set-up since these parameters are the closest to the surface and are the only measurement that is available for the entire cruise. For comparison purposes, the surface temperature from ERA5 and CERES are also considered.

The surface albedo for the radiative transfer simulations is based on CERES. Additionally, surface albedo from ERA5 is considered for comparison. Both data sets are interpolated in space and time to the position of *Polarstern* at 1-minute resolution
the entire PS106 cruise.

All input data sets are interpolated to the standard grid using linear interpolation. For trace gases, the climatological values from the Air Force Geophysics Laboratory (AFGL) sub-Arctic summer atmosphere (Anderson et al., 1986) are used. The pressure level ERA5 data set is used in the model by interpolating atmospheric pressure, temperature, specific humidity, and ozone mass mixing ratio. Considering the Cloudnet cloud properties, $r_{E,L}$ and $r_{E,I}$ are linearly interpolated onto the standard
grid. The values of $q_L$ and $q_I$ are converted to values of $Q_L$ and $Q_I$ by multiplying with the layer thickness.

For the LW and SW simulations, the RRTMG parameterizations for ice and liquid cloud optical properties are based on the radiative transfer model Streamer (Key, 1996) and Hu and Stamnes (1993), respectively, have been selected. The parameterization for ice clouds assumes spherical ice crystals with $r_{E,I}$ values with an allowed range between 5.0 and 131.0 $\mu$m. In the case of $r_{E,L}$, the model only allows values within the range from 2.5 to 60 $\mu$m. To maximize data coverage, the Cloudnet $r_{E,L}$
values below 2.5 $\mu$m have been clipped to this range. Note that this modification increases the original values by less than 0.05 % of the number of observations. Therefore, this choice modification does not significantly change the distribution or mean value of $r_{E,L}$.

The surface input parameters, such as the ship measurement of pressure, temperature, and the albedo from CERES, are also interpolated to 1-minute resolution. The surface emissivity is set to a constant value based on the fraction of sea ice in
the vicinity of *Polarstern*, using CERES as source. When the sea ice fraction exceeds 50 %, a constant surface emissivity of 0.9999 is used, while a value of 0.9907 is used below this threshold. These constant values are based on Wilber et al. (1999).

The T-CARS output provides vertical profiles of broadband upward and downward, LW and SW fluxes, and heating rates for cloudy and clear-sky conditions along the PS106 cruise track. Additionally, the geographic coordinates, the quality flags mentioned in Sect. 2.1.2, profiles of temperature and pressure levels, as well as and the cloud top and cloud base height obtained
from the Cloudnet data set and the cloud boundaries from the analysis of LLS described in Griesche et al. (2020) are included as output variables.





The present study defines the CRE as the difference between the all-sky and clear-sky net fluxes in $\mathrm{W\,m^{-2}}$, following, e.g., Mace et al. (2006). The net CRE is obtained as the sum of the $\mathrm{CRE}_{LW}$ and $\mathrm{CRE}_{SW}$ components, which are calculated using Eq. (1). In this equation, "$x$" stands for either LW or SW and is computed both at the SFC and TOA. Given the net $\mathrm{CRE}_{SFC}$

and $\mathrm{CRE}_{TOA}$, the net CRE throughout the atmosphere ($\mathrm{CRE}_{ATM}$) is obtained by subtracting of the values TOA and at the SFC.

$$CRE_x = (F_x^{\downarrow} - F_x^{\uparrow})_{all-sky} - (F_x^{\downarrow} - F_x^{\uparrow})_{clear-sky} \tag{1}$$

## 3 Results and Discussion

The main results of the present investigation are described and discussed in this section. First, the atmospheric and surface con-

ditions during PS106 are described in Sect. 3.1. In Sect. 3.2, a sensitivity study of the radiative fluxes in clear-sky conditions is given in order to quantify the expected uncertainty of the radiative transfer simulations and to quantify the effect of atmospheric and surface variability on fluxes. Sect. 3.3 presents three case studies, comparing our own radiative transfer simulations based on T-CARS, the CERES-based flux products and ship-based flux observations. An assessment of RC for the CERES fluxes considering the entire PS106 cruise is given in Sect. 3.4. The radiation budget and its modulation by clouds as quantified by

the CRE for the PS106 expedition is investigated in Sect. 3.5. Results obtained along the ship track and for the whole Arctic are compared in order to assess the representativeness of the observations conducted during the PS106 expedition.

### 3.1 Atmospheric and surface conditions

A general description of the meteorological and synoptic conditions during the PASCAL expedition (leg 1 of PS106) has already been given by Knudsen et al. (2018). Here, a complementary and more specific description is provided, focusing in

particular on aspects influencing radiative fluxes, as well as the in-situ observations, satellite and ancillary data sets required as input for radiative transfer simulations.

Time series of surface albedo, near-surface and skin temperature along the PS106 cruise track are presented in Fig. 2. The track covered open ocean, the marginal ice zone, and ocean covered by dense sea ice. These regions can be differentiated by their surface albedo, which has been obtained here from the CERES and ERA5 data sets (see Fig. 2a). The retrieval of

surface albedo used by CERES is described in Minnis et al. (2020) and references therein. Based on the sea-ice concentration, the ERA5 broadband surface albedo is calculated as a linear combination of the sea-ice and open water contributions. For the considered time period, the sea-ice concentration in ERA5 is based on the corresponding operational product provided by EUMETSAT's Ocean and Sea Ice Satellite Application Facility (OSI SAF; Eastwood et al. (2014)) and the Operational Sea surface Temperature and Ice Analysis (OSTIA) data set (Hirahara et al., 2016). For PS106, there is, in general, a relatively

good agreement between both surface albedo data sets along the cruise track, with a standard deviation of about 0.1. The largest differences occur during the second leg for the period from 25 June to 8 July 2017, when the CERES albedo is systematically higher than the ERA-5 based values. Some of the observed differences might be attributable to the different spatial resolutions





of the data sets (1° for CERES versus 0.25° for ERA5). Another potential cause for discrepancies is the omission of the effect of melt-ponds in ERA5, which results in a systematic underestimation of the broadband surface albedo by ERA5 from June to mid-August, as discusses in Pohl et al. (2020).

The CERES surface albedo is used later as input for the radiative transfer simulations. It is based directly on solar reflectance observations, and due to the maturity of CERES products, it is expected to yield accurate results (Rutan et al., 2015). Its use also ensures consistency with the CERES flux products and allows us to better focus on the influence of other parameters. Nevertheless, the large influence of surface albedo on the solar radiation budget is investigated and discussed in more detail (see Section 3.5).

The skin temperature and the near-surface air temperature are warmest at the beginning of the expedition when *Polarstern* was located over open ocean. For the rest of the expedition, the temperatures maintained a relatively steady value near 270 K, except for 8 June and 3 July 2017, when the temperatures dropped to around 266 K. In general, all temperatures presented in Fig. 2b are in good agreement, except for 29, 9-10, 22-23 June, 3-4, and 15 July 2017. Most of these differences might be due to local variability, as there is good agreement of onboard measurements and radiosondes. The largest difference between CERES and ERA5 is found for 23 June 2017, which was the start of the second leg where *Polarstern* was located in Svalbard (see Fig. 1). This difference might be due to the challenges posed by a realistic representation of the marginal sea ice zone.

The anomalies of the vertical profiles of atmospheric temperature and specific humidity based on ERA5 are shown in Fig. 3 for the PS106 track, together with the mean profiles and the sub-Arctic summer standard atmosphere (Anderson et al., 1986).

Panels 3a and 3b, indicate a strong temperature and humidity inversion layers from 1 June to 9 June 2017, followed by a generally warmer atmosphere from 10 June to 14 June 2017. Moreover, relatively warm and humid conditions are observed at the end of leg 2 (11 July to 16 July 2017), caused by water vapour transport as described in Knudsen et al. (2018) and Viceto et al. (2021). Based on the ship-borne observations, the near-surface temperature varied more strongly during leg 1 than during leg 2 of PS106 (see Fig. 2b). Most of the humidity intrusions observed in Fig. 3b have southerly and westerly origins of the

advected air masses, based on the wind direction obtained from the radiosondes (not shown). The mean vertical profiles of atmospheric temperature and specific humidity indicate good agreement of radiosondes and ERA5, which are colder and dryer than the climatological values of the sub-Arctic summer standard atmosphere.

A characterisation of cloud conditions during the PS106 expedition is given next, considering CF, vertical layer structure and thermodynamic phase. The Cloudnet vertical profiles of CF are shown in Fig. 4a, while a comparison of daily mean CF is

315 presented in Fig. 4b. For the latter panel, Cloudnet and LLS clouds are combined in the comparison to CF values of CERES. All CF values have been aggregated from hourly to daily means. To ensure consistent temporal sampling, hourly values with data gaps in the ship-borne observations have been excluded from CERES. It is worth mentioning that the combination of LLS and Cloudnet CF improved the analysis by reducing the fraction of data gaps from 25.2 % to 6.6 % for the entire PS106 period.

The comparison aims to determine the consistency of the CERES and Cloudnet cloud fraction, despite their different instru-

320 mental origin, perspectives, spatial and temporal sampling, and retrieval algorithms. It is worth noting that CERES provides a spatially averaged cloud fraction for a 1° latitude by 1° longitude region, while Cloudnet yields vertically resolved information on cloud cover as time series for the location of *Polarstern*, as shown in Fig. 4a. Mean values of cloud fraction are 86.7 % and





76.1 % for CERES and Cloudnet plus LLS, respectively. The CERES cloud fraction without exclusion of Cloudnet data gaps is higher at 86.9 %, indicating that data gaps occur more frequently in cloudy conditions.

A description of the type of clouds observed during PS106 is shown in Fig. 5. The used classification is based on the Cloudnet target classification and supplemented by the analysis of LLS clouds. While the cloud type classification was already presented in Fig. 18c of Griesche et al. (2020), data quality and a more detailed description of mixed-phase clouds are considered here. Details of the classification methodology for the three flags are explained in Section 2.1.2.

During PS106, approximately 45.4 % and 35.6 % of the time, single and multilayer clouds were observed by Cloudnet,
respectively (Fig. 5a). The remaining 12.4 % of the time, clear-sky conditions were detected, and data gaps occurred for 6.6 % of the time. While single-layer clouds were more frequent during the first leg, the frequency of multilayer clouds was higher during the second leg (see Fig. 5a and Fig. 18c in Griesche et al. (2020)). The frequencies of single-layer and multilayer clouds are typical for early summer conditions, as previously reported by other studies (e.g., Shupe et al. (2011); Nomokonova et al. (2019)).

The cloud phase flag was included to analyse the thermodynamic phase of clouds. Even though it is available for the entire PS106 time series, the focus is here directed to the thermodynamic phase of single-layer clouds, since these cases are the most frequent, and an analysis is less complex than for multilayer conditions. Fig. 5b indicates an occurrence frequency of 32.6 % for single-layer mixed-phase clouds of type two (ice and super-cooled droplets), 16.8 % for mixed-phase clouds of type one (well-separated ice and liquid phase), and 15.6 % for single layer ice clouds. The remaining period is composed of single-layer
clouds with precipitation (13.6 %), clear-sky periods (12.1 %), and single layer liquid clouds (2.7 %).

The quality status flag shows that only 40.1 % of the time, optimum observation conditions were identified (see Fig. 5c). For about 39.0 % of the time, low-level stratus (LLS) with a cloud base below 155 m prevailed, implying that during these periods, the cloud base height lay below the altitude detection limit of 155 m of the cloud radar (see Fig. 18a in Griesche et al. (2020)). Precipitation (PPT) alone and moments with LLS and PPT occurred for about 5.1 % and 9.2 % of the time, respectively. Under
LLS and PPT conditions, the observations of cloud macro and microphysical parameters have a reduced accuracy. Thus, for only about 40.1 % of the time, observations are of sufficient quality, e.g., for radiative transfer simulations, while observations with a degraded quality occurred for 59.9 % of the period.

### 3.2 Sensitivity analysis of clear-sky radiation

This section presents a sensitivity analysis of the radiative fluxes at the surface for clear-sky conditions. Its goal is to quantify
the response of radiative fluxes to variability and uncertainty of various atmospheric and surface parameters during PS106, which are required as input to radiative transfer simulations. The accuracy of clear sky radiative fluxes is of particular interest, as it serves as a reference for the calculation of the CRE and thus places a limit on its accuracy. The propagation of the input uncertainties to radiative fluxes is thus used to establish uncertainty limits for the subsequent RC study.

A clear-sky atmosphere has been created as a reference case, based on the conditions at a position of 81.9° N and 32.51°
E on 3 July 2017. This day was selected because it is the day with the longest clear-sky period during PS106 (see Fig. 1, Fig. 4 and Fig. 5). For the sensitivity analysis, all atmospheric and surface parameters required as input were prescribed by the




actual conditions, with the exception of surface albedo, which was kept constant and set to the daily mean value of 0.65. The intention is to avoid resulting variations in fluxes due to fluctuations of the surface albedo, which ranged from 0.58 to 0.78 on that day (see Fig. 2a). For the day, the CERES ice coverage lay above 0.5. Thus, the surface emissivity was set to a constant

value of 0.9999. The solar zenith angle (SZA) ranged from 59° to 75°, which does not cover the full range of SZA encountered during the PS106 expedition (47.6° to 80.1 ° for the period from 31 May to 16 July 2017). Despite such minor discrepancies, we assume here that the conditions of this reference day are representative for the entire PS106 cruise for the purpose of this sensitivity analysis.

The response of radiative fluxes to perturbations of atmospheric parameters, including temperature, ozone, and humidity,

has been quantified. In addition, the effects of variations of skin temperature, surface albedo, and surface emissivity are also considered. This analysis focuses on both the downward and upward fluxes at the surface, as well as the upward fluxes at the TOA for both the LW and SW broadband radiation. A summary of the main sensitivities is presented in Fig. 6, while the full results are listed in the Appendix as Tables A1 to A4. Note that the values used later as uncertainty limits for the RC analysis are highlighted in bold.

The perturbation of atmospheric temperature is based on the instrumental uncertainty of $\pm0.5$ K of the radiosonde temperature sensor, as well as the observed range of atmospheric temperature anomalies of about $\pm7$ K during PS106 (see Fig. 3c). The variation of $\pm0.5$ K has the strongest effect on the LWD flux at the surface, resulting in a variation of $\pm1$ W m$^{-2}$. Note that 3 July 2017 is a day with slightly colder than average temperatures for the first 8 km of the atmosphere, and warmer than average temperatures at about 10 km (see Fig. 3a). Considering the larger temperature perturbation of $\pm7$ K results in a

variation of up to $\pm14.9$ W m$^{-2}$ (see Table A1). At the TOA, the variations of $\pm0.5$ and $\pm7$ K yield daily mean differences of $\pm1.1$ W m$^{-2}$ and $\pm15.4$ W m$^{-2}$, respectively.

Ozone reduces the SW flux in the atmosphere by absorption at ultraviolet ($\lambda \lesssim 0.35$ $\mu$m) and visible wavelengths (0.5 $\mu$m $\lesssim \lambda \lesssim 0.7$ $\mu$m). To quantify the sensitivity to ozone, the findings of Bahramvash Shams et al. (2019) are used as basis, who investigated the variation of the vertical profiles of ozone at four Arctic sites from 2005-2017. Perturbation of the ozone column

amount by $\pm12.5$ % and $\pm25$ % are assumed. The former value is used to approximate the monthly variation of ozone during summer months (see Fig. 5 in Bahramvash Shams et al. (2019)), which is similar to the uncertainty of $\pm10$ % ozonesondes (Deshler et al., 2017). The smaller variation leads to a decrease of $\pm0.7$ W m$^{-2}$ and $\pm0.3$ W m$^{-2}$ at the SFC and TOA for the SW flux, respectively, and $\pm0.3$ W m$^{-2}$ both at the SFC and the TOA for the LW. Variations of ozone concentration are particularly important in the stratosphere, where reduced ozone causes colder temperatures, which enhances the decrease of

ozone (Randel and Wu, 1999). It is also worth noting that ozone concentration is linked to synoptic mechanisms through interactions with atmospheric dynamics and photochemistry (Anstey and Shepherd, 2014).

Water vapour is the dominant absorber throughout most of the LW (Delamere et al., 2010) and has strong absorption bands in the SW ($\lambda > 900$ nm). The strength of SW absorption also depends on the SZA (Wyser et al., 2008). Our sensitivity analysis considers a variation of 5 %, which is used to represent the instrumental uncertainty of radiosondes, and 15 %, which

approximates the range between the minimum and maximum column amounts. The variation of 5 % leads to differences of $\pm1$ W m$^{-2}$ and $\pm0.9$ W m$^{-2}$ for the SWD and LWD, respectively, at the surface. At the TOA, this perturbation yields





differences of $\pm 0.5\,\mathrm{W\,m^{-2}}$ and $\pm 0.7\,\mathrm{W\,m^{-2}}$ for the upward LW (LWU) and SW (SWU), respectively (see Table A1). The column amount of water vapour is largest in July, and a positive trend in water vapour since the 1990s has been reported, especially during summer (Di Biagio et al., 2012; Rinke et al., 2019). In addition, recurrent episodes of humidity intrusions

can increase the water vapour column by about 8 times above the background level and increase the LWD flux at the surface by up to $16\,\mathrm{W\,m^{-2}}$ (Doyle et al., 2011).

During PS106, the temperature measurement onboard *Polarstern* closest to the surface was located at 10 m above sea level. The accuracy of the temperature of $\pm 0.3\,^{\circ}\mathrm{C}$ is used for the sensitivity study, together with a perturbation of $\pm 5\,^{\circ}\mathrm{C}$, which corresponds to the largest difference between ship measurements and skin temperature obtained from ERA5 and CERES (see

Fig. 2b). The variation of this parameter only influences the LW fluxes, causing changes of $\pm 0.3\,\mathrm{W\,m^{-2}}$ and $\pm 1.4\,\mathrm{W\,m^{-2}}$ for the LWD and LWU, respectively, at the SFC. At the TOA, a difference of $\pm 0.5\,\mathrm{W\,m^{-2}}$ in the LWU. Naturally, more considerable variation in skin temperature yields to larger flux differences as indicated in Table A2 and Table A4, which is particularly relevant for days when these differences are more pronounced (e.g., 6 to 8 June, 22 to 23 June, 2 to 5 July 2017; see Fig. 2b).

Surface albedo is an extremely important parameter for the SW radiative fluxes (Shupe et al., 2005; Sedlar et al., 2011; Ebell et al., 2020; Stapf et al., 2020). The retrieval of this parameter from satellites in the Arctic is particularly challenging due to the difficulty of cloud detection over snow- and ice-covered surfaces and rapid temporal changes induced by melting. In contrast, ground-based observations often have limited spatial representativeness. Figure 2a shows the time series of hourly surface albedo from CERES and ERA5. From 27 June to 8 July 2017, the difference is most noticeable. For the PS106 cruise, the

mean difference of surface albedo between CERES and ERA5 is 0.08. For the sensitivity study, the daily mean value of 0.65 is used for the surface albedo. Variations by $\pm 0.08$ are used to investigate the sensitivity of radiative fluxes. Additionally, the minimum (0.05), mean (0.53), maximum (0.84) values of the CERES surface albedo for the entire cruise are used to quantify the sensitivity of SW fluxes to surface albedo during PS106 cruise.

A variation of the surface albedo by $\pm 0.08$ yields a mean flux difference of $\pm 2.1\,\mathrm{W\,m^{-2}}$ and $\pm 33.6\,\mathrm{W\,m^{-2}}$ at the SFC for

the SWD and SWU, respectively. This difference also depends strongly on the SZA. For instance, a SZA of $59^{\circ}$ leads to a flux difference of the SWU at the SFC of $\pm 46.6\,\mathrm{W\,m^{-2}}$, whereas at a SZA of $75^{\circ}$ this difference is $\pm 20.8\,\mathrm{W\,m^{-2}}$ (not shown). The values presented in Table A2 and A4 for the surface albedo also indicate the large contrast between the additional values studied (0.84, 0.53, 0.3, and 0.05) minus the constant surface albedo of 0.65. At the SFC, open ocean (e.g., a surface albedo equal to 0.05) causes a large reduction of the SWD and SWU by $13.9\,\mathrm{W\,m^{-2}}$ and $241.1\,\mathrm{W\,m^{-2}}$, respectively. On the other

hand, the highest surface albedo observed during PS106 (0.84) results in an increase of the SWD and SWU of $5.1\,\mathrm{W\,m^{-2}}$ and $8.4\,\mathrm{W\,m^{-2}}$, respectively. At the TOA, the daily mean values indicate a reduction by $217.6\,\mathrm{W\,m^{-2}}$ and an increase by $72\,\mathrm{W\,m^{-2}}$ for surface albedos of 0.05 and 0.84, respectively. To the best of our knowledge, the importance of surface albedo under clear-sky Arctic conditions has also been analysed in other studies (Wyser et al., 2008; Di Biagio et al., 2012; Sedlar and Devasthale, 2012). However, the analysis of the SWU flux was not further investigated individually. Due to the values

mentioned, the SWU is the most relevant parameter sensitive to large differences when estimating the radiation budget and the cloud radiative effect.



The last parameter considered in the analysis is the surface emissivity. A value of 0.9999 is used for surfaces covered by ice, while a value of 0.9907 is used for water surfaces based on Wilber et al. (1999). Additionally, the constant value of 0.988 used by Riihelä et al. (2017) is considered, who used this value for a wider Arctic area, based on Hori et al. (2006). The variation of

this parameter only affects the LWU. Contrasting the surface emissivity over sea ice and open-ocean, a mean flux difference of $0.8 \ \mathrm{W \, m^{-2}}$ and $0.7 \ \mathrm{W \, m^{-2}}$ at the SFC, and TOA, respectively, are found. The comparison with the value used in Riihelä et al. (2017) yields a smaller difference of $0.2 \ \mathrm{W \, m^{-2}}$ at the SFC and $0.1 \ \mathrm{W \, m^{-2}}$ at the TOA.Scatter plots comparing the $\mathrm{CRE}_{SFC}$ with several parameters. Panel (a) shows the comparison with cloud fraction (c). Panel (b) shows the comparison of surface albedo ($\alpha$). Panels (c) and (d) show the comparison between the liquid water path ($Q_L$) at low surface albedo ($<0.4$) and high

surface albedo ($>0.4$). All scatter plots are colour coded with the values of the solar zenith angle (SZA).

A summary of the results of the sensitivity study is shown in Fig. 6. Considering the propagation of the uncertainty of input parameters to radiative fluxes, a total uncertainty of the LWD and SWD fluxes at the surface of $\pm 2.6 \ \mathrm{W \, m^{-2}}$ and $\pm 3.7 \ \mathrm{W \, m^{-2}}$, respectively, is inferred. The largest part of this uncertainty is introduced by the amount of water vapour and the atmospheric temperature for the LWD, and by the surface albedo for the SWD. Combining these values with the instrumental uncertainties

presented in Sect. 2.1.1, total uncertainty limits of $\pm 16 \ \mathrm{W \, m^{-2}}$ and $\pm 24 \ \mathrm{W \, m^{-2}}$ for the LWD and SWD, respectively, at the surface are inferred as basis of the subsequent RC assessment for clear-sky conditions.

### 3.3 Case studies

The following subsection presents 3 case studies, which have been selected to cover typical sky conditions during PS106. Radiative fluxes and the CRE at the SFC obtained from the T-CARS simulations, CERES products and the ship-borne observa-

tions are compared to illustrate periods with a good agreement and larger discrepancies. A day with a long clear-sky period, a day with single, multilayer and mixed-phase clouds are presented. For the cloud cases, the comparison of fluxes is accompanied by a comparison of cloud properties to investigate their role as a potential source for observed differences in fluxes.

For these cases, it is assessed whether RC can be reached between the ship-borne observations on the one hand and the T-CARS simulations and CERES products on the other hand. For clear-sky conditions, the uncertainty limits have been estab-

lished in Sect. 3.2. In contrast to the RC assessment at the surface, the lack of independent observations and inputs only allows an assessment of the consistency of fluxes simulated by the T-CARS setup and CERES.

A common figure layout is used to give an overview of meteorological conditions and radiative fluxes to present the case days in Figs. 7, 8 and 10. Their first panels show the Cloudnet target classification overlaid with the cloud top and base heights obtained from CERES as time series. Panels (b) and (c) show the time series of downward (upward) fluxes at the

SFC (TOA) for the SW and LW fluxes averaged to 10-minute temporal resolution. Background shading is used for periods when environmental conditions or instrumental limitations compromised the observations. The pink background indicates periods when the ship-borne flux observations were obstructed by *Polarstern's* superstructure, mainly affecting the SW flux. The pale yellow background is used for periods when the Cloudnet retrievals were unable to correct for attenuation, and light blue indicates the periods of clear-sky conditions that are agreeable between CERES and Cloudnet. Panels (d) and (e) show

the distribution of the flux difference between ship-borne observations and simulations for the SWD and LWD, respectively.





Panels (f) shows the comparison between CERES products and T-CARS simulations for the upward SW flux, panel (g) shows the same as (f), but for the LW flux.

The comparison of the microphysical properties of clouds such as water path ($Q$) and effective radius ($r_E$) between Cloudnet and CERES are shown in panels (a) and (b), respectively, in Figs. 9 and 11. Additionally, the time-series of the CRE at the SFC and the TOA based on T-CARS and CERES are presented in panel (c) of Figs. mentioned.

### 3.3.1 Clear-sky case: 3 July 2017

As mentioned before, 3 July was the day with the longest cloud-free period during PS106 (see Fig. 4, Fig. 5, and Fig. C1). An overview of meteorological conditions and radiative fluxes for this day is given in Fig. 7. Panel (a) shows that ice clouds were observed by Cloudnet from 00:00Z to around 02:30Z on that day. Later periods when CERES reported clouds while Cloudnet did not were further analysed by means of the all-sky camera images. They showed a thin cloud at the horizon for the period between 21:30Z to 23:59Z, which is outside the field of view for the zenith-pointing active remote sensing instruments.

Good agreement is found for the ship-borne flux observations, the T-CARS simulations, and the CERES products for the SWD at the SFC. The mean difference of the simulations minus the observations lies below $8 \, \mathrm{W \, m^{-2}}$, a value that is well within the uncertainty limit of $\pm 24 \, \mathrm{W \, m^{-2}}$ used for the RC assessment. For the LWD fluxes, there is a larger difference between the ship-borne observations on the one hand and the T-CARS simulations and CERES products on the other hand. However, the mean difference remains below $14 \, \mathrm{W \, m^{-2}}$, again smaller than the uncertainty limit of $\pm 16 \, \mathrm{W \, m^{-2}}$ chosen for the RC assessment. Therefore, our results confirm that RC is reached for both the downwelling SW and LW fluxes for this clear-sky case.

To analyse the consistency of T-CARS simulations and CERES products at the TOA, the SWU and LWU fluxes are considered. Figure 7a shows a similar temporal behaviour of the SWU fluxes for the considered period. The mean difference between T-CARS and CERES is $10.0 \, \mathrm{W \, m^{-2}}$, with the largest instantaneous differences occurring after 18:40Z. Since the T-CARS simulations use the same surface albedo as CERES as input, the reason for this difference is most likely due to the spatio-temporal interpolation of CERES (Young et al., 1998). For the LWU fluxes, the mean difference is $3.7 \, \mathrm{W \, m^{-2}}$, possibly due to differences in the CERES skin temperature and the ship-borne measurement of near-surface temperature used by T-CARS (see Fig. 2). The mean differences of the radiative fluxes are sufficiently small to confirm the consistency of CERES and T-CARS fluxes, in agreement with similar studies which compared CERES with other radiative transfer simulations (e.g., Dong et al. (2016); Dolinar et al. (2016)).

### 3.3.2 Single and multilayer ice cloud case: 2 July 2017

Single and multilayer clouds were present for 45.4 % and 35.6 % of the time during PS106, respectively (see Fig. 5b). 2 July 2017 has been chosen as case day with a dominant presence of this type of clouds. Based on the Cloudnet target classification, this day was characterized by well-defined single and multilayer ice clouds (see Fig. 8a and Fig. C2). Middle and high-level ice clouds were observed for most of the day, with an average cloud base at or above 2.6 km. An exception is the period from 5:57Z to 8:35Z, when a relatively thin cloud layer consisting of ice and super-cooled droplets was identified (see Fig. 8a). For



most of the day, the cloud top height from CERES is significantly lower than that from Cloudnet. The cloud base height is
relatively close to the base obtained from the Cloudnet target classification. It is well-known that the retrieval of cloud top
height from passive satellite instruments is limited by large uncertainties for thin ice clouds and polar regions (e.g. Yost et al.,
2020), so these discrepancies are not surprising.

A comparison of CERES and T-CARS fluxes against observations for the SWD and LWD reveals good agreement, with val-
ues below the clear-sky uncertainty limits established in Sect. 3.2 (see Table 2). This comparison suggests that RC is achieved
for T-CARS and CERES. At the TOA, however, there is a more significant difference of -15.4 $\mathrm{W\,m^{-2}}$ and -15.0 $\mathrm{W\,m^{-2}}$ for
the LWU and SWU, respectively. This difference is likely linked to the differences in cloud properties retrieved by Cloudnet
and CERES that are displayed in Fig. 9. Discrepancies in other parameters such as the skin temperature, atmospheric profiles,
cloud top height, and cloud geometrical thickness might also be relevant.

Panels a and b of Fig. 9 show the time series of the $Q$ and $r_e$ obtained from Cloudnet and CERES, respectively. Despite the
difference in retrieval methods, there is, in general, a good agreement of the values of $Q_I$ and $r_{E,I}$ from CERES and Cloudnet
(see Fig. 9a and 9b). Based on CERES, the entire day was characterized by the presence of a mixed-phase cloud. The values
shown for $Q_L$ are relatively large, especially during the period from 17:00Z to 19:00Z. It is possible that within the CERES
footprint, a cloud with a such large $Q_L$ might have been present. Nevertheless, it is worth noting that the fluxes shown in Fig.
8b and (c) are calculated considering the cloud fraction, which remained below 15 % from 12:30Z until the end of the day
(not shown). The latter suggests that care must be taken when comparing cloud properties obtained from the ship-borne active
instruments and from CERES with its coarse spatial footprint.

For this case study, it can be concluded that the net $\mathrm{CRE}_{SFC}$ is consistent between T-CARS and CERES, despite the
noted discrepancies in cloud properties. The net $\mathrm{CRE}_{SFC}$ has values of 1.3 $\mathrm{W\,m^{-2}}$ and 2.7 $\mathrm{W\,m^{-2}}$ for CERES and T-CARS,
respectively. At the TOA, the difference is more significant due to the mentioned differences in the LW. The net $\mathrm{CRE}_{TOA}$
derived by T-CARS is 8.9 $\mathrm{W\,m^{-2}}$, whereas CERES suggests a cooling by -11.1 $\mathrm{W\,m^{-2}}$. Such inconsistencies need to be
clarified and further investigated in future studies.

### 3.3.3 Mixed-Phase clouds: 26 June 2017

The last case study chosen is 26 June 2017. This day was selected due to the presence of well-defined cloud layers consisting
of ice and liquid droplets, corresponding to a mixed-phase cloud of type 1. A further reason is the long period of optimum
observation conditions reported by Cloudnet (Fig. 5c). Moreover, this day is also of interest due to an underestimation of high-
level cloud amount by CERES in comparison to Cloudnet, as is corroborated in Fig. 10a. This day is also characterized by
changing surface conditions, as the ship crossed through the sea-ice transit zone (see Fig. 1, Fig. 2a) and Fig. C3).

This day was characterized by a low-level cloud located within the first 2 $\mathrm{km}$ of the atmosphere, and by several periods with
a relatively thin ice cloud layer located between 6 and 9 $\mathrm{km}$ height (Fig. 10a). According to Cloudnet, the well-separated liquid
phase layer within the ice clouds is characterized by low radar reflectivity values, upward-directed Doppler velocity, and high
lidar backscatter. There were two moments around 10:00Z and 23:00Z where, due to uncorrected attenuation, $Q_L$ could not





be derived by Cloudnet (Fig. 10a and b). These two moments, marked by the pale-yellow shaded areas, are excluded from the T-CARS and CERES histogram analysis for a fair comparison (Fig. 10d, e, f, g).

Panel (b) and (c) in Fig. 10 indicate a good agreement between observations and CERES fluxes. The daily mean difference
of fluxes is 16.1 W m$^{-2}$ and -0.6 W m$^{-2}$ for the SWD and LWD, respectively. For the T-CARS simulations, the mean flux difference is significantly larger, with 77.1 W m$^{-2}$ for the SWD and -12.0 W m$^{-2}$ for the LWD (see Table 2). Based on these results, RC can be confirmed for the LWD and SWD fluxes from CERES products and the LWD T-CARS simulations. The difference in the T-CARS SWD fluxes exceeds the expected uncertainty limits. To investigate the reasons for this, the cloud properties from both data sets are compared.

Panels (a) and (b) of Fig. 11 show the time series of cloud properties from the Cloudnet and CERES data sets. In contrast to Cloudnet, CERES reports only three periods with the presence of a mixed-phase cloud. The largest difference in cloud properties occurs in the cloud water path products. For Cloudnet, the mean $Q_L$ is 56.2 g m$^{-2}$, and for $Q_I$, the mean is 1.9 g m$^{-2}$. In the case of CERES, these values are 119.7 g m$^{-2}$ and 38.1 g m$^{-2}$ for $Q_L$ and $Q_I$, respectively. The variable and lower values of $Q$ are likely responsible for the rapid changes and the positive (negative) bias of the SWD (LWD) flux visible in Fig. 10a
and b.

The radiative effect of clouds on this day has a strong cooling influence both at the SFC and TOA that is enhanced by the surface albedo. In Fig. 11c. An abrupt change of the CRE at the SFC and the TOA is visible at 05:00Z in Fig. 11c, due to a simultaneous rapid reduction of surface albedo from a value of 0.6 to 0.27 (see also Fig. 2a).

Based on CERES, a daily-mean net CRE of -79.5 W m$^{-2}$ and -127.9 W m$^{-2}$ is found at the surface and TOA. The T-CARS
simulations also indicate radiative cooling at the SFC and the TOA, but smaller in magnitude (see Table 2). As the downward SW and LW fluxes at the surface from CERES are more consistent with observations, the CERES values are considered to be more accurate.

### 3.4 Radiative closure assessment based on PS106 observations

In this subsection, CERES fluxes and clear-sky T-CARS simulations are compared to the ship-borne observations of the down-
welling broadband SW and LW fluxes for the PS106 expedition. This comparison enables an assessment of RC for the entire expedition and to identify conditions with significant discrepancies. In subsections 3.4.1 and 3.4.2, clear-sky and all-sky conditions are considered separately.

### 3.4.1 Clear-sky radiative fluxes

For the clear-sky comparison, simulated and observed fluxes have been analysed based on the atmospheric classification de-
scribed in Sect. 3.1 (see Fig. A1). Furthermore, to improve the data quality, all-sky camera images were used to screen periods with larger differences. With this supplementary information, periods with broken cloud conditions and periods with external factors which could potentially compromise the radiation observations were excluded (e.g., the exhaust plume of *Polarstern*).

The comparison of SWD and LWD fluxes from T-CARS and CERES with ship-borne observations for clear-sky conditions is presented in Fig. 12 and summarized in Table B1. Figure 12a shows a histogram of the differences of the radiative fluxes from





T-CARS minus observations of LWD. This comparison indicates a skewed left distribution with a negative mean bias of the T-CARS simulations of -24.9 $\mathrm{W\,m^{-2}}$. After applying the improved quality screening described above, a mean flux difference of -14.2 $\mathrm{W\,m^{-2}}$ is found, together with a correlation coefficient of 0.92, and a more symmetric distribution than without this quality screening. The mean flux difference below the uncertainty limit of $\pm 16\,\mathrm{W\,m^{-2}}$, and the good correlation coefficient confirms that RC is achieved for the T-CARS simulations under clear-sky conditions.

In the case of CERES, the mean flux difference between simulations minus observations for the LWD flux is -4.6 $\mathrm{W\,m^{-2}}$, with a standard deviation of 20.9 $\mathrm{W\,m^{-2}}$, and a RMSE of 18.5 $\mathrm{W\,m^{-2}}$. While the bias suggests that RC is found for CERES, the rather low correlation coefficient of 0.476 indicates that the values do not reproduce variability as well as the T-CARS simulations and might be affected by the presence of clouds within the CERES footprint (see Fig. 12c). To test this hypothesis, clear-sky and pristine CERES products were also considered. For both data sets, the correlation coefficient reached values

above 0.765, which confirms that the all-sky CERES correlation coefficient is reduced by the presence of clouds. With this change, the bias however increased to -19.8 $\mathrm{W\,m^{-2}}$ and -18.2 $\mathrm{W\,m^{-2}}$ for the clear-sky and pristine data sets, respectively (see Fig. 12d).

The negative bias found for both T-CARS and CERES fluxes might also be caused by a positive bias of the ship-borne pyrgeometer observations, e.g., due to the influence of the exhaust plume of *Polarstern*. As there was only one pyrgeometer

measurement aboard *Polarstern*, it is impossible to further investigate this hypothesis. However, for future campaigns, it is recommended here to operate two pyrgeometers installed in different locations of the research vessel to exclude such influences.

The comparison for the SWD flux uses a stricter screening of data, which also excludes all periods when the pyranometer's field of view was obstructed by the superstructure of *Polarstern*. For T-CARS, a positive bias of 44.2 $\mathrm{W\,m^{-2}}$ and a correlation coefficient of 0.85 was found initially without this screening. With screening, a bias of 9.5 $\mathrm{W\,m^{-2}}$ and a correlation coefficient

of 0.95 were obtained. In the case of CERES, the biases for all-sky, clear-sky, and pristine conditions were found to have values of -27.1 $\mathrm{W\,m^{-2}}$, 3.6 $\mathrm{W\,m^{-2}}$, and 12.0 $\mathrm{W\,m^{-2}}$, respectively. These values confirm that the larger negative bias for all-sky conditions is due to the presence of clouds within the CERES footprint. The biases of both T-CARS and CERES are both within the uncertainty limit of $\pm 20\,\mathrm{W\,m^{-2}}$ indicating that radiative closure is achieved for both data sets.

Previous studies reported a similar magnitude of differences between simulated and observed downward fluxes for clear-

sky conditions. For instance, the analysis by Ebell et al. (2020) focused on Ny-Ålesund and reported a mean (median) flux difference of -5.0 (-5.5) $\mathrm{W\,m^{-2}}$ and 12.6 (-2.6) $\mathrm{W\,m^{-2}}$ for the LWD and SWD, accordingly (see their Fig. 3). The studies of Shupe et al. (2015) and Miller et al. (2015) found a median difference of simulations and observations for the LWD (SWD) flux of -6.9 (5.4) $\mathrm{W\,m^{-2}}$ and -5.5 (15.6) $\mathrm{W\,m^{-2}}$ for the Barrow and Summit-Greenland sites, respectively.

The treatment of aerosol in the CERES products is based on the aerosol optical depth (AOD) obtained from the Model of

Atmospheric Transport and Chemistry (MATCH; Collins et al. (2001)). The comparison between the CERES clear-sky and pristine fluxes yields a mean difference of 0.4 $\mathrm{W\,m^{-2}}$ and -7.5 $\mathrm{W\,m^{-2}}$ for the LWD and SWD, respectively. Considering the entire PS106 cruise, mean flux differences of 0.5 $\mathrm{W\,m^{-2}}$ for the LWD and -10.8 $\mathrm{W\,m^{-2}}$ for the SWD are found (see Fig B1a, b, e, and f).



While the LW value indicates that LW aerosol effects are negligible for most purposes, the SW values are relatively large in
comparison to the direct aerosol effect of -0.44 to -2.6 $\mathrm{W\,m^{-2}}$ reported by Rastak et al. (2014).

### 3.4.2 All-sky radiative fluxes

In this subsection, the CERES products are compared to the ship-borne observations for all-sky conditions. The T-CARS
simulations analysed as a time series are not considered in the discussion due to the instrumental limitations that occurred
during precipitation, LLS conditions or uncorrected attenuation instances, making it impossible to conduct a RC assessment
including cloudy-sky conditions. The comparison between observations and CERES fluxes is shown in Fig. 13 and Fig. 14.
Figure 13a shows the time series of the LWD observations and CERES all-sky simulations, indicating mostly good agreement.
Some periods can be identified with a reduced agreement. These cases occur particularly during precipitation periods, which
might affect the pyrgeometer measurements (e.g., 12 June, 20 June, 28 to 29 June, and 11 to 14 July). Larger discrepancies are
also observed in the presence of multilayer clouds (e.g., 20 June, 5 to 9 July), which feature a more challenging structure and
605 pose challenges for passive remote sensing (Minnis et al., 2019; Yost et al., 2020).

The discrepancy shown for 13 June 2017 might also be attributed to precipitation as it was also the case on 12 June. However,
this cannot be confirmed by the observations due to missing data. This day was also brought to attention by Barrientos Velasco
et al. (2020) since the near-surface temperature measured on the ice-floe by several instruments reached a mean temperature
of 281.1 K, about 4 K warmer than the temperature measured aboard *Polarstern* (Barrientos Velasco et al., 2020), CERES and
610 ERA5 skin temperature. This day was also characterized by a more humid than usual upper atmosphere, a cyclonic weather
system, northerly winds (Barrientos Velasco et al., 2020), strong temperature inversions, and an intense persistent fog leading
to a very low horizontal visibility (see Fig. 18b in Griesche et al. (2020)). This humidity intrusion can lead to an additional
energy flux to the surface, enhancing the fog to persist (Tjernström et al., 2019). The fluctuations of atmospheric temperature
and relative humidity described might be different from the atmospheric profiles used by CERES, causing the difference of up
to 20 $\mathrm{W\,m^{-2}}$ (Fig. 13a).

The mean flux difference of the LWD fluxes between CERES and observations for all-sky conditions is -6.0 $\mathrm{W\,m^{-2}}$, a
value that is within the instrumental uncertainty of $\pm13$ $\mathrm{W\,m^{-2}}$ of the pyrgeometer (see Fig. 13c). Hence, the RC for the
LWD flux can be confirmed for CERES. Figure 14a shows a scatter plot comparing CERES surface fluxes with the ship-borne
observations. The linear regression is calculated and illustrated in the same plot to determine the correlation of both data sets,
which indicates a correlation coefficient ($R^2$) of 0.69.

Similarly to LWD, there is relatively good agreement between the CERES simulations and observations for the entire PS106
(see Fig. 13b). Most of the discrepancies are caused by precipitation (e.g., 13 to 14 July), broken cloud conditions, and instances
when *Polarstern*'s superstructures compromised the pyranometer observations (see A1). Figure 13d shows the distribution of
the time series of the SWD flux. This panel shows a similar distribution between the observations and the SWD flux simulated
by CERES. This comparison indicates that CERES SWD flux is positively biased by 23.1 $\mathrm{W\,m^{-2}}$, with a standard deviation
of 59.3 $\mathrm{W\,m^{-2}}$. This value is considered acceptable since the instrumental uncertainty is $\pm20$ $\mathrm{W\,m^{-2}}$, and the moments with
broken cloud conditions or obstruction of the observations are not excluded from the comparison. Nevertheless, it is important





to stress that obtaining good quality ship-borne observations of downward fluxes is challenging due to recurrent obstructions of the view of the radiometer's sensor.

The study of Riihelä et al. (2017) presents a comparison of radiative fluxes between the CERES edition 3 products and in-situ observations from the drifting Tara ice camp from April to September 2007. Their results are presented as daily means and indicate a RMSE of 24.5 W m$^{-2}$ and 17.1 W m$^{-2}$ for the SWD and LWD, respectively. In our study, results are relatively similar for the LWD, with a RMSE of 12.3 W m$^{-2}$. In the case of the SWD, the RMSE found for PS106 has a value of 46.5 W m$^{-2}$. It should be noted that our results are based on hourly means instead of the daily mean reported by Riihelä

et al. (2017). Furthermore, the observations made during the Tara ice floe camp were unaffected by interference of the ship superstructure with the observations.

The study by Dong et al. (2016) focuses on the radiative closure of SWD flux for single-layer overcast liquid-phase Arctic stratus clouds over the snow-free and snow-covered surfaces. Their analysis considers CERES edition 2 and 4 products, which are subsequently compared to radiation observations and cloud retrievals at the Atmospheric Radiation Measurement North

Slope of Alaska (ARM NSA) site at Utqiaġvik, Alaska. CERES edition 4 shows a mean bias against observations below 10 W m$^{-2}$ for the SWD and for snow-free as well as snow conditions. Evidently, our flux difference of 23.1 W m$^{-2}$ is larger than the value reported by Dong et al. (2016); however, the present study includes more complex cloud situations, which enlarges the differences of radiative fluxes.

**3.5 Cloud radiative effect and radiation budget during PS106**

This section presents an analysis of the radiation budget and CRE for the summer-time Arctic and for the period of the PS106 expedition based on CERES products. While it was initially planned to also include the T-CARS simulations in this analysis, their limited temporal coverage precludes meaningful results. In particular, the exclusion of situations with the prevailing low-level stratus clouds is expected to bias mean fluxes. CERES data for the period of the first (from 16:45Z on 28 May until midnight on 20 June 2017) and the second leg of PS106 (22 June to 16 July 2017) are used for consistency with the temporal

analyses given in the previous sections. The cruise track during this period lay entirely within the Arctic, defined here to cover the range from 70°-90°N, and consistent with the definition used by several previous studies (e.g., Walsh et al. (2009); Huang et al. (2017)). In addition to the effects of clouds on the radiation budget, the relevance of the direct aerosol radiative effect for the radiation budget is briefly discussed based on CERES data sets.

An overview of the most important components of the radiation budget is given in Fig. 15, showing mean values along the

PS106 cruise track and for the entire Arctic as separate panels. A full list of the different flux components can be found in the Appendix in Table C1. Along the PS106 cruise track and under all-sky conditions, the radiation budget at the SFC indicates a strong warming influence by the SW net flux with 110.9 W m$^{-2}$, while the LW fluxes cool the surface by -22.1 W m$^{-2}$. The presence of clouds enhances the LWD flux at the SFC by about 62.1 W m$^{-2}$, while reducing the SWD and SWU by 121.0 W m$^{-2}$ and 51.5 W m$^{-2}$, respectively. For the period of interest, the net radiation budget at the SFC has a value of

88.8 W m$^{-2}$ for PS106, and a value of 94.4 W m$^{-2}$ for the entire Arctic. The difference in the net radiation budget for the PS106 track and the entire Arctic is relatively small with a value of 5.6 W m$^{-2}$, and can be attributed to differences in the mean



incoming SW radiation at the TOA, downwelling LW radiation, the transmission of SW radiation through the atmosphere, and the surface albedo.

At the TOA and for the PS106 cruise, the mean net radiation budget is $0.2\,\mathrm{W\,m^{-2}}$, while for clear-sky conditions, the
net radiation budget would be $48.6\,\mathrm{W\,m^{-2}}$. At the TOA, the presence of clouds increases the reflected solar radiation by $62.9\,\mathrm{W\,m^{-2}}$ and reduces outgoing LW flux by $14.5\,\mathrm{W\,m^{-2}}$. For the Arctic, the net radiation budget under all-sky conditions has a value of $7.8\,\mathrm{W\,m^{-2}}$, versus a value of $47.9\,\mathrm{W\,m^{-2}}$ for clear-sky conditions. This indicates that the cooling by the cloud radiative effect is smaller for the entire Arctic than for PS106. This difference can mainly be attributed to an increased reflection of solar radiation due to a larger cloud fraction and higher surface albedo along the PS106 track compared to the Arctic.

To consider the radiation budget of the atmosphere, the flux divergence has been calculated as the difference of the mean values at the TOA and the SFC. The net values found for the entire Arctic ($-86.6\,\mathrm{W\,m^{-2}}$) and PS106 ($-88.6\,\mathrm{W\,m^{-2}}$) show a significant cooling, and are once more relatively similar in magnitude. Since the net radiation budget for PS106 and the Arctic are relatively similar at the SFC, TOA and within the atmosphere, the PS106 expedition can be considered representative for the entire Arctic (see Table C1).

Additionally, the direct radiative effect of aerosols on the radiative fluxes and the radiation budget has been examined for clear-sky and all-sky conditions. By considering the different CERES flux products, specifically the all-sky (AS), cloudy without aerosols (NA), clear-sky(CS), and pristine (P) fluxes an estimate of the direct radiative effect of aerosols can be obtained. A summarized version of Table C1 is given in Table C2 in the Appendix, showing the mean perturbation in radiative fluxes arising from aerosols. To be expected, aerosol effects are mainly limited to the SW radiation and are the largest under
clear-sky conditions. A decrease of SWD (SWU) of $10.8\,\mathrm{W\,m^{-2}}$ ($6.1\,\mathrm{W\,m^{-2}}$) is found at the surface for the PS106 track. Slightly larger values are calculated for the entire Arctic, with a reduction of the SWD (SWU) by $13.9\,\mathrm{W\,m^{-2}}$ ($7.4\,\mathrm{W\,m^{-2}}$. Aerosols have a small warming effect in the LW, leading to an increase no larger than $0.7\,\mathrm{W\,m^{-2}}$ and $0.1\,\mathrm{W\,m^{-2}}$ for the LWD and LWU both for the PS106 track and the entire Arctic, respectively. At the TOA, the effect of aerosols does not surpass more than $0.2\,\mathrm{W\,m^{-2}}$ for the upwelling SW and LW fluxes.

Considering net fluxes for clear-sky and cloudy conditions, the direct aerosol radiative effect at the surface along the PS106 track and the entire Arctic are $-4.2\,\mathrm{W\,m^{-2}}$ and $-5.9\,\mathrm{W\,m^{-2}}$, respectively. At the TOA, the radiative effect of aerosols causes a minor cooling by $-0.1\,\mathrm{W\,m^{-2}}$ for both the PS106 track and the Arctic for clear-sky conditions, which changes to a small warming effect of $2.2\,\mathrm{W\,m^{-2}}$ and $1.9\,\mathrm{W\,m^{-2}}$ for PS106 and the Arctic, respectively in the presence of clouds. These values are consistent with the study of Markowicz et al. (2021), who determined the aerosol radiative effect based on radiative transfer
simulations and the long-term aerosol reanalysis provided by the Navy Aerosol Analysis and Prediction System from 2003 to 2015. Focusing on the Arctic, they report an annual mean net direct (indirect) aerosol radiative effect of $-3.01$ ($-1.88$) $\mathrm{W\,m^{-2}}$ and $-0.73$ ($0.31$) $\mathrm{W\,m^{-2}}$, at the surface and TOA, respectively (see their Table 5). While the values calculated in our study are relatively larger than their annual-mean values, our values are consistent with the early summer values reported for the months from May to July (see their Fig. 12a and Fig. 13a).

It is worth pointing out that the values of aerosol radiative effects reported here critically depend on an accurate representation of aerosol properties in the radiative transfer calculations underlying the CERES products. As aerosol properties are represented




based on the assimilation of MODIS products into the MATCH aerosol transport model, its accuracy in the Arctic determines the accuracy of our findings, and the use of reanalysis properties can have significant biases (e.g., Witthuhn et al. (2021)).

Given that the main interest of this study is the radiative effect of clouds in the Arctic, the CRE along the cruise track of the
700 PS106 expedition is shown as a time series with hourly resolution in Fig. 16 for the SW, LW, and net components at the SFC and the TOA. The LW CRE at the SFC and TOA has mean values of 60.6 and 14.4 $\mathrm{W\,m^{-2}}$, respectively. These values imply a significant warming of the surface and cooling of the atmosphere caused by clouds. Moments when the LW CRE at the surface surpassed 80 $\mathrm{W\,m^{-2}}$ occurred mostly for persistent mixed-phase single-layer low-level clouds (e.g., 3 to 6 June, 16 to 17 June, 25 June, and 1 July 2017; see Fig. 5a and 5c). The persistent cloud structure observed from 5 June to 7 June 2017 is studied
in more detail in Egerer et al. (2021), who suggest that humidity inversions supply moisture to the cloud layer, increasing the persistence of the cloud. Time periods with the largest LW CRE and cloud conditions described occurred mostly over sea ice or the marginal zone.

The SW CRE shows a clear dependency on the surface albedo and SZA, as shown by the oscillations visible at the SFC and the TOA in Fig. 16b. The colour band shown at the bottom of Fig. 16 indicates whether *Polarstern* was located in the open sea,
the sea ice-marginal zone, or within the sea ice. The surface conditions and the SZA strongly modulate the magnitude of the SW CRE. Thus, the highest values of SW CRE are found for open ocean rather than for sea ice due to the high surface albedo of ice, which increases the amount of reflected SW radiation and reduces the SW CRE.

The magnitude of CRE depends on several additional factors, including cloud macro- and microphysics, surface conditions, and SZA (Shupe et al., 2005; Sedlar et al., 2011; Ebell et al., 2020; Stapf et al., 2020). To investigate these dependencies, the
715 values of $\mathrm{CRE}_{SFC}$ are plotted versus these parameters in a scatter plot in Fig. 17 using colour-coding on the points based on the SZA. The figure clearly illustrates that neither of the considered parameters alone can explain the magnitude of the CRE. Instead, it is modulated by the interplay of several parameters. Nevertheless, it can be recognized that the SZA determines whether clouds cool (low SZA) or warm the surface (high SZA) and that an increase in CF also causes an increase of the CRE.

In panel 17b, the CF is compared with the $\mathrm{CRE}_{SFC}$ to investigate the dependence of the net CRE on this parameter. The
720 dependence on surface albedo shown in Fig. 17b indicates that larger surface cooling by clouds occurs over open ocean than over ice. Displaying cases with a surface albedo lower (higher) than 0.4 in separate panels (Fig. 17c and Fig. 17d, respectively). The role of $Q_L$ as reported by CERES is investigated for different surface types. Over open ocean, the $\mathrm{CRE}_{SFC}$ depends more strongly on SZA than on $Q_L$. Over highly reflective surfaces, in contrast, larger values of $Q_L$ more often cause a warming than a cooling effect (see Fig. 17d) at the SFC.

The $\mathrm{CRE}_{SFC}$ has also been compared to the surface albedo to determine how this parameter influences the CRE. Figure 17b shows a strong interaction among the $\mathrm{CRE}_{SFC}$, the surface albedo and the SZA. While the correlation between SZA and the $\mathrm{CRE}_{SFC}$ is evident, the surface albedo values also indicate that the largest cooling $\mathrm{CRE}_{SFC}$ occur over the open ocean than over ice and that most of the warming of the $\mathrm{CRE}_{SFC}$ occurred over highly reflective surfaces. Furthermore, Fig. 17b has been subdivided in two by separating the data points with a surface albedo higher or equal than 0.4 (Fig. 17d), and lower than 0.4
(Fig. 17c). This subdivision aims to investigate the role of $Q_L$ derived by CERES, which largely contribute to the CRE (Shupe and Intrieri (2004); Ebell et al. (2020)) on both types of surface albedo. This analysis indicates that over the open ocean, the





$CRE_{SFC}$ is more dependent on the SZA than the magnitude of $Q_L$. Over high reflective surfaces, the dependence of SZA over the $CRE_{SFC}$ is also evident; however, in this case, larger values of $Q_L$ caused warming at the surface more frequently than a cooling effect (Fig. 17d) at the surface.

For the cruise track of the PS106 expedition, SW cooling by clouds dominates over LW warming, leading to a net cooling of -8.8 $\mathrm{W\,m^{-2}}$ at the surface (see Table 3). For a similar time period, the net CRE at the surface was also investigated by two other studies, both reporting stronger cooling. For Ny-Ålesund, NO, Ebell et al. (2020) find values within a range of about -20 $\mathrm{W\,m^{-2}}$ to -40 $\mathrm{W\,m^{-2}}$ (see Fig. 6c in Ebell et al. (2020)). Considering the SW CRE, the study of Stapf et al. (2020) focused on the ACLOUD airborne campaign, calculated a mean SW CRE at the surface of -32 $\mathrm{W\,m^{-2}}$; however, by considering a proposed

cloud-free retrieval of surface albedo taking into account spectral effects, this value increases to -62 $\mathrm{W\,m^{-2}}$, which is similar to our result of -68.8 $\mathrm{W\,m^{-2}}$ (see Table 3). It is worth noting that Ebell et al. (2020) take into account explicitly the effects of white-sky and black-sky albedo, and values obtained by CERES represent a cloud-free albedo (Chen et al., 2006). Due to the strong sensitivity of the SW CRE to surface albedo, a critical assessment of the accuracy of surface albedo used by CERES, including spectral and directionality effects, e.g., with in situ observations, is recommended.

A similar analysis has been applied to the entire Arctic to embed the results obtained for PS106 in a wider context. Panels (f) and (i) of Fig. 18 show the mean CRE at the surface and TOA for the entire time period of PS106, respectively. Additionally, the mean surface albedo, the TOA albedo, and the cloud fraction are shown in panels (a), (b), and (c), respectively.

For the considered time period and the Arctic region, a larger cloud fraction is found over open ocean than over land. This might be attributable to an increase of lower-tropospheric stability over land, which inhibits the mixing between the free

troposphere and cloud layer, thus reducing cloud fraction (Morrison et al., 2012). The latter effect is most evident over the Barents Sea, central Arctic, Baffin Bay, and Siberian Sea (Fig. 18c). Over sea-ice, the cloud fraction is also large, with a particularly high occurrence of low and mid-low level clouds (not shown).

Based on CERES, low-level clouds are frequently found over the entire Arctic Ocean, with enhanced occurrence frequencies over the Barents Sea, Kara Sea, Laptev Sea, and the central Arctic. Mid-level clouds are mainly present over Greenland, East

Arctic-Russia, Ellesmere Island and the central Arctic Ocean. High-level clouds occur more frequently over land, specifically over the Arctic-Russia, near Iqaluit in Canada, North of Sweden, Finland, and Norway (not shown). The spatial distribution of cloud fraction as visible in Fig. 18c is consistent with the results of Palm et al. (2010) who report about an anticorrelation between sea ice extent and cloud fraction based on satellite-based lidar measurements from the CALIOP mission for a 5-year period.

Differences in the spatial distribution of cloud fraction yield differences in CRE values both at the SFC and the TOA. As indicated by Fig. 18f, the surface CRE is highly dependent on surface albedo, suggesting a warming effect of clouds by up to 20.5 $\mathrm{W\,m^{-2}}$ over highly reflective regions covered by snow/sea ice (Fig. 18a). At the TOA, the CRE shows a similar dependence on surface albedo but with a stronger cooling effect. The mean CRE for the considered period is -9.3 $\mathrm{W\,m^{-2}}$ at the surface, while it is -40.1 $\mathrm{W\,m^{-2}}$ at the TOA (see Table 3). The TOA albedo depends on cloud fraction and surface albedo. Low

values are only expected for clear-sky conditions over ocean, while high values correspond to either snow-covered or opaque clouds. This effect is, in particular, visible for the Barents Sea area, which corresponds to open ocean with a low surface albedo





(Fig. 18a); thus, the relatively high values found in Fig. 18b are caused by the frequent occurrence of clouds, as indicated in Fig. 18c. This region is of particular interest because mean values of CRE at the TOA and SFC are larger than for the rest of the Arctic.

Previous studies have also investigated the CRE at the surface in the Arctic; see Table 4 for a list. The SHEBA expedition carried out observations in the Beaufort and Chuchki Sea, and its observations were used to estimate the $CRE_{SFC}$ for an entire annual cycle. Based on SHEBA, Intrieri et al. (2002) concluded that clouds warm the surface for most of the year. During early July, however, a cooling effect of the surface by about -4 $W\,m^{-2}$ was found. Based on the Arctic Summer Cloud Ocean Study (ASCOS) ship-borne campaign, the $CRE_{SFC}$ was also calculated, finding a warming effect, with values ranging from

5 to about 75 $W\,m^{-2}$ (see Fig. 8 in Sedlar et al. (2011)). Long-term observations can be used to resolve details of the annual cycle. Based on 10-years of observations at the North Slope Alaska (NSA) and NOAA Barrow Observatory (BRW) sites, the analysis of Dong et al. (2010) found a surface CRE ranging from -15 to -35 $W\,m^{-2}$ for the time period from end of May to middle of July (see Fig. 3 in Dong et al. (2010)). In contrast, the mixed-phase Arctic Clouds Experiment (M-PACE) conducted at the NSA site found a $CRE_{SFC}$ ranging from 40 to 80 $W\,m^{-2}$ for the months from September to December (de Boer et al.,

2011), which is significantly larger than values reported by Dong et al. (2010).

    In contrast to the remainder of the Arctic, Greenland is covered by ice and snow during the entire year. Clouds have a warming effect during the entire annual cycle, with the highest values of CRE found from July to August (see Fig. 5 in Miller et al. (2015)). Miller et al. (2015) report values within a range of about 25 to 45 $W\,m^{-2}$, which is larger than the mean values reported here, reaching a maximum of 20.5 $W\,m^{-2}$ over Greenland (see Fig. 18).

Kay and L'Ecuyer (2013) derived the CRE based on a combination of active and passive satellite-based remote sensing observations over the Arctic Ocean and report a mean $CRE_{SFC}$ of about -30 to -40 $W\,m^{-2}$ at the surface for the period from the end of May to mid of July. These values are evidently larger than the mean value of -9.3 $W\,m^{-2}$ found in the present study. However, the results of Kay and L'Ecuyer (2013) focus mostly on the Arctic Ocean and exclude Greenland.

    The comparison and discussion of our results with previous studies provide valuable context on the CRE across the Arctic

for different seasons and locations. Nevertheless, it is worth noting that such a comparison cannot fully account for all factors affecting the results, including the consideration of particular sites, regions, cloud conditions, seasons, data sets, methods to obtain surface albedo, and different temporal averaging.

## 4   Conclusions and Outlook

This study has investigated the characteristics of Arctic clouds, and their effect on radiative fluxes and the radiation budget

during the *Polarstern* cruise PS106 held in early summer of 2017. An intercomparison of cloud properties derived from the ship-borne cloud remote sensing observations using the CERES SYN1deg Ed. 4.1 satellite products has been conducted first, followed by an intercomparison of radiative fluxes from ship-borne observations, radiative transfer simulations, and CERES. For this purpose, a radiative transfer setup for the simulation of SW and LW fluxes denoted as T-CARS has been implemented



using atmospheric profiles from the ERA5 reanalysis, Cloudnet cloud products, and other ancillary data as input. This setup
has been also been used to quantify the sensitivity of clear-sky radiative fluxes to various input parameters.

Considering the different perspectives and sensitivities of the satellite and ship-borne remote sensing observations, the horizontally resolved field of view of CERES and the vertically resolved view of the active remote sensing instruments aboard
*Polarstern* are found to offer complementary information on Arctic clouds.

A list of the main conclusions of this paper is given here:

1. Based on the CERES (Cloudnet) products, clouds occurred for about 86.7 % (76.1 %) of the time during PS106. Differences between the CERES and Cloudnet are mainly due to the different spatial resolution, data gaps, and moments
   of misidentifying clouds by Cloudnet. In the case studies, situations were identified where CERES underestimates high-
   level clouds, likely due to previously-reported limitations of CERES products in polar regions (Trepte et al., 2019;
   Sun-Mack et al., 2018).

2. A case study comparing of the CERES products (T-CARS simulations) with ship-based observations of downward fluxes
   in clear-sky conditions yields satisfactory agreement, with flux mean differences of -13 $\mathrm{W\,m^{-2}}$ (-13.6 $\mathrm{W\,m^{-2}}$) for the
   LWD, and -2.1 $\mathrm{W\,m^{-2}}$ (-7.9 $\mathrm{W\,m^{-2}}$) for the SWD. This finding holds despite the harsh environmental conditions and
   ship-borne operation, which likely increase the instrumental uncertainties (estimated to be $\pm 20$ $\mathrm{W\,m^{-2}}$ for pyranome-
   ter/SWD and $\pm 13$ $\mathrm{W\,m^{-2}}$ for pyrgeometers/LWD). While the T-CARS simulations currently neglect aerosol effects,
   the CERES products suggest that on average, aerosols increase the LWD flux by 0.5 $\mathrm{W\,m^{-2}}$, and decrease the SWD by
   10.8 $\mathrm{W\,m^{-2}}$ (Fig. B1), confirming that aerosol effects are minor for the LW and only relevant for the SW.

3. For all-sky conditions, CERES surface radiative fluxes and observations agree well during PS106, with a mean difference
   of 6.0 $\mathrm{W\,m^{-2}}$ and 23.1 $\mathrm{W\,m^{-2}}$ for the SWD and LWD, respectively. The comparison yields a correlation coefficient of
   0.69 (0.77) and RMSE of 12.2 $\mathrm{W\,m^{-2}}$ (46.5 $\mathrm{W\,m^{-2}}$) for the LWD (SWD). These results are consistent with the findings
   by Dong et al. (2016), and Riihelä et al. (2017), who also performed comparisons between in-situ radiative fluxes and
   CERES products at the ARM NSA site, as well as the Tara ice camp and on the Greenland Ice Sheet, respectively.
   Instrumental limitations arise from the ship-based operation of instruments and the harsh environmental conditions,
   which cause a reduced accuracy of the observed radiative fluxes and limits their accuracy for radiative closure studies.

4. Based on the CERES products, the mean radiation budget has been estimated along the cruise track and for the entire
   Arctic for the period of PS106. A dominating contribution of the SW radiation to the surface radiation budget is found,
   leading to a net surface flux of 88.8 $\mathrm{W\,m^{-2}}$, and 94.4 $\mathrm{W\,m^{-2}}$ for PS106 and the entire Arctic, respectively. Moreover,
   the effect of clouds on the radiation budget has been investigated. The mean net CRE during PS106 along the cruise
   track is -8.8 $\mathrm{W\,m^{-2}}$ at the SFC and -48.4 $\mathrm{W\,m^{-2}}$ at the TOA, implying an atmospheric cooling of 39.6 $\mathrm{W\,m^{-2}}$ (Table
   3). For the entire Arctic, the net CRE is similar to PS106, with values of -9.3 $\mathrm{W\,m^{-2}}$, and -40.1 $\mathrm{W\,m^{-2}}$, at the SFC and
   TOA, respectively (Fig. 18). The similarity of the local and regional CRE suggests that the PS106 cloud observations
   along the cruise track are representative of the cloud conditions found over the Arctic during this time of the year. Our





results are also consistent with the summer results obtained from Ebell et al. (2020), who calculated the CRE during 2017 at Ny-Ålesund - NO (see their Fig. 6).

In the future, we plan to carry out a similar analysis based on observations from the Multidisciplinary drifting Observatory

for the Study of Arctic Climate (MOSAiC) expedition (https://mosaic-expedition.org/, last access: 06 March 2020). This will extend the temporal coverage of observations to a full annual cycle in the central Arctic, and thus increase the climatological relevance of findings.

The radiative closure assessment using ship-borne remote sensing observations as input for the T-CARS simulations has been limited to specific cases in the present study. An extension to the full period of the cruise would allow a more in-depth

investigation of the accuracy of the Cloudnet cloud products for specific cloud conditions. Particular attention should be given to the accuracy of the Cloudnet retrievals for relatively low values of $Q_L$ and $Q_I$. For MOSAiC, cloud products based on the Shupe-Turner retrievals (Shupe et al., 2015) and distributed by ARM will also be available, allowing a comparison of the Cloudnet and ARM products with respect to their ability to accurately represent the optical properties of Arctic clouds. Particular attention will be given to periods when the LW radiative properties are sensitive to small changes in $Q_L$ (e.g., <

50 $\mathrm{gm}^{-2}$; Turner (2007); Tjernström et al. (2015); Achtert et al. (2020), which have been found to be poorly captured in our study.

Satellite observations and products are necessary to extend the analysis of cloud radiative effects to a wider regional and decadal perspective. This would allow to investigate long-term changes of the cloud radiative effect and the radiation budget. To contribute to the latter, we acknowledge the particular importance of the LW radiation budget across the entire annual cycle

in the Arctic region (Sedlar and Tjernström, 2017). Therefore, we will expand our analysis by implementing radiative kernel techniques to diagnose climate feedback based on ground-based and satellite remote sensing observations (Soden et al., 2008; Tan and Storelvmo, 2019).

*Data availability.*

The analysed Cloudnet data is available at Griesche et al. (2020a), Griesche et al. (2020b), Griesche et al. (2020c), Griesche

et al. (2020d), and Griesche et al. (2020e). The analysis of low-level stratus clouds introduced in Griesche et al. (2020) is based on the data set available at Griesche et al. (2020f).

The data used for surface parameters based on single layer hourly ERA5 data is available at Hersbach et al. (2018b) and for pressure levels is available at Hersbach et al. (2018a).

The CERES products were obtained from the NASA Langley Research Center Atmospheric Science Data Center, which is

available at NASA/LARC/SD/ASDC (2017).

The T-CARS simulations are available at Barrientos Velasco et al. (2021).



*Author contributions.* CBV, HD and AH conceptualized the manuscript. PS and HGJ were responsible for the data curation of the observations. CBV performed the formal analysis, methodology, visualization and writing of the original draft preparation. AM was responsible for the funding acquisition and the project administration. All authors contributed to the subsequent improvement of the analysis and editing of the manuscript.

*Competing interests.* The authors declare that they have no conflict of interest.

*Acknowledgements.* We gratefully acknowledge the funding by the Deutsche Forschungsgemeinschaft (DFG, German Research Foundation) – Project Number 268020496 – TRR 172, within the Transregional Collaborative Research Center "ArctiC Amplification: Climate Relevant Atmospheric and SurfaCe Processes, and Feedback Mechanisms (AC)[3]". We thank Captain Thomas Wunderlich and the entire crew of *Polarstern* for their logistic support. We also thank our colleagues at AWI, DWD, Leipziger Institut für Meteorologie and TROPOS for their logistic support and scientific cooperation. We thank the anonymous reviewers, for their suggestions and comments, which significantly improved the final version of this manuscript. We would like to acknowledge Copernicus Climate Change Service (C3S) for making their ERA5 products easily accessible. Similarly, we deeply acknowledge the NASA Langley Research Center Atmospheric Science Data Center for making available their products for the analysis of this study.





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





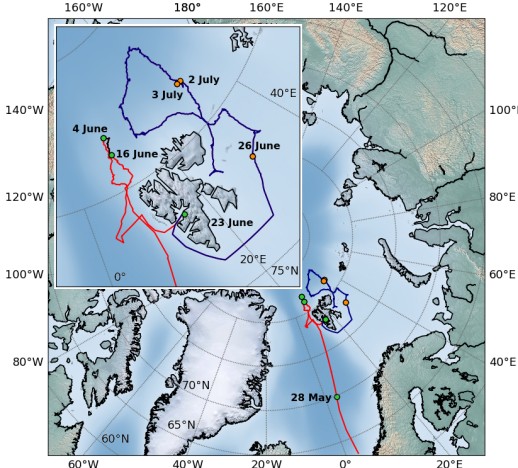

**Figure 1.** Cruise track of the PS106 *Polarstern* expedition is shown on a polar stereographic map of the Arctic. The red line shows the track for the first leg (PS106/1, also denoted as PASCAL), the black line indicates the position for the ice floe camp during PASCAL, and the blue line shows the cruise track for the second leg (PS106/2, also denoted as SiPCA). The upper left box zooms in the ice floe camp drift. Green dots indicate the start of PASCAL and SiPCA legs, and the beginning and end of the ice floe camp. Orange dots depict the location of several case studies.



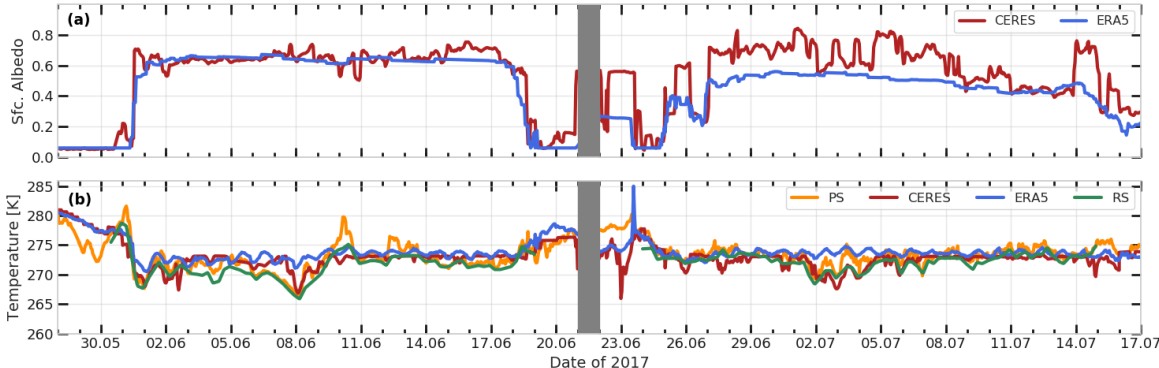

**Figure 2.** Time series of surface properties along the PS106 cruise track. Panel (a) shows the surface albedo based on CERES (red) and ERA5 (blue). Panel (b) shows the temperature based on *Polarstern* (PS) observations and radiosondes (RS) at 10 m above sea level in orange and green, respectively, and the skin temperatures from ERA5 and CERES in blue and red, respectively.





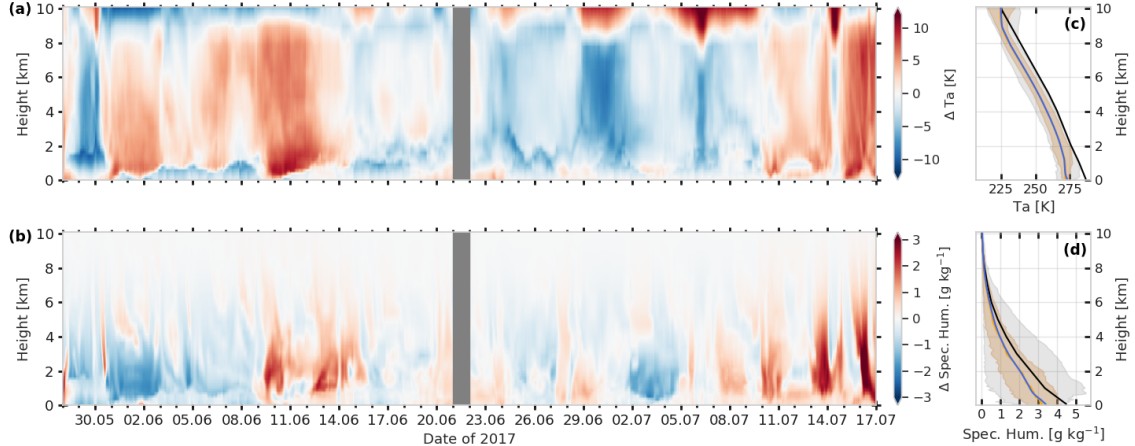

**Figure 3.** Time-height indicator plot of atmospheric profiles obtained from ERA5 along the PS106 cruise track. Panel (a) shows the atmospheric temperature anomaly ($\delta$Ta), panel (b) shows specific humidity anomalies. In (c) and (d), the mean profiles of atmospheric temperature and specific humidity, respectively, are shown. The blue and orange lines correspond to ERA5, radiosondes, and the Sub-Arctic Summer standard atmosphere (Anderson et al., 1986) is displayed in black. The grey-shaded area indicates the minimum and maximum while, while the brownish-shaded area shows the interquartile range of ERA5 vertical profiles.

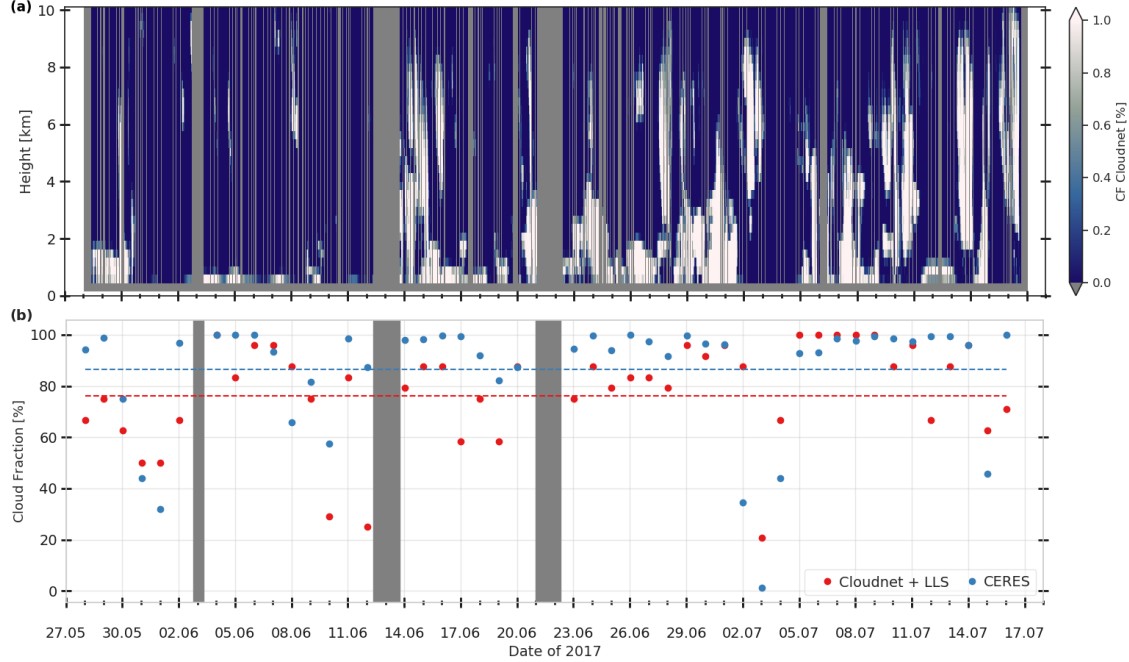

**Figure 4.** Cloud fraction (CF) observed along the PS106 cruise track. Panel (a) shows the vertically resolved mean Cloud Fraction (CF) based on Cloudnet observations at hourly resolution. Data gaps are indicated in grey. Panel (b) shows daily mean cloud fraction based on Cloudnet plus detection of low-level stratus clouds (LLS) introduced by Griesche et al. (2020) (red) and CERES (blue).

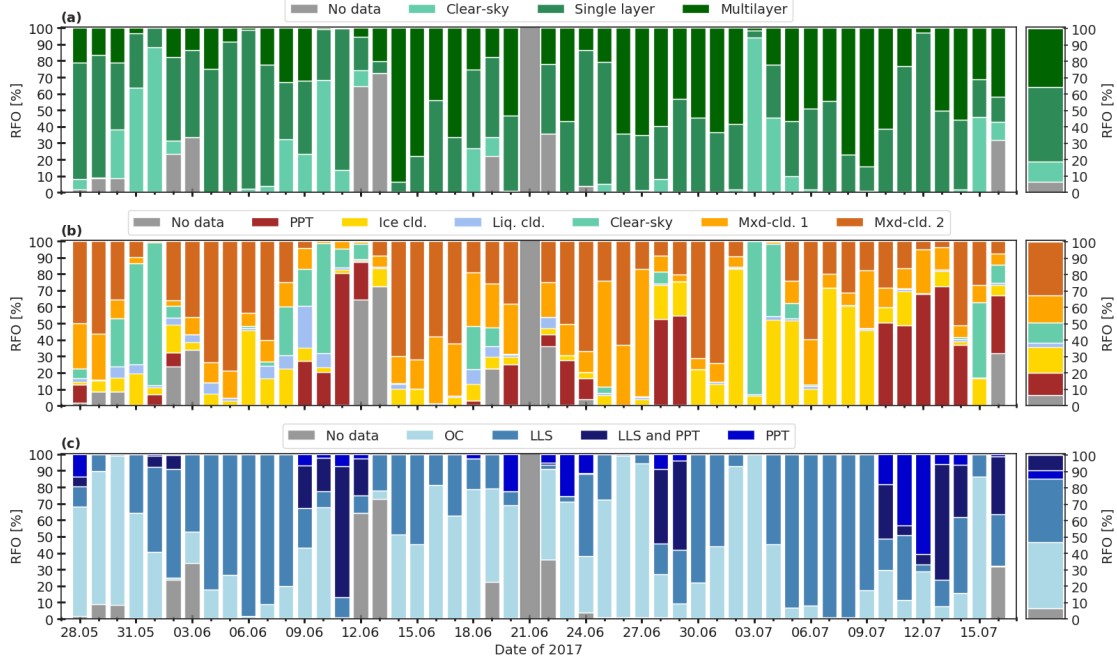

**Figure 5.** Daily and overall relative frequency of occurrence (RFO) of various cloud characteristics. Panel (a) shows the RFO of clear-sky, single-level clouds, and multilayer clouds. Panel (b) shows the RFO of the thermodynamic phase of single-layer clouds, differentiating periods of ice clouds, liquid clouds, mixed-phase clouds of type 1 or 2, precipitation (PPT), and clear-sky. Data gaps are marked in grey in all panels. Panel (c) is the RFO of various quality flags indicating optimum conditions (OC), low-level stratus (LLS), PPT and simultaneous occurrence of LLS and PPT.





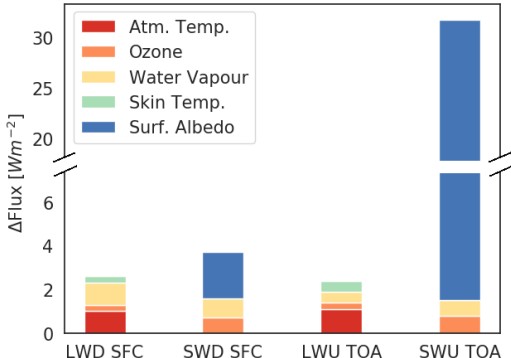

**Figure 6.** Results of the sensitivity analysis visualized as stacked bar chart of the absolute change in clear-sky fluxes in response to perturbations of various input parameters of the T-CARS simulation. See text for a description of the unperturbed inputs and the magnitude of the perturbations. Changes are shown for the downward LW (LWD) and SW (SWD) fluxes at the surface (SFC) and the upward LW (LWU) and SW (SWU) fluxes at the top of the atmosphere (TOA), in response to changes in atmospheric temperature, ozone and water vapour column, skin temperature and surface albedo.



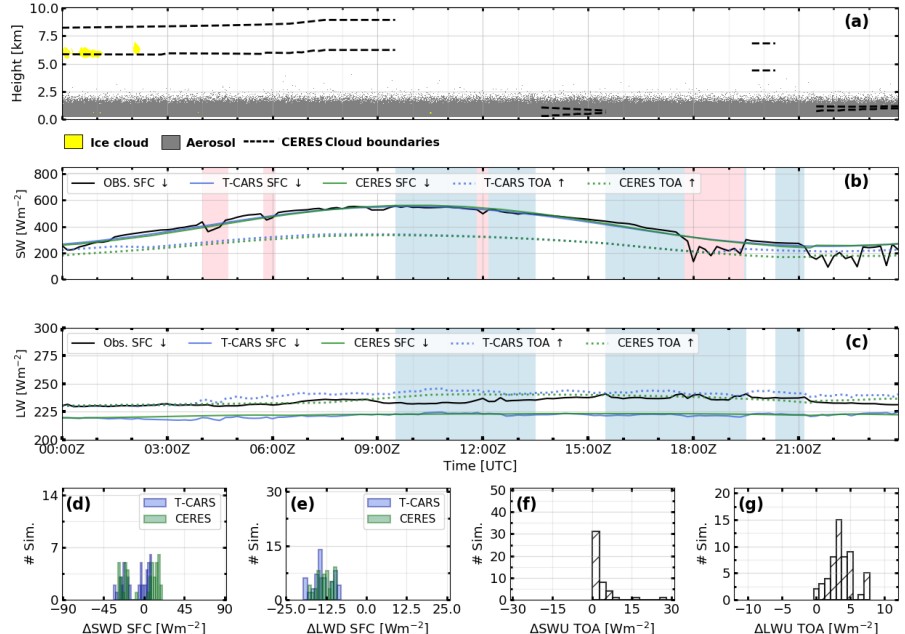

**Figure 7.** Overview of radiative fluxes for the 3 July 2017 case. Panel (a) shows the time series of the Cloudnet target classification and the CERES-based cloud boundaries (dashed black lines). Panel (b) and (c) show the time series of the SW fluxes and LW fluxes, respectively. In both panels (b and c), the down-looking arrow indicates (↓) the downward fluxes at the surface (SFC), and the up-looking arrow (↑) shows the upward fluxes at the top of the atmosphere (TOA). Panels (d) and (e) show histograms of the difference of T-CARS/CERES SWD and LDW fluxes and observations at the surface, respectively. Panels (f) and (g) show histograms of the difference of T-CARS minus CERES SWU and LWU fluxes at the TOA, respectively. The number of 10-minute data points contributing to the histogram in panel (d) is 44, while it is 55 for panels (e)-(g). Pink shading indicates periods when the ship's superstructures obstructed the ship-borne flux observations. Light-blue background indicates the period of clear-sky considered for the analysis.



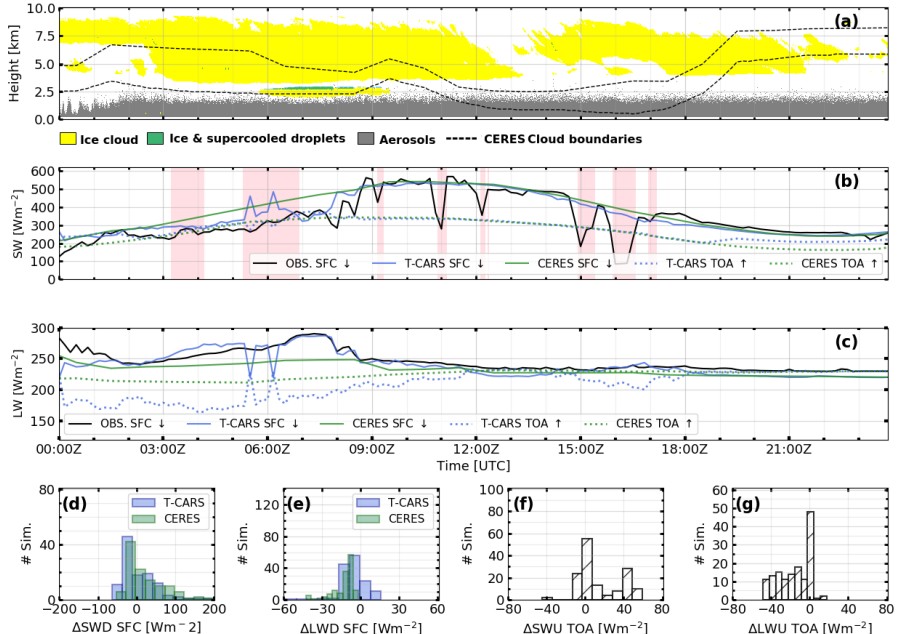

**Figure 8.** Same as Figure 7, but for the 2 July 2017 case.





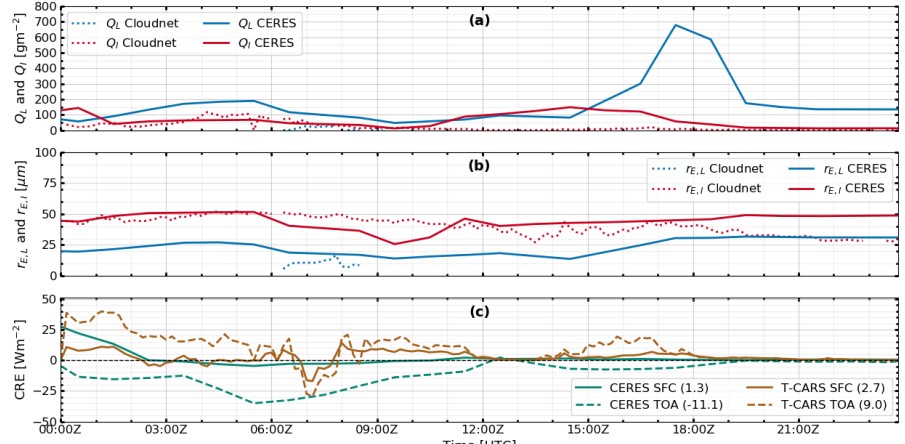

**Figure 9.** Time series of cloud microphysics properties and CRE between CERES and Cloudnet for 2 July 2017 case. Panel (a) shows the comparison between CERES and Cloudnet of liquid water path ($Q_L$; blue) and ice water path ($Q_I$; red). Panel (b) shows the liquid effective radii ($r_{E,L}$; blue) and the ice effective radii ($r_{E,I}$; red). Panel (c) shows the cloud radiative effect (CRE) at the surface (SFC; solid lines) and the top of the atmosphere (TOA; dashed lines) from T-CARS (brown) and CERES (green).





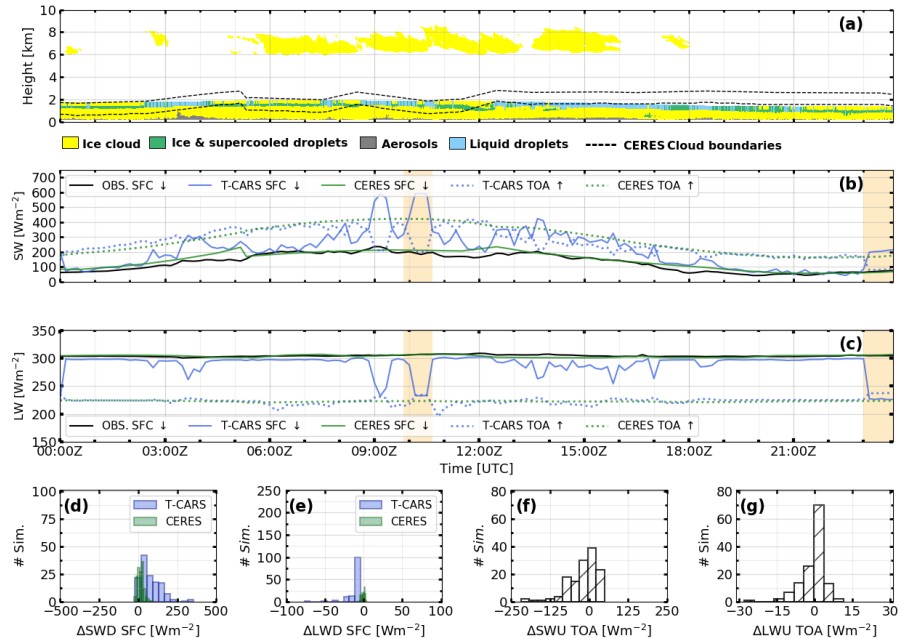

**Figure 10.** Same as Figure 7, but for the 26 June 2017 case. The pale yellow shading indicates periods with uncorrected attenuation by the Cloudnet products.





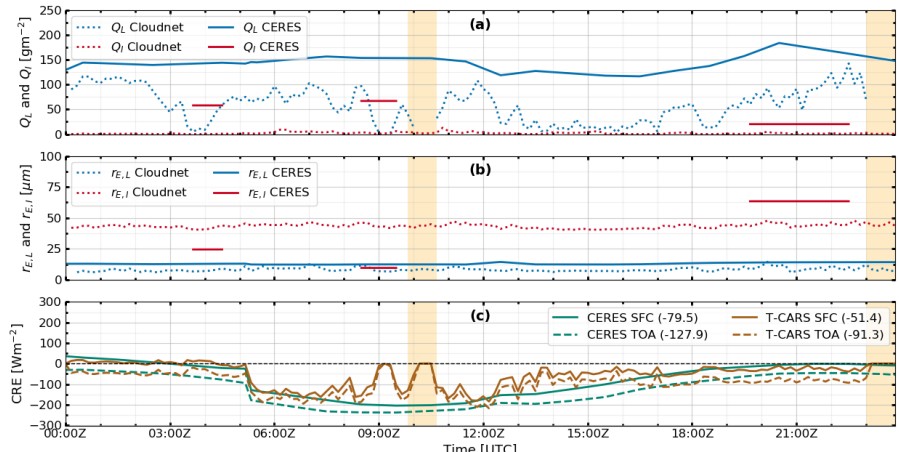

**Figure 11.** Same as Figure 9, but for the 26 June 2017 case. The pale yellow shading indicates periods with uncorrected attenuation by the Cloudnet products.


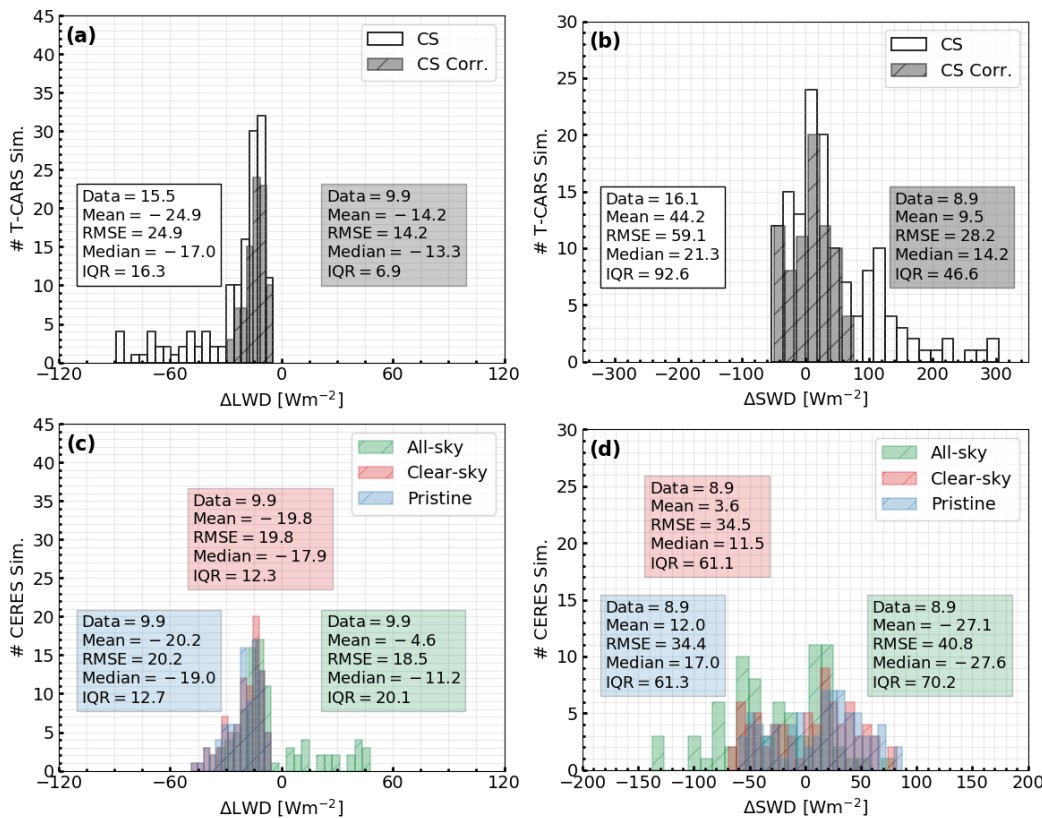

**Figure 12.** Histograms of flux difference (FD) of simulations minus observations for downward LW (LWD) flux (panels a and c) and the downward SW (SWD) flux (panels b and d) for clear-sky (CS) conditions. Panels a and b show T-CARS comparison, where the filled histograms show the filtered data by excluding the moments where the observations were compromised. Panes c and d show the comparison of CERES for All-sky (green), clear-sky (red) and pristine simulations (blue) for the same filtered time steps as in T-CARS.





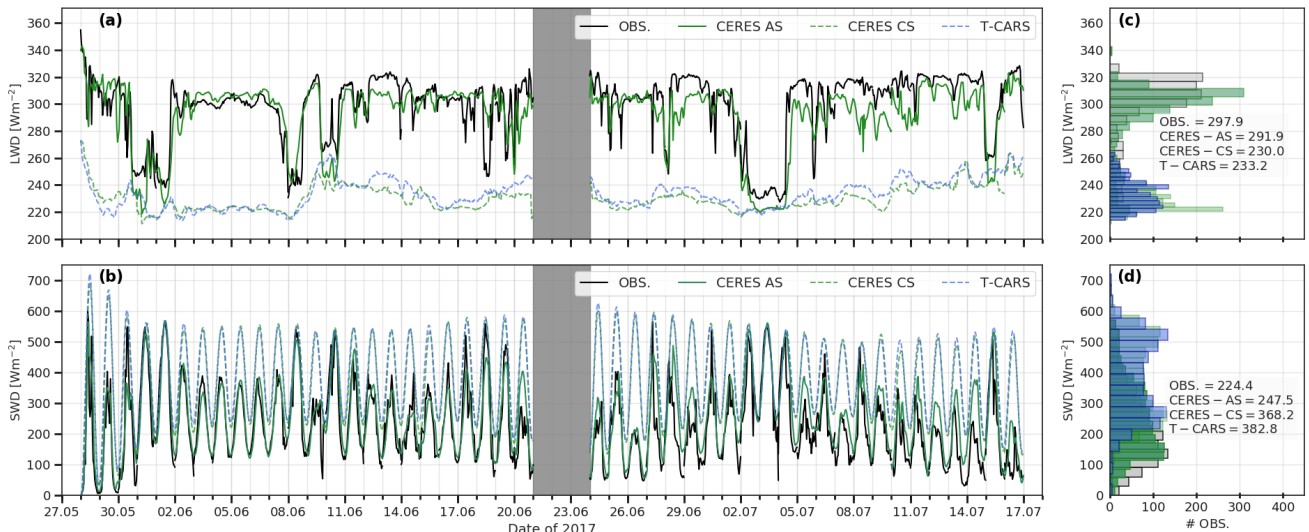

**Figure 13.** Time series of downward LW flux (LWD; panel a) and downward SW (SWD; panel b) during PS106. In panels (a) and (b), observations are shown in solid black lines, CERES all-sky (AS) products in the solid green line, clear-sky (CS) in the dashed green line, and clear-sky T-CARS in the dashed blue line. The distribution of each time series is shown in panels (c) and (d).




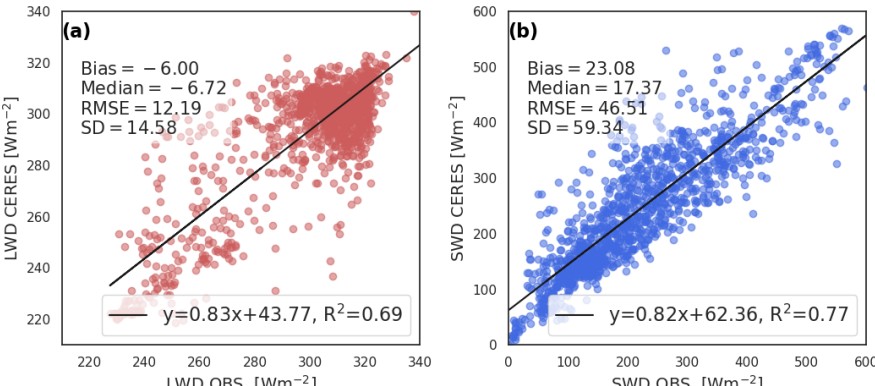

**Figure 14.** Scatter plots comparing the CERES fluxes and ship-based radiative flux observations (Obs.) for the downward LW (a) and SW (b) fluxes. The black line represents the best linear fit. The resulting fit equation and the square of the Pearson correlation coefficient ($R^2$) are shown in each panel.





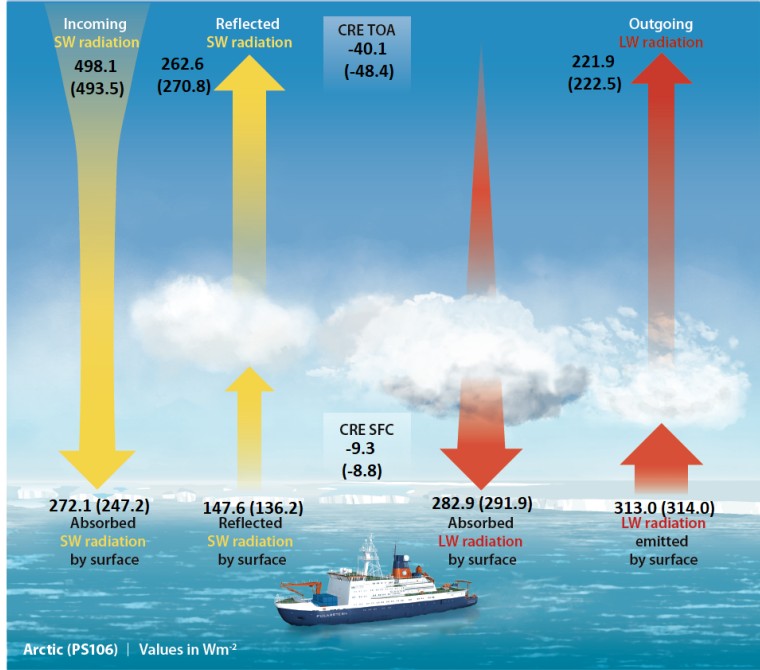

**Figure 15.** Radiation budget for the period of the PS106 cruise. Values show mean values for the Arctic (top) and PS106 expedition (bottom in parentheses). Shortwave (SW) and longwave (LW) in yellow and red, respectively. Values in grey indicate the mean cloud radiative effect (CRE) at the surface (SFC) and the top of the atmosphere (TOA). Average values are given in $\mathrm{W\,m^{-2}}$.





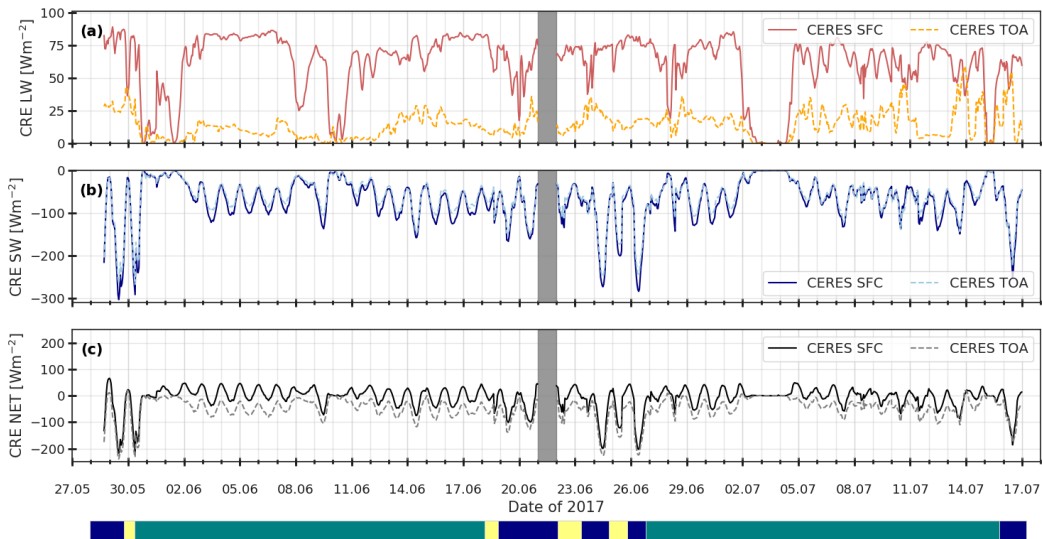

**Figure 16.** Time series of the cloud radiative effect (CRE) for the cruise track of the PS106 expedition based on CERES fluxes. Panel (a) shows the LW CRE at the surface (SFC) and top of the atmosphere (TOA). Panel (b) shows the SW CRE at the SFC and TOA, and panel (c) shows the net CRE at the SFC and TOA. The lower coloured band indicates the times when *Polarstern* was located in open ocean (blue), the marginal sea ice zone (yellow) and in mostly sea ice (teal).





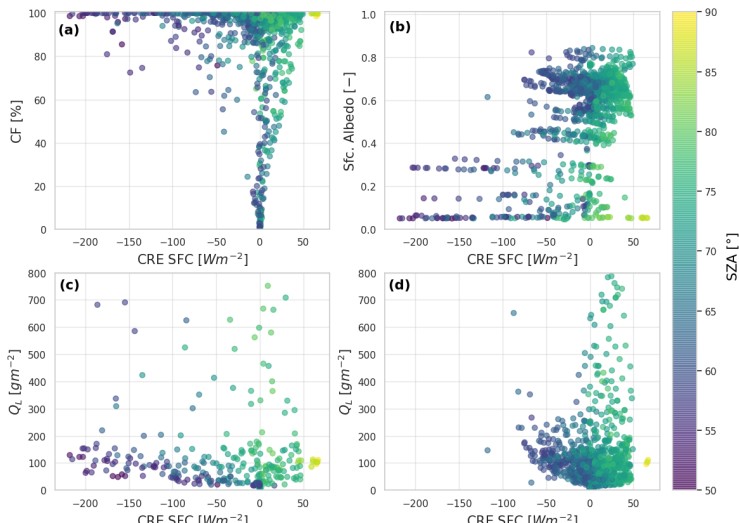

**Figure 17.** Scatter plots comparing the $CRE_{SFC}$ with several parameters. Panel (a) shows the comparison with cloud fraction (CF). Panel (b) shows the comparison of surface albedo ($\alpha$). Panels (c) and (d) show the comparison between the liquid water path ($Q_L$) at low surface albedo (<0.4) and high surface albedo (>0.4). All scatter plots are colour coded with the values of the solar zenith angle (SZA).





**Figure 18.** Maps of several time-averaged CERES data sets for the Arctic (70° to 90°N) and the time period of the PS106 expedition. Panel (a) shows the mean surface albedo, (b) the top of the atmosphere (TOA) albedo, and (c) the mean cloud fraction (CF). Panels (d), (e), and (f) show the CRE for the SW, LW, and net spectral regions, respectively, at the surface (SFC), while panels (g), (h), and (i) present the same at the TOA. An orthographic projection is used, and the PS106 cruise track is indicated in magenta.



**Table 1.** Table indicating references and remote sensing instrumentation used to derive cloud macro and microphysical properties for CERES and Cloudnet data sets. Units are shown in brackets.

| Cloud parameter | CERES | Cloudnet |
|---|---|---|
| Fraction [%] | MODIS and CERES (Minnis et al., 2020) | Lidar and radar (Illingworth et al., 2007; Hogan et al., 2009) |
| Base [hPa] or [m] | CERES (Minnis et al., 2020) | Lidar and radar (Illingworth et al., 2007; Griesche et al., 2020) |
| Top [hPa] or [m] | MODIS and CERES (Minnis et al., 2020) | Lidar and radar (Illingworth et al., 2007) |
| $Q_L$ and $r_{E,L}$ [gm$^{-2}$, $\mu$m] | MODIS and CERES (Minnis et al., 2020) | MWR and radar (Frisch et al., 1998, 2002) |
| $Q_I$ [gm$^{-2}$] | MODIS and CERES (Minnis et al., 2020) | MWR and radar (Hogan et al., 2006) |
| $r_{E,I}$ [$\mu$m] | MODIS (Minnis et al., 2020) | Cloud radar (Griesche et al., 2020) |





**Table 2.** Summary of case studies results. Values indicate the bias and the cloud radiative effect (CRE) for each case study in $\mathrm{W\,m^{-2}}$ at the surface (SFC) and the top of the atmosphere (TOA).

| Summary of case studies of 2017 | | | | | | | |
|---|---|---|---|---|---|---|---|
| Flux | FD | | | CRE | | | |
| | SFC↓ | | TOA↑ | SFC | | TOA | |
| July 3 | T-CARS | CERES | T-CARS | T-CARS | CERES | T-CARS | CERES |
| LW | -13.6 | -13.0 | 3.7 | - | - | - | - |
| SW | -7.9 | -2.1 | 10.0 | - | - | - | - |
| Net | - | - | - | - | - | - | - |
| 2 July | T-CARS | CERES | T-CARS | T-CARS | CERES | T-CARS | CERES |
| LW | -5.7 | -13.1 | -15.4 | 19.3 | 11.6 | 22.7 | 7.8 |
| SW | 6.5 | 22.2 | 15.0 | -16.6 | -10.4 | -13.7 | -18.8 |
| Net | - | - | - | 2.7 | 1.3 | 8.9 | -11.1 |
| 26 June | T-CARS | CERES | T-CARS | T-CARS | CERES | T-CARS | CERES |
| LW | -12.0 | -0.6 | 0.4 | 62.2 | 78.0 | 13.1 | 17.4 |
| SW | 77.1 | 16.1 | -26.6 | -113.6 | -157.2 | -111.7 | -140.6 |
| Net | - | - | - | -51.4 | -79.5 | -98.6 | -127.9 |





**Table 3.** The table indicates the averaged results of the cloud radiative effect (CRE) for the LW, SW, and net fluxes based on CERES for the PS106 cruise and the entire Arctic. The standard deviation is given in parentheses.

| CRE [W m$^{-2}$] | | PS106 | | | Arctic R. | | |
|---|---|---|---|---|---|---|---|
| | | LW | SW | **Net** | LW | SW | **Net** |
| TOA | Mean | 14.5 (10.2) | -62.8 (45.8) | -48.4 (42.4) | 14.3 (4.0) | -55.8 (30.1) | -40.1 (27.5) |
| | Median | 13.2 | -54.9 | -39.6 | 14.3 | -46.9 | -21.9 |
| | IQR | 13.2 | 47.3 | 45.3 | 3.6 | 25.6 | 22.8 |
| SFC | Mean | 60.8 (23.0) | -69.6 (51.9) | -8.8 (44.2) | 51.3 (9.5) | -60.6 (32.1) | -9.3 (28.1) |
| | Median | 68.2 | -60.2 | 0.6 | 51.0 | -51.8 | 2.2 |
| | IQR | 22.3 | 57.9 | 37 | 11.7 | 27.4 | 26.7 |



**Table 4.** Summary of literature comparison of the cloud radiative effect at the surface (CRE$_{SFC}$) from different studies. Values in parentheses indicate the mean for the entire study.

| Site/Project | Type | Reference | Period | Region | CRE$_{SFC}$ [W m$^{-2}$] |
|---|---|---|---|---|---|
| SHEBA | Ship-based | Intrieri et al. (2002) | Nov. 1997 to Oct.1998 | Beaufort and Chuchki Sea | CRE$_{SW}$ = 0 to 118 (-10) <br> CRE$_{LW}$ = 5 to 65 (38) |
| ASCOS | Ship-based | Sedlar et al. (2011) | Aug.-Sept. of 2008 | Svalbard and Central Arctic | CRE$_{SW}$ = 0 to -50 <br> CRE$_{LW}$ = 65 to 85 |
| M-PACE | Observatory | de Boer et al. (2011) | Sep.-Nov. of 2004 | North Slope Alaska (NSA) | CRE$_{SW}$ = 0 to -55 <br> CRE$_{LW}$ = 30 to 85 |
| Summit Station | Observatory | Miller et al. (2015) | Jan. 2011 to Oct. 2013 | Greenland | CRE$_{SW}$ = (0 to -20) <br> CRE$_{LW}$ = (30 to 60) |
| Greenland | Sat., Reanalysis Model and in situ, | Wang et al. (2019) | May 2008 to Aug. 2013 | Greenland | CRE$_{SW}$ = 0 to -70 <br> CRE$_{LW}$ = 30 to 70 |
| AWIPEV | Observatory | Ebell et al. (2020) | June 2016 to Sep. 2018 | Ny-Ålesund Svalbard | CRE$_{SW}$ = 0 to -150 <br> CRE$_{LW}$ = 20 to 60 |
| ACLOUD | Airborne | Stapf et al. (2020) | May-June 2017 | Svalbard | CRE$_{SW}$ = (-32), (-62) |
| AFLUX | | Stapf et al. (2021) | March-April 2019 | Svalbard | CRE$_{LW}$ = 75 |
| Barrow Alaska | Observatory | Dong et al. (2010) | June 1998 to May 2008 | BRW and NSA | CRE$_{SW}$ = 0 to -90 <br> CRE$_{LW}$ = 5 to 65 |
| Arctic Region | Satellite | Kay and L'Ecuyer (2013) | March 2000 to Feb. 2011 | Arctic | CRE$_{SW}$ = 0 to -75 (-32) <br> CRE$_{LW}$ = 35 to 70 (42) |





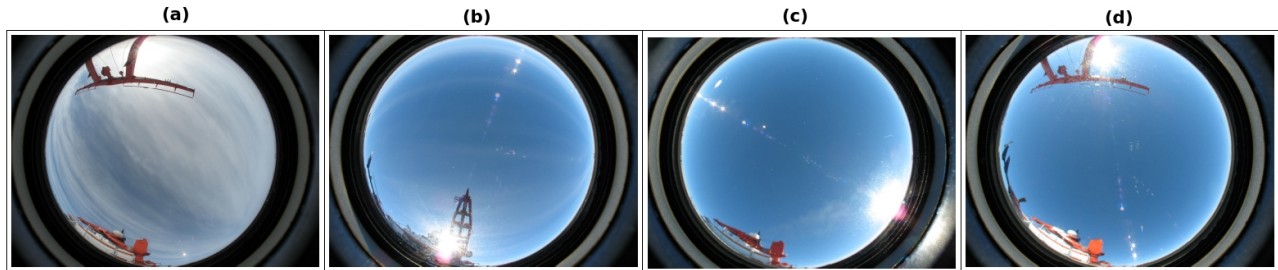

**Figure A1.** Sky-camera photographs for 2 July 2017 04:00:08Z (a), for 2 July 2017 12:13:33Z (b), 3 July 2017 04:14:09Z (c), and 3 July 2017 12:04:08Z (d). All times in UTC.





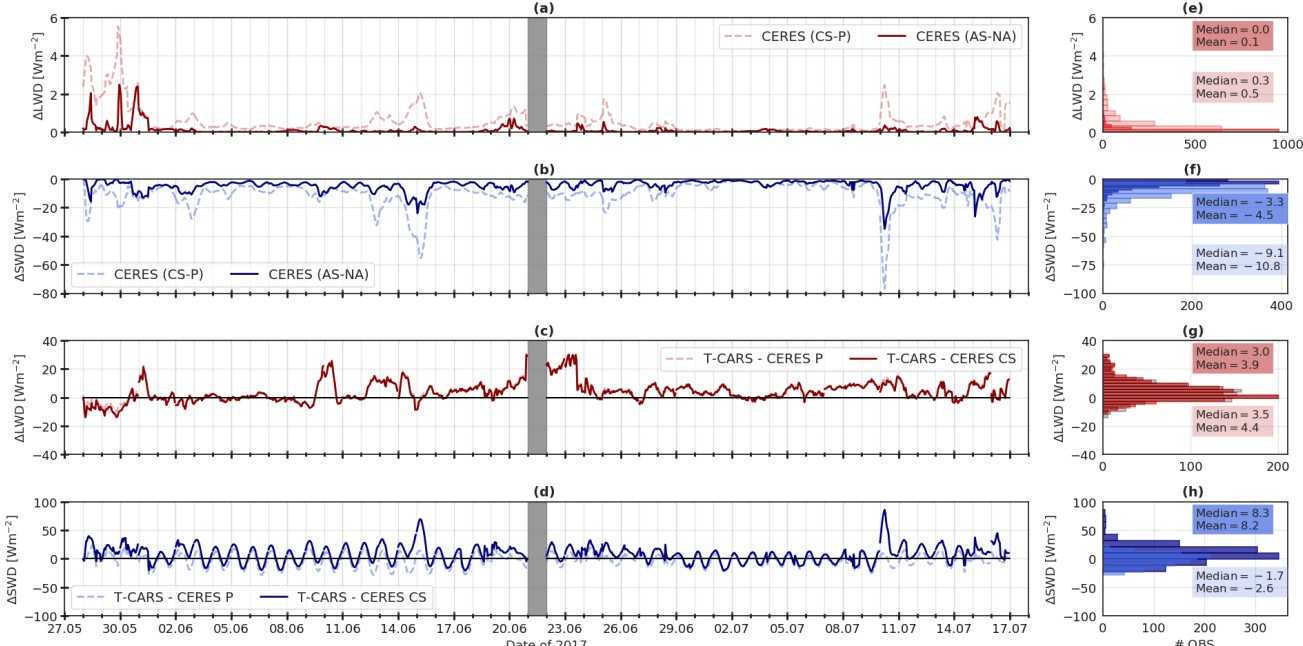

**Figure B1.** Time series of the difference of downward surface radiative fluxes between CERES and T-CARS for pristine (P), clear-sky conditions (CS), all-sky (AS), and cloudy without aerosols (NA). Panel (a) shows the difference of the downward LW (LWD) CERES (CS-P) and CERES (AS-NA). Panel (b) shows the same as (a), but the downward SW (SWD) flux. Panel (c) shows the LWD differences between T-CARS and CERES P, and T-CARS minus CERES CS. The histograms of each left-hand side panel are shown on the right. The LWD and SWD fluxes are shown in red and blue, respectively.



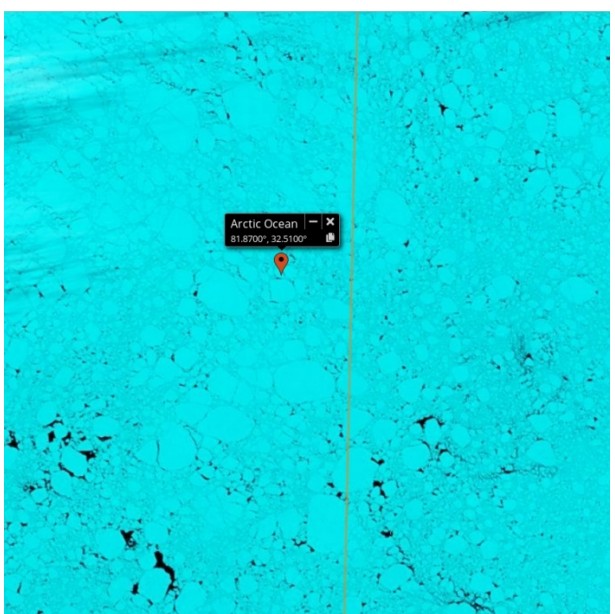

**Figure C1.** Moderate Resolution Imaging Spectroradiometer (MODIS) screenshot for 3 July 2017. The red symbol indicates *Polarstern*'s location. The orange line indicates MODIS track. Image obtained from EOSDIS Worldview 2021 Version 3.15.





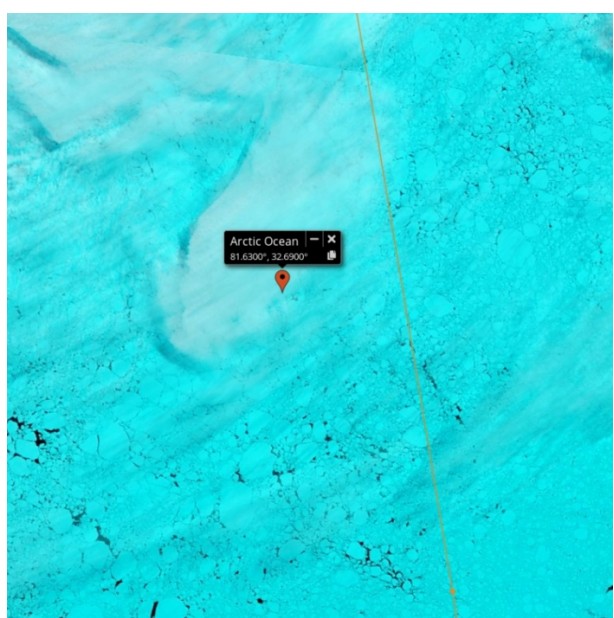

**Figure C2.** Same as Fig. C1, but for 2 July 2017. Image obtained from EOSDIS Worldview 2021 Version 3.15.





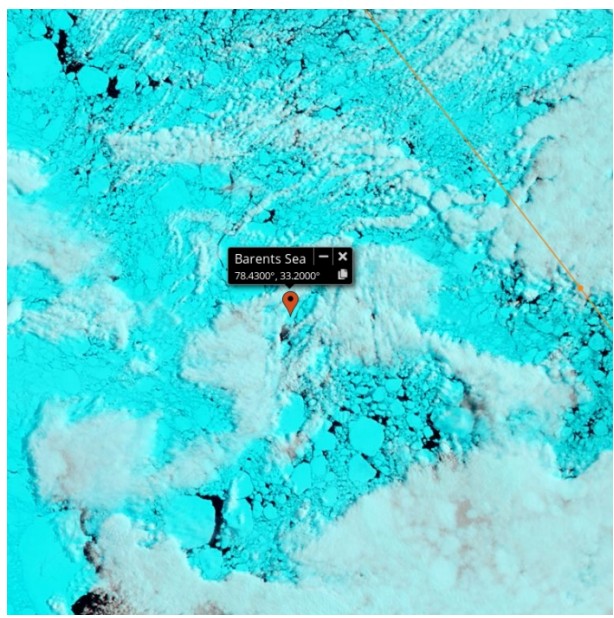

**Figure C3.** Same as Fig. C1, but for 26 June 2017. Image obtained from EOSDIS Worldview 2021 Version 3.15.





**Table A1.** Results of the sensitivity analysis, varying several atmospheric parameters. The table indicates the mean upward (U) and downward (D) SW and LW flux differences at the surface of clear-sky perturbed simulations minus idealized atmosphere simulation. Values in $\mathrm{W\,m^{-2}}$.

| | Atmospheric parameters | | | | | | | |
|---|---|---|---|---|---|---|---|---|
| Parameter | Variation | LWD | LWU | Net LW | SWD | SWU | Net SW | **Net** |
| Temperature | ±7 [K] | ±14.9 | - | ±14.9 | ±0.2 | ±0.1 | ±0.1 | ±14.8 |
| | ±0.5 [K] | ±**1.0** | - | ±1.0 | ±0.0 | ±0.0 | ±0.0 | ±1.0 |
| Ozone | ±25 [%] | ±0.6 | - | ±0.6 | ±1.4 | ±0.9 | ±0.5 | ±0.0 |
| | ±12.5 [%] | ±**0.3** | - | ±0.3 | ±**0.7** | ±0.5 | ±0.3 | ±0.0 |
| Water vapour | ±15 [%] | ±3.3 | - | ±3.3 | ±2.7 | ±1.8 | ±0.9 | ±2.4 |
| | ±5 [%] | ±**1.0** | - | ±1.0 | ±**0.9** | ±0.6 | ±0.3 | ±0.7 |





**Table A2.** Variation of several surface parameters. The table indicates the mean upward and downward SW and LW flux differences at the surface of clear-sky perturbed simulations minus idealized atmosphere simulation. Values in $\mathrm{W\,m^{-2}}$.

| Surface parameters | | | | | | | | |
|---|---|---|---|---|---|---|---|---|
| Parameter | Variation | LWD | LWU | Net LW | SWD | SWU | Net SW | **Net** |
| Skin temperature | ±5 [K] | ±5.3 | ±23.6 | ±18.3 | - | - | - | ±18.3 |
| | ±0.3 [K] | **±0.3** | ±1.4 | ±1.1 | - | - | - | ±1.1 |
| Albedo[0.65] | 0.05 [-] | - | - | - | -13.9 | -241.1 | 227.2 | 227.2 |
| | 0.30 [-] | - | - | - | -8.5 | -142.8 | 134.3 | 134.3 |
| | 0.53 [-] | - | - | - | -3.0 | -49.7 | 46.7 | 46.7 |
| | 0.84 [-] | - | - | - | 5.1 | 8.4 | -75.3 | -75.3 |
| | ±0.08 [-] | - | - | - | **±2.1** | ±33.6 | ±31.5 | ±32.0 |
| Emissivity [0.9999] | 0.9980 [-] | - | -0.2 | 0.2 | - | - | - | 0.2 |
| | 0.9907 [-] | - | -0.8 | 0.8 | - | - | - | 0.8 |





**Table A3.** Variation of several atmospheric parameters. The table indicates the mean upward SW, LW, and net flux differences at the top of the atmosphere of clear-sky perturbed simulations minus the created idealized atmosphere. Values in $\mathrm{W\,m^{-2}}$.

| Atmospheric parameters | | | | |
|---|---|---|---|---|
| Parameter | Variation | LWU | SWU | **Net** |
| Temperature | ±7 [K] | ±15.4 | ±0.1 | ±15.3 |
| | ±0.5 [K] | ±**1.1** | ±0.0 | ±1.1 |
| Ozone | ±25 [%] | ±0.6 | ±1.7 | ±2.3 |
| | ±12.5 [%] | ±**0.3** | ±**0.8** | ±1.1 |
| Water vapour | ±15 [%] | ±1.5 | ±2.2 | ±3.7 |
| | ±5 [%] | ±**0.5** | ±**0.7** | ±1.2 |





**Table A4.** The table indicates the mean upward SW, LW, and net flux differences at the top of the atmosphere of clear-sky perturbed simulations minus idealized atmosphere simulation. Values in $\mathrm{W\,m^{-2}}$.

| Surface parameters | | | | |
|---|---|---|---|---|
| Parameter | Variation | LWU | SWU | **Net** |
| Skin Temperature | ±5 [K] | ±8.1 | ±0.0 | ±8.1 |
| | ±0.3 [K] | ±**0.5** | ±0.0 | ±0.5 |
| Albedo [0.65] | 0.05 [-] | - | -217.6 | -217.6 |
| | 0.30 [-] | - | -128.7 | -128.7 |
| | 0.53 [-] | - | -44.7 | -44.7 |
| | 0.84 [-] | - | 72.2 | 72.2 |
| | ±0.08 [-] | - | ±**30.2** | ±30.2 |
| Surface emissivity [0.9999] | 0.9980 [-] | -0.1 | 0.0 | -0.1 |
| | 0.9907 [-] | -0.7 | 0.0 | -0.7 |





**Table B1.** Hourly averaged radiative flux comparison of T-CARS, CERES and ship-borne observations of downward SW and LW fluxes for clear-sky conditions (simulations - observations). Values indicate the bias, the standard deviation of the flux difference (STD), the correlation coefficient ($R^2$), and the root-mean-square error (RMSE).

|     | Data set | Bias (W m$^{-2}$) | Median | STD (W m$^{-2}$) | $R^2$ (-) | RMSE (W m$^{-2}$) |
| --- | --- | --- | --- | --- | --- | --- |
|     | T-CARS$_{Corr}$ | -14.2 | -13.3 | 5.6 | 92.4 | 14.2 |
|     | T-CARS | -24.9 | -17.0 | 31.0 | 44.1 | -24.9 |
| LWD | CERES | -4.6 | -11.2 | 20.9 | 47.6 | 18.5 |
|     | CERES$_{CS}$ | -19.8 | -17.9 | 9.3 | 77.3 | 19.8 |
|     | CERES$_P$ | -20.2 | -19.0 | 9.4 | 76.6 | 20.2 |
|     | T-CARS$_{Corr}$ | 9.5 | 14.2 | 31.9 | 95.0 | 28.2 |
|     | T-CARS | 44.2 | 21.3 | 74.0 | 0.85 | 59.1 |
| SWD | CERES | -27.1 | -27.6 | 44.0 | 93.0 | 40.8 |
|     | CERES$_{CS}$ | 3.6 | 11.5 | 38.0 | 95.0 | 34.5 |
|     | CERES$_P$ | 12.0 | 17.0 | 40.8 | 95.0 | 34.4 |





**Table C1.** Radiation budget for PS106 for all-sky (AS) and clear-sky (CS) conditions based on CERES data set. Values represent mean and standard deviation (in parentheses) from 28 May to 20 June, 22 June to 16 July 2017.

| Radiative Flux | | LW | | | SW | | | Net |
|---|---|---|---|---|---|---|---|---|
| | | ↓ | ↑ | Net | ↓ | ↑ | Net | |
| PS106 CS | TOA | - | 237.0(4.1) | -237.0(4.1) | 493.5(150.0) | 207.9(80.9) | 285.6(121.6) | 48.6(120.5) |
| | ATM | -229.8(9.2) | -75.7(7.3) | -154.1(9.6) | 125.3(24.0) | 20.2(14.9) | 105.1(24.2) | -49.0(25.0) |
| | SFC | 229.8(9.2) | 312.7(9.9) | -82.9(10.0) | 368.2(132.9) | 187.7(94.2) | 180.5(104.9) | 97.6(101.9) |
| PS106 AS | TOA | - | 222.5(10.8) | -222.5(10.8) | 493.5(150.0) | 270.8(87.1) | 222.8(94.0) | 0.2(91.9) |
| | ATM | -291.9(24.5) | -91.5(15.3) | -200.4(22.0) | 246.4(82.9) | 134.5(64.2) | 111.8(30.1) | -88.6(33.7) |
| | SFC | 291.9(24.5) | 314.0(10.0) | -22.1(22.0) | 247.2(116.6) | 136.2(85.8) | 110.9(75.1) | 88.8(67.1) |
| Arctic CS | TOA | - | 236.2(8.9) | -236.2(8.9) | 498.1(179.8) | 207.3(95.0) | 290.8(147.8) | 47.9(150.5) |
| | ATM | -230.4(23.1) | -75.6(14.0) | -154.8(11.1) | 126.7(34.6) | 21.0(18.1) | 105.7(31.1) | -55.8(47.0) |
| | SFC | 230.4(23.1) | 311.8(21.9) | -81.4(11.1) | 371.4(156.3) | 186.4(110.2) | 185.1(123.7) | 103.7(122.4) |
| Arctic AS | TOA | - | 221.9(13.3) | -221.9(13.3) | 498.1(179.8) | 262.6(99.4) | 235.5(117.7) | 7.8(113.6) |
| | ATM | -282.9(34.0) | -91.0(21.9) | -191.8(23.8) | 226.0(99.5) | 115.0(76.4) | 111.0(36.7) | -86.6(45.3) |
| | SFC | 282.9(34.0) | 313.0(22.0) | -30.1(23.8) | 272.1(142.8) | 147.6(104.4) | 124.5(91.0) | 94.4(85.1) |





**Table C2.** Radiation budget for PS106 for pristine (P) and cloudy without aerosols (NA) conditions based on CERES data set from 28 May to 20 June, 22 June to 16 July 2017.

| Radiative | | LW | | | SW | | | **Net** |
|---|---|---|---|---|---|---|---|---|
| Flux | | ↓ | ↑ | Net | ↓ | ↑ | Net | |
| PS106 P | TOA | - | 237.1 | -237.1 | 493.5 | 207.7 | 285.8 | 48.7 |
| | SFC | 229.2 | 312.6 | -84.4 | 379.0 | 193.8 | 185.2 | 100.8 |
| PS106 NA | TOA | - | 222.6 | -222.6 | 493.5 | 272.6 | 220.9 | -1.7 |
| | SFC | 291.8 | 314.0 | -22.2 | 251.7 | 138.9 | 112.8 | 90.6 |
| Arctic P | TOA | - | 236.3 | -236.3 | 498.1 | 207.1 | 291.0 | 54.7 |
| | SFC | 229.7 | 311.7 | -82.0 | 385.3 | 193.8 | 191.5 | 109.5 |
| Arctic NA | TOA | - | 222.0 | -222.0 | 498.1 | 264.7 | 233.4 | 11.4 |
| | SFC | 282.7 | 313.0 | -30.3 | 279.3 | 151.8 | 127.5 | 97.2 |