# Peer review of "Radiative closure and cloud effects on the radiation budget based on satellite and ship-borne observations during the Arctic summer research cruise PS106"

_Atmospheric Chemistry and Physics, 2021_

## Author Response (AR1)

**Authors' response – ACP**

**Radiative closure and cloud effects on the radiation budget based on
satellite and ship-borne observations during the Arctic summer
research cruise PS106
by Barrientos Velasco et al.**

Response to Anonymous **Referee #1** and **Referee #2**

We would like to thank Anonymous Referee # 1 and Referee #2 for dedicating time and giving suggestions to the improvement of the manuscript by providing us with valuable comments. We have revised the initial submission version and hope that the manuscript is now acceptable for publication.

The point-by-point response to the review comments is written here in italic-grey font. Additionally, in green is marked the location of the modification of the text in the diff.pdf file.

**Overall summary of major changes:**

We would like to inform the referee about the following major changes based on the comments of Referee #1 and #2:

- Figure 17 and Table B1 were deleted from the manuscript.

**Clarifications for specific comments**

- 160-163: please provide more details on how q_L is determined (equation?) No information on the retrieval of r_E,L is given. How do you handle mixed-phase clouds, since in these cases the "standard" retrievals for liquid clouds do not work?

  *It was not our aim to describe in detail the Cloudnet methodology since this was covered in Griesche et al., 2020. However, we hope that with the modified version of the manuscript the question has been answered. See below an extract of the modified text in Section 2.1.2 (Page 6).*

  Once a cloudy pixel is identified, the cloud water content and effective radius are determined regardless if the cloud is detected as a single or mixed-phase cloud. The liquid water path $(Q_L)$ is retrieved based on the HATPRO MWR measurements using the retrieval method developed in Löhnert and Crewell (2003). This method relies on a long-term radiosonde training data set, which in this case is based on Ny-Ålesund, NO (78.9°N, 11.85°E, WMO Code 6260). Once $Q_L$ is known, the liquid water content $(q_L)$ and $r_{E,L}$ are determined. The $r_{E,L}$ considers the radar reflectivity factor and measurements of the integrated cloud liquid water based on the methodology described in Frisch et al. (2002). The retrieval of $q_L$ is obtained by distributing the observed the $Q_L$ adiabatically among the identified liquid and mixed-phase cloud pixels identified classified by the Cloudnet algorithm. This method assumes a log-normal cloud-droplet distribution, which is constant with height. The uncertainties of $q_L$ are calculated by error propagation assuming a typical uncertainty of 20-25 gm$^{-2}$ gm$^{-2}$ in $Q_L$ (Löhnert and Crewell, 2003). The adiabatic increase of liquid water is based on Brenguier (1991) and calculated by the following equation in $\text{kg kg}^{-1}\,\text{m}^{-1}$.

  $$\frac{dq_L}{dz} = -\left(1 - \left(\frac{C_p \cdot T}{L \cdot e}\right)\right) \cdot \left(\frac{1}{\left(\frac{C_p \cdot T}{L \cdot e}\right) + \left(\frac{L \cdot qs \cdot \rho a}{p - es}\right)}\right) \cdot (\rho a \cdot g \cdot e \cdot es) \cdot (p - es)^{-2}. \tag{1}$$

  In Eq. 1, $T$ is the atmospheric temperature in Kelvin, $p$ is atmospheric pressure in Pascals, $qs$ is the specific humidity mixing ratio in $\text{kg kg}^{-1}$, $es$ is the saturated vapour pressure, $\rho a$ is the density of air in $\text{kg m}^{-3}$ (see Eq. 2), $e$ is the ratio of the molecular weight of water vapour of dry air equal to 0.62198, $g$ is the acceleration due to gravity -9.81 $\text{m s}^{-2}$, $C_p$ is the heat capacity of air at a constant pressure 1005.0 $\text{J kg}^{-1}\text{K}^{-1}$, $L$ is the latent heat of evaporation $2.5 \times 10^6$ $\text{J kg}^{-1}$, $R_d$ is the specific gas constant for dry air 287.04 $\text{J kg}^{-1}\text{K}^{-1}$.

  $$\rho a = \frac{p}{R_d \cdot (1 + 0.6 \cdot qs) \cdot T}. \tag{2}$$

- 164-166: maybe Q_i can be introduced here as well. I assume that it is calculated by vertically integrating over q_i?

  *Yes, it is vertically integrated over q_i. The text has been clarified to (Page 6 and 7):*

  *'The ice water content ($q_I$) is obtained based on the measurements from the cloud radar for pixels flagged as ice or mixed-phase cloud (Hogan et al., 2006). The ice water path $Q_I$ is calculated by integrating vertically $q_I$. These parameters depend on temperature (T; °C) and cloud radar reflectivity (Ze; dBZ).'*

- 202-206: I assume that the cloud properties from CERES are not vertically resolved. Please clarify in the text.

  *The methodology of CERES vertically bines the microphysical values at four different heights based on the MODIS cloud products which have higher spatial resolution and several assumptions. The text has been clarified to (Page 8):*

  *'The parameters considered in this study are the cloud fraction (CF), QL, QI , $r_{E,L}$, $r_{E,I}$ , cloud base ($P_B$) and top pressure ($P_T$). The cloud properties are reported for four atmospheric pressures intervals.*

*Nevertheless, it has been decided to consider the total atmospheric values for the analysis. Note that cloud properties mentioned are retrieved based on MODIS retrievals of cloud emissivity, cloud effective temperature, cloud particle effective radius, and cloud optical thickness.'*

- 220-: Is the vertical grid used in T-CARS determined by the Cloudnet vertical grid? Do you need to interpolate cloud properties in time? I assume that the Cloudnet temporal grid has a resolution of 30 s.

*The radiative transfer model has a limit of 200 atmospheric levels. Therefore, it was not possible to proceed with the exact same vertical grid as Cloudnet since Cloudnet has 413 atmospheric levels equally separated every by ~31.2 metres, starting from 165 metres to 13 km. It was opted to use the double Cloudnet's vertical grid (i.e., 62.5 m) for the first 10 km of the atmosphere. The temporal resolution of Cloudnet is about 30 s, so the cloud properties were interpolated to every minute for the simulations. This is explained in more detail in section 2.3 of the manuscript.*

- 229: Do you have independent measurements of surface skin temperature (maybe for a shorter time period) during the Polarstern cruise? How well does T10m with the surface skin temperature agree? I am wondering how large the uncertainty of the LW upward flux at the surface is due to this assumption.

*Unfortunately, the skin temperature was not measured during PS106., thus we do not know how well T10m agrees with the surface skin temperature based on point measurements. However, if Figure 2 is shown a relatively good agreement between T10m measurements and the skin temperature from CERES SYN1deg and ERA5, with the exception of particular cases (e.g., 31. May.2017, 7. June.2017, 22. June.2017, 2-5.July.2017).*

*As our first aim was to compare simulated downward fluxes with observations, we considered the skin-temperature to be relevant only on the LW upward fluxes. It should be note, however, that this will be also part of our future analysis to quantify the uncertainty of the LW upward based on that assumption for MOSAiC (Multidisciplinary drifting Observatory for the Study of Arctic Climate). We edited the outlook section to make clear this aspect.*

*(Page 27) 'Moreover, given the importance of the surface albedo and the skin temperature to the interpretation of the radiation budget, it is planned to evaluate the local values observed during MOSAiC to the values used in this study.'*

- 279- ... and Fig 2a: Is the intercomparison of CERES and ERA5 surface albedo really needed? You further use CERES, which you state "yield accurate results". An evaluation of ERA5 surface albedo is in my option out of the scope of this study. Since the paper is already very long, this discussion could be removed and just the CERES albedo shown.

*After a careful consideration and based also on suggestion from Referee #2, we have decided to exclude the comparison of surface albedo between ERA5 and CERES.*

- 296-... and Fig 2b) Is there a need to show the T10m from the radiosonde? There are sometimes larger discrepancies between T10m from the mast and the radiosonde. I am also not sure how trustworthy the radiosonde measurements are at that height. I suggest to omit the RS T10m here. If you have surface skin temperature measurements which have been taken during the Polarstern cruise (also for shorter time periods), it would be interesting to see those ones.

*We agree with the Referee, and we decided to exclude the temperature from the radiosondes from Fig 2b. As mentioned earlier, we did not measure surface skin temperature. Therefore, the comparison suggested for shorter periods was not made.*

- Fig 3. (but also the other figures): The figure captions could be more concise. In this way the relevant information can be better captured. E.g. for Fig 3:

   "Time-height plot of atmospheric profiles obtained along the PS106 cruise track. (a) ERA5 atmospheric temperature anomalies. (b) ERA5 specific humidity anomalies. Anomalies have been calculated with respect to the mean profiles of the cruise. (c) Mean profiles of atmospheric temperature and (d) mean profiles of specific humidity for ERA5 (orange) and radiosondes (blue). The sub-Arctic summer standard atmosphere (Anderson et al., 1986) is displayed in black. The grey-shaded area indicates the minimum and maximum values, while the brownish-shaded area shows the interquartile range of the ERA5 profiles."

   *Thanks for the example. We made the captions more concise.*

- Fig. 4a: I am not sure which information to take from this plot. It is also not really discussed in the text. I would remove this plot. Instead you could add a plot of the vertical cloud fraction/ frequency of cloud occurrence for the entire cruise.

   *We opted to remove the plot for simplicity. Our aim is to compare Cloudnet and CERES cloud fractions; therefore, we now show just panel b.*

- Fig. 5 a: I find that, in stacked column charts, individual values are sometimes difficult to capture. Since the focus in Fig. 5b is on single-layer clouds, it would be good if the values (RFO) of the single-layer clouds could be easily read. So I suggest to change the order of the columns. At the bottom "single-layer", followed by "multi-layer", then "clear sky" and then "no data" on top. The colors could be the same.

   *We understand the Referee's concern. We changed the order of Figure 5a as suggested, and change the position of 'No data' in c for consistency.*

- Fig. 5b: I am confused here. In Fig. 5 b, the phase of single-layer clouds shall be shown. Since you focus on this class only, why are there again the "clear-sky" and "no data" categories? Either you show "clear-sky", "no data", "multi-layer", and "single-layer", while for single-layer you discriminate also the different phases (that would be somehow the same plot as in Fig. 5a except that the single-layer class has further sub-classes). Or you simply show the single-layer statistics only so that the phase types sum up to 100%. The latter would be my preference.

   *The aim was to show the sub-classification of cloud phase of single-layer clouds in a realistic percentage. We thought that if the percentages of single layer clouds sum up 100% for a day with considerable data gaps or clear-sky it could mislead the reader. However, considering that the reader can obtain this information from Fig. 5A, we decided to change the Fig.5 b as the Referee suggests.*

   Why can the thermodynamic phase not be given in case that there is precipitation?

   *As per the standarized Cloudnet classification only liquid is classified as precipitation ('drizzle/rain and cloud droplets' and drizzle or rain). When there is snowfall, there is a flag within Cloudnet data*

*related to an uncorrected attenuation. It has not been planned to subdivide precipitation within the standard Cloudnet target classification.*

- ll 387-396: As far as I understood, water vapor is also taken from the ERA5 reanalysis. What is the uncertainty due to the fact that hourly values are interpolated to the minute resolution of the RT simulations? You only include instrumental uncertainties in your sensitivity analysis but I think that the interpolation of hourly values might also cause uncertainties at least in the same order of magnitude. Same for the temperature profile. You have temporally highly resolved IWV measurement on board from the HATPRO radiometer. Why didn't you include these ones?

  *It is a good suggestion. In the analysis, we conducted several experiments using several data sets as inputs [i.e., data obtained directly from radiosondes, GDAS and ERA5]. Given the good agreement between radiosondes and ERA5, we opted for the most consistent option that can cover from the surface to 20 km height. However, this suggestion can be implemented and tested for MOSAiC.*

- ll 397-404: Also for the uncertainty of the surface skin temperature a small value of 0.3°C is used (at least in the final estimate of the overall uncertainty). This uncertainty is related to measurement accuracy of the T10m temperature sensor but does not include the uncertainty due the assumption that Ts=T10m. Can you comment?

  *As mentioned in the comment on line 229, the skin temperature was not measured during PS106. Thus, an estimate of its uncertainty was not derived. Given that the ship-borne instrument at 10 m was used as an alternative, we considered the uncertainty of this sensor.*

- ll 435-437: I would argue that the uncertainty estimates that you use to determine the overall uncertainty are rather at the lower end and most likely much higher due to the reasons mentioned before. Also for the SW, you omit the uncertainty related to aerosols which probably also has a high impact.

  *The values obtained from the sensitivity analysis are an approximate based on the input parameters used for the radiative transfer simulations during clear-sky conditions. Thus, it is used as a reference for subsequent studies.*

  *While we did not include aerosols in the simulations due to the lack of observations, we quantified the radiative effect on the SWD from CERES with Figure B1. Under clear-sky conditions, the aerosols can contribute with a mean of -10.8 Wm$^{-2}$ and under cloudy conditions up to -4.5 10.8 Wm$^{-2}$.*

- ll 450-451: I would add "at the TOA" at the end of the sentence.

  *The text has been changed accordingly.*

- ll 477: "Therefore, our results confirm..."
  It is true that the bias of LW_down in the clear sky case is smaller than your uncertainty estimate. Still I am a little bit concerned by this bias since you clearly see a systematic error which hints at too low temperatures and/or humidity in the RT simulations. Did you try to provide a better estimate of your input profiles, i.e. including the T10m in the lowest atmospheric level or by including the IWV from MWR or GPS?

  *Initially, we compared different atmospheric input data sets into the radiative transfer model [i.e., radiosondes, ERA5] and analysed which results were closer to the observations. However, we did not*

*find large differences between both simulations since, in general, there is a good agreement between ERA5 and radiosondes (see Fig. 3c and 3d). For consistency, we decided to use ERA5 as a homogeneous input parameter. Additionally, we created several sensitivity analyses varying the humidity and temperature profiles to determine how more humid or warmer the atmosphere needed to be to match the LWD observations. In some cases, the mean flux difference was larger than the radiosonde uncertainties (see section 3.2 and Table A1). Therefore, we hypothesised that the biases of the downward LW might be due to the temperature fluctuation of shipborne instrumentation working near the pyrgeometer (see conclusion point 3).*

I am also wondering if the bias only is a good indicator that RC is reached. Looking at the SWD flux differences there are quite some larger differences (second peak around -30 Wm-2, where does this come from?). So shouldn't be also the RMSE or STD discussed with respect to radiative closure?

*The second peak around -30Wm-2 come from instances where the position of the sun was near the edge of Polarstern (65.6° in solar zenith angle) at 17:00Z approximately (see Fig. 7b). The multiple reflections caused by the relatively low sun position might have caused an increase in the SWD, increasing the bias from the simulations and observations.*

*While the RMSE and STD contribute to the interpretation of the radiative comparison, we opted highlighting the importance of the bias since this term is associated with the accuracy of the simulations in contrast to the observations. The RMSE refers more to how spread the simulated residual errors are and might weigh the effect of outliers significantly. The SD is a good indicator of how the simulated values match the mean observations. For the variability of the data points, the interpretation of the SD can be misleading for a general interpretation, especially in the shortwave when the fluxes vary largely due to the sun's position.*

[Figure]

*Figure 1: Shipborne sky-camera image for 03.07.2017 at 17:05:02 UTC.Image obtained during PS106/2 based on sky-camera installed on Oceanet container*

- Fig. 7: Could you add directly bias, RMSD,.... in d-b instead of putting them in the Table 2? (and enlarge Figures d-b). This would be much easier for the reader. Same for the plots of the other case studies.

  *We considered a version of the plots with the RMSD, but unfortunately the values were too small or over-layed the histograms, so we decided to include a table with these values.*

- ll 499-500: "...that RC is achieved..." Again, this holds maybe for the daily mean value when you refer to the bias, but for individual 10 min intervals, RC is not always reached. Maybe you can more carefully differentiate here. RC depends also on the averaging time (10-min values, daily mean values,..)

  *We changed the text style to (Page 17):*
  *'In general, this comparison suggests that RC is achieved for T-CARS and CERES SYN1deg considering the daily mean'.*

- 502-503: I think that also 3D cloud effects play a substantial role. Can you comment on this?

  *It is an interesting remark. Perhaps part of the biases was due to 3D cloud effects. Unfortunately, we cannot confidently confirm this statement based on the 1-D set-up of our radiative transfer model.*

- 504-511: How do you compare the vertical profile of ref_ice and ref_liq with the values from CERES (which are most likely not vertically resolved and representative for only a certain part of the cloud?)?

  *Cloudnet provides vertical profiles of ref_ice and ref_liq. CERES provides total values for the 1° x 1° as well as some estimates at four different heights (Low: Surface-700 mb, Mid-low: 700-500mb, Mid-high:500mb-300mb, High:300mb).*

  *For the comparison we considered the mean value obtained from CERES and the maximum value derived by Cloudnet are displayed. The text has been modified to (Page 17):*

  *'Panels a and b of Fig. 10 show the time series of the Q and $r_E$ obtained from Cloudnet and CERES SYN1deg, respectively. The comparison of Q shows the integrated values for the entire atmosphere, whereas panel b is displayed the mean values obtained from CERES SYN1deg and the maximum derived by Cloudnet. Despite the difference in retrieval methods, there is, in general, a good agreement of the values of $Q_l$ and $r_{E,l}$ from CERES SYN1deg and Cloudnet (see Fig. 10a and 10b).'*

- 512-516: I would be careful in stating that the net CRE_sfc from T-CARS and CERES are consistent. The daily mean values are similar but there are quite big differences in the individual values.

  *We agree with the comment. The text has been changed to the following (Page 17):*

  *'For this case study, the net $CRE_{SFC}$ is similar between T-CARS and CERES SYN1deg, despite the noted discrepancies in cloud properties. The net $CRE_{SFC}$ has mean values of 1.3 Wm² and 2.7 Wm² for CERES SYN1deg and T-CARS, respectively.'*

- 525-527: Q_L is from the MWR. So why could it not be derived "due to uncorrected attenuation"?

  *Usually this occurs when no values of Q_L are observed due to instrumental issues.*

- 535-540: I see the point that in case of low LWP, the relative uncertainty of MWR LWP is high. Do you have LWP from IR measurements on this day as well? Maybe this would give you a better idea. However, I do not trust the CERES LWP neither. How realistic are the constant reff values of CERES?

  *We did not have IR measurements on this day. During PS106 there were IR measurements, however for this day no measurements were made. (See Richter et al., 2021; https://essd.copernicus.org/preprints/essd-2021-284/)*

  *CERES LWP are based on MODIS observations and in the same way as the Referee we are aware of the limitations that these observations might had in the polar regions.*

- 561-564: Still I am concerned about this bias in the LW simulation in T-CARS…

  *The LW bias has been observed for CERES and T-CARS (see Fig. 12a and Fig. 12c). Our hypothesis also includes that the operation of the shipborne instrument near the pyrgeometer might have caused an increase in the instrumental uncertainties, or the ship acted as a rather warm island in contrast to the ice floe. We, therefore, plan to compare shipborne and ice floe pyrgeometer measurements carried out during MOSAiC to test this hypothesis. (see conclusion number 2).*

- 569: What is the "pristine" CERES product? The different CERES products need to be introduced in more detail in the data section.

  *We agree with the Referee's comment. The text has been changed to the following (Page 8):*

  *'The CERES SYN1deg flux products considered in this study provide fluxes based on an all-sky (AS), cloudy without aerosols (NA), clear-sky (CS), and virtually pristine (P, neither clouds nor aerosols) scenario.'*

- 580- …: "All-sky", shouldn't this be discussed rather in the next section?
  Fig 12/Table B1: all values of Table B1 are already included here. Table B1 is not needed.

  *In this comparison, we emphasized that the clear-sky comparison was made with the clear-sky classification obtained by Cloudnet and the use of the fish-eye sky camera. However, CERES has a wider spatial domain on which it might have detected a cloud during the same period it was a clear sky aboard the ship.*

  *We deleted Table B1.*

- 601/Fig. 13: Why is CERES CS and T-CARS shown as well in Fig.13? The focus is on the CERES all-sky flux. The other variables should be removed from all subplots in Fig. 13.

  *We agree with the Referee's comment. The Figure has been changed accordingly.*

- 621: "Similarly to LWD, there is relatively good agreement between the SWD CERES simulations and observations …" "SWD" to be added.

  This is difficult to see from Fig 13b alone. Please refer to 14 b as well.

  *The text has been modified as Referee suggested*

- 675-698: The different CERES products have not been introduced in detail which is quite important for this section since it is not straight forward how these products have been derived and what are they representing in detail. However, I strongly recommend to remove this whole section from the manuscript. The manuscript is already very long and this part opens a completely new topic which

distracts from the actual topic (as also mentioned in l 699). Such a study is definitely of interest but should be presented in a separate paper.

*We agree with the Referee. This has been clarified in the introduction of Satellite data set in section 2.2. We included the following text* (Page 8).

*'A description of the retrievals is presented in Minnis et al. (2020). A summary of the cloud parameters used in this study is presented in Table 1. The simulated CERES SYN1deg products used in this study focus on an all-sky (AS), cloudy without aerosols (NA), clear-sky (CS), and virtually pristine (P) atmosphere.'*

*The lines mentioned (675-698) have been removed from the manuscript.*

- 716: Maybe it is hard to see since the net CRE (I assume it is the net CRE, please clarify in text and figure caption) is shown while surface albedo, for example, only affect the SW CRE.

  I am also not sure if Fig 17 provides new insights since many statements in the text could also be made without this figure. I am wondering if this detailed discussion is really needed and the whole section (l 708-734) could be shortened.

  *We decided to remove the plot and shorten the text.*

- 725 - 734: This paragraph is a repetition of the previous paragraph. Please check.

  *This repetition has been removed..*

- Fig. 15 How is "the Arctic" in this case defined? Please mention this also in the figure caption and remind the reader of the time period considered. I cannot distinguish "grey" from "black".

  *The caption has been clarified. The description indicates central boxes instead the grey colour.*

Typos/grammar

l 84: macro with "-"

*Changed.*

l 345: macro with "-"

*Changed.*

ll 432-433: "Scatter plots..." not a full sentence.

*Corrected.*

l 721: This is not a full sentence.

*This has been corrected.*
* * *
**Response to Referee #2**

**Clarifications for specific comments:**

- Ln 49-50: The data sources used by Riihelä et al. were CERES, GEWEX SRB (a separate dataset), and flux components calculated with the FluxNet-Streamer RT code driven primarily by CLARA cloud and surface parameters. Please clarify this point and note GEWEX data.

*This point has been clarified, and the text has been changed to (Page 2):*

*'The investigation by Riihelä et al. (2017) presents an intercomparison between ground- based observations and several satellite products of surface radiative fluxes. Downward and upward LW and SW radiative flux observations from the Tara drifting ice camp and long-term observations on the Greenland Ice Sheet are compared to the CERES SYN1deg ed.3A, FluxNet, and Satellite Application Facility on Climate Monitoring cLoud, Albedo and RAdiation (CLARA) data sets (Karlsson et al., 2017), and the Global Energy and Water Exchanges (GEWEX) SRB (Wu and Fu, 2011). This study concludes that CERES SYN1deg has the smallest root-mean-square error (RMSE) compared against in-situ fluxes. This study recommends to further investigate differences in the surface and cloud properties that lead to discrepancies in flux retrievals.'*

- Ln 147: Upon first read, I expected to find the specs for the horizontal size and resolution of the "pixel grid", only realizing later that the authors wanted to say that there is only one 'stack' of grid cells in the vertical direction. Please revise to clarify, noting at least the ballpark figure or estimate of the horizontal coverage/footprint of the shipborne measurements.

  *The text has been modified and this aspect clarified as follows (Page 6):*

  *'As a first step, the measurements are averaged onto a common pixel grid with a vertical and temporal resolution of 31.18 m and 30 s, respectively, leaving a total of 595 vertical pixel grids and, in general, more than 2700 time-steps (Griesche et al., 2020)'*

- Ln 159: If the QL retrievals are based on training against radiosondes, are you certain that the relationships based on a single source site in Ny-Ålesund are sufficiently robust to work anywhere else over the Arctic Ocean?

  *There is no long-term data set of radiosondes profiles within the central Arctic Ocean from which a retrieval can be derived. Given that PS106 covered mostly the Svalbard region, it has been assumed that Ny-Ålesund provides consistent and reliable data set to train the LWP retrieval.*

- Ln 165 and 170-172: The impacts of rain and liquid/ice mixtures on QL are noted, but isn't QI affected just as well, as cloud radar reflectivity is a driver for it too?

  *Yes, it is. The text has been modified as follows (Page 7):*

  *'Precipitation conditions compromise the retrieval accuracy of $Q_L$ and $Q_I$ from the MWR and cloud radar, respectively.'*

- Ln 183-185: Here it was difficult to follow what it means when "Cloudnet pixel type...(is) assigned value to zero". Does it mean that aerosols and insects are discarded from analysis entirely? Yet the later manuscript estimates CERES aerosol radiative effects, would there have been a chance to analyze similar aerosol effects from in situ data? This is a bit confusing.

  *Yes, the pixels classified as "aerosols" or "insects" were excluded from the analysis. As per suggestion of Referee #1, the analysis of CERES aerosol effect has been removed from the paper to avoid distracting the reader from the main focus of the manuscript.*

  *For clarification, the text was changed as follows (Page 7).*

  *'Thus, as a first step, any Cloudnet pixel of "aerosols", "insects", and "aerosols and insects" are removed by changing its assigned value to zero to discard them from the analysis.'*

Section 2.2: The CERES data product background is nicely described, but please also state the name of the data product used. Is it SYN1deg?

*Yes. We opted to use the general name for simplicity. Nevertheless, for precision the new version considers CERES SYN1 for precision and consistency with the literature.*

- Ln 227 – 228: The text reads like the PS106 radiosonde data was assimilated into ERA5. Was this indeed the case?

*Yes. As mentioned in the paper. ERA5 assimilates the radiosondes launched from Polarstern.*

- Ln 242 – 244: The impact of ice crystal habits on RT has been investigated and the effects are not negligible (e.g. Wendisch et al., 2005: https://doi.org/10.1029/2004JD005294). Please provide some consideration for the potential impacts of assuming spherical ice crystals in T-CARS?

*We agreed on the high importance of the selection of the parameterization of the ice-crystals. We revised the text and edited it as follows* *(Page 9)*:

*'The parameterization for ice clouds assumes spherical ice crystals with $R_{E,I}$ values with an allowed range between 5.0 and 131.0 µm. Radiative fluxes are known to be sensitive to assumptions about the crystal habit, eg., hexagonal shape (Wendisch et al., 2005). However, the decision was made based on the availability of parameterizations in RRTMG and to be consistent with the Cloudnet parameterization of ice crystals.'*

- Ln 289: Please be careful here – the ERA5 underestimation described by Pohl et al. had its roots also in the use of (simple) literature-based constants for the albedo of various ocean/ice surfaces – if SIC and ice albedo were perfectly simulated but melt ponds missing, the resulting albedo should be an overestimation since melt ponds darken the surface relative to snow or bare ice. The text now suggests that missed melt ponds will result in albedo underestimation, which is not generally so.

*After careful consideration and also the suggestion from Referee #1, we have decided to exclude the comparison of surface albedo between ERA5 and CERES since no additional analysis is provided after the comparison.*

- Ln 414: The SWU effect is very large, but quite consistent with e.g. radiative kernel calculations for radiative energy balance disturbance following a certain change in albedo (e.g. Bright and O'Halloran, 2019) - you may wish to note this for reinforced belief in the result given.

Bright, R. M., & O'Halloran, T. L. (2019). Developing a monthly radiative kernel for surface albedo change from satellite climatologies of Earth's shortwave radiation budget: CACK v1. 0. Geoscientific Model Development, 12(9), 3975-3990.

*Thank you for the reference. We consider the citation appropriate to emphasize the finding.*

- Ln 432 – Looks like a broken reference here to a scatterplot figure X?

*Yes. Unfortunately, we repeated that text by mistake.*

- Ln 581 - 582: A larger negative bias in CERES all-sky fluxes due to "the presence of clouds" seems like a half-formed sentence. Clouds are included in all-sky fluxes in every case, how do they now contribute to bias increases? Please be more specific.

*The clear-sky comparison is based on the Cloudnet classification. There were periods when clouds did not pass directly over the active remote sensing instruments, which is just a point measurement. Thus, no cloud observations were obtained; however, their presence have been captured on the larger spatial footprint by CERES. The text has been clarified to (Page 20):*

*'These values confirm that the larger negative bias for all-sky conditions is due to the presence of clouds that were captured within the CERES footprint but did not pass over the shipborne remote sensing instrumentation.'*

- Ln 611: Interesting to see a fog case noted, since those would be expected to be the ones where satellite-based fluxes could be very biased since fog conditions are challenging for them. Was this the only case of fog during the cruise?

  *In general, the fog events were characterized by the classification of low-level stratus clouds. This suggests that fog events were often present during the PS106 cruise (Griesche et al., 2020). However, the case described was characterized by a dense fog that lasted the longest during the entire cruise.*

- Ln 719 – 734: Here the attention seemingly slipped, resulting in broad repetition of content between the two paragraphs and generally hard to follow descriptions. Fig 17c and d are not really "subdivisions" of 17b since the y-axis unit is not the same, but they are the same sample set divided by albedo threshold. Please revise this section carefully for consistency and clarity.

  *We agree with the comment. However, the text has been deleted considering the comment from Referee #1 (Page 24):*

- Figures 2 to 4: Since you already have the visualization available on Polarstern being in open water, MIZ, or dense ice in Fig 16, why not include the same information here? It is especially relevant for Fig 2.

  *Figure 2 to 4 were edited including this recommendation.*

- Figures 8 and 10: Please note that light yellow is a color very easily lost during printing, perhaps a shade or two darker would be more apparent.

  *The colour followed the standard Cloudnet's colours. Nevertheless, the new color is darker.*

- Figure 11: The "pale yellow" shading appeared either red or orange (on screen and paper) – or is it the rectangular regions at ~10Z and ~23Z that you refer to here? Also, on this figure it seems that the Cloudnet-CERES differences in QL and QI are quite stable in time, but the CRE difference fluctuates considerably? I may have missed the explanation in the text, but why is this the case?

  *Yes, we referred to the rectangular region at ~10Z and ~23Z. The shading is changed to gray. We explained that the abrupt change of surface albedo was due to the simultaneous rapid reduction of surface albedo from a value of 0.6 to 0.27. The end of section 3.3 explains it (Page 18).*

  *'The radiative effect of clouds on this day has a strong cooling influence both at the SFC and TOA that is enhanced by the surface albedo. In Fig. 11c. An abrupt change of the CRE at the SFC and the TOA is visible at 05:00Z in Fig. 11c, due to a simultaneous rapid reduction of surface albedo from a value of 0.6 to 0.27 (see also Fig. 2a).'*

---

## Referee Report (RR1)

Review of "Radiative closure and cloud effects on the radiation budget based on satellite and ship-borne observations during the Arctic summer research cruise PS106"
by Carola Barrientos-Velasco, Hartwig Deneke, Anja Hünerbein, Hannes J. Griesche, Patric Seifert, and Andreas Macke

I thank the authors for the thorough replies and adjustments to the manuscript. Before publication, I think that it is important that some few further changes are applied to the manuscript. Some important information, for example, which has been given in the reply to the reviewer, has not been or only partly included in the manuscript itself. Please see my specific comments below.
To better follow the previous discussion, I have included my original questions/comments (in black), the answer of the authors (grey) and my follow-up comment (red).

**Comment 1**

old comment: please provide more details on how $q_L$ is determined (equation?) No information on the retrieval of $r_{E,L}$ is given. How do you handle mixed-phase clouds, since in these cases the "standard" retrievals for liquid clouds do not work?

*answer: It was not our aim to describe in detail the Cloudnet methodology since this was covered in Griesche et al., 2020. However, we hope that with the modified version of the manuscript the question has been answered. See below an extract of the modified text in Section 2.1.2 (Page 6).*

new comment: I didn't meant to have a detailed explication/equation on how the adiabatic liquid water content is calculated. Lines 170- 178 are not needed (but thanks for clarifying!). Just mentioning that the liquid water content profile is assumed to be an adiabatic profile is fine. Are you scaling the profile with the LWP of the MWR? This information could be added after the sentence in line 167.
What is not clear to me is how reff can be calculated by Frisch et al (2002) in case of mixed-phase clouds. Since Z is dominated by the ice you do not have a Z for the liquid cloud droplets only. Please comment and also add further inormation in the manuscript.

**Comment 2**

old comment: Is the vertical grid used in T-CARS determined by the Cloudnet vertical grid? Do you need to interpolate cloud properties in time? I assume that the Cloudnet temporal grid has a resolution of 30 s.

*answer: The radiative transfer model has a limit of 200 atmospheric levels. Therefore, it was not possible to proceed with the exact same vertical grid as Cloudnet since Cloudnet has 413 atmospheric levels equally separated every by ~31.2 metres, starting from 165 metres to 13 km. It was opted to use the double Cloudnet's vertical grid (i.e., 62.5 m) for the first 10 km of the atmosphere. The temporal resolution of Cloudnet is about 30 s, so the cloud properties were interpolated to every minute for the simulations. This is explained in more detail in section 2.3 of the manuscript.*

new comment: Please expicitely mention in the manuscript what you did with the cloud properties, e.g. averaged 2 bins vertically and linear interpolation in time. You just mention which grid you use in the RTM but not how you adjust the cloud properties accordingly.

**Comment 3**

old comment: Fig. 5 b: Why can the thermodynamic phase not be given in case that there is precipitation?

*answer: As per the standarized Cloudnet classification only liquid is classified as precipitation ('drizzle/rain and cloud droplets' and drizzle or rain). When there is snowfall, there is a flag within Cloudnet data related to an uncorrected attenuation. It has not been planned to subdivide precipitation within the standard Cloudnet target classification.*

new comment: I am sorry. This question was misleading. What I was hinting at: single-layer clouds can be devided into liquid, ice, mixed-phase. Why is there another category precipitation for single-layer clouds? To me it is not clear why single-layer clouds which are precipitating can not be classified as well.

In the figure caption you mention "mixed-phase clouds of type 1 or 2" but only the first one is shown.

**Comment 4**

old comment: I would argue that the uncertainty estimates that you use to determine the overall uncertainty are rather at the lower end and most likely much higher due to the reasons mentioned before. Also for the SW, you omit the uncertainty related to aerosols which probably also has a high impact.

*answer: The values obtained from the sensitivity analysis are an approximate based on the input parameters used for the radiative transfer simulations during clear-sky conditions. Thus, it is used as a reference for subsequent studies.*

*While we did not include aerosols in the simulations due to the lack of observations, we quantified the radiative effect on the SWD from CERES with Figure B1. Under clear-sky conditions, the aerosols can contribute with a mean of $-10.8$ Wm$^{-2}$ and under cloudy conditions up to $-4.5$ $10.8$ Wm$^{-2}$.*

new comment: I would have liked to seen a least a short critical discussion in the manuscript about how realistic these assumptions on the uncertainties in the input parameters are and which uncertainties might be there in addition which have not been considered. As I mentioned, there are other uncertainties (using hourly model data not capturing the full temporal variability of water vapor and temperature, the assumption of Ts=T10m,…)

I think that it is really important to remind the reader of those ones as well.

**Comment 5**

old comment: "Therefore, our results confirm..."
It is true that the bias of LW_down in the clear sky case is smaller than your uncertainty estimate. Still I am a little bit concerned by this bias since you clearly see a systematic error which hints at too low temperatures and/or humidity in the RT simulations. Did you try to provide a better estimate of your input profiles, i.e. including the T10m in the lowest atmospheric level or by including the IWV from MWR or GPS?

*answer: "Initially, we compared different atmospheric input data sets into the radiative transfer model [i.e., radiosondes, ERA5] and analysed which results were closer to the observations. However, we did not find large differences between both simulations since, in general, there is a good agreement between ERA5 and radiosondes (see Fig. 3c and 3d). For consistency, we decided to use ERA5 as a homogeneous input parameter. Additionally, we created several sensitivity analyses varying the humidity and temperature profiles to determine how more humid or warmer the atmosphere needed to be to match the LWD observations. In some cases, the mean flux difference was larger than the radiosonde uncertainties (see section 3.2 and Table A1). Therefore, we hypothesised that the biases of the downward LW might be due to the temperature fluctuation of shipborne instrumentation working near the pyrgeometer (see conclusion point 3)."*

and later to a similar comment:
*"The LW bias has been observed for CERES and T-CARS (see Fig. 12a and Fig. 12c). Our hypothesis also includes that the operation of the shipborne instrument near the pyrgeometer might have caused an increase in the instrumental uncertainties, or the ship acted as a rather warm island in contrast to the ice floe. We, therefore, plan to compare shipborne and ice floe pyrgeometer measurements carried out during MOSAiC to test this hypothesis. (see conclusion number 2)."*

new comment: I think that this information, i.e. the further analyses you did to better understand the bias (i.e. varying input, testing with radiosonde,…) and the conclusion that the pyrgeometer measurements are likely influenced e.g. by the shipborne instrumentaion are of high relevance. This should definitely be included in more detail in the manuscript and in particular when you present and discuss the results of Fig.12. Just mentioning it as a aside in the conclusion points 2 and 3 is not sufficient.

**Comment 6**

old comment: I am also wondering if the bias only is a good indicator that RC is reached. Looking at the SWD flux differences there are quite some larger differences (second peak around -30 Wm-2, where does this come from?). So shouldn't be also the RMSE or STD discussed with respect to radiative closure?

*answer: The second peak around -30Wm-2 come from instances where the position of the sun was near the edge of Polarstern (65.6° in solar zenith angle) at 17:00Z approximately (see Fig. 7b). The multiple reflections caused by the relatively low sun position might have caused an increase in the SWD, increasing the bias from the simulations and observations. […]*

new comment: The reason for this second peak should also be included in the manuscript, i.e. when discussing of Fig. 12.

---

## Author Response (AR2)

**Authors' response – ACP**

**Radiative closure and cloud effects on the radiation budget based on
satellite and ship-borne observations during the Arctic summer
research cruise PS106
by  Barrientos Velasco et al.**

We thank the reviewers for the time and effort that they invested into the review of our manuscript, and for their helpful comments and suggestions. The description of the modifications considered in the manuscript can be found below. To address each point, we copied in red the last comment from the Referee and described our changes in black. Additionally, we added a screen-shot of the modified manuscript for some of the comments. In the screen-shots are marked in red the text that has been deleted and in blue the text that has been added in the latest version of the manuscript.

**Comment 1**

new comment: I didn't meant to have a detailed explication/equation on how the adiabatic liquid water content is calculated. Lines 170- 178 are not needed (but thanks for clarifying!). Just mentioning that the liquid water content profile is assumed to be an adiabatic profile is fine. Are you scaling the profile with the LWP of the MWR? This information could be added after the sentence in line 167.

What is not clear to me is how reff can be calculated by Frisch et al (2002) in case of mixed-phase clouds. Since Z is dominated by the ice you do not have a Z for the liquid cloud droplets only. Please comment and also add further information in the manuscript.

Response: Lines 170-178 were deleted from the manuscript. It was also specified that the profile was scaled with the LWP that comes from the MWR in line 167.

Regarding the calculation of reff, we agree that the reflectivity which is used to calculate the liquid droplet effective radius might be influenced by ice crystals. During PS106, most liquid droplets in mixed-phase clouds were observed in clouds with cloud top temperatures >-10°C (see Figure below). Bühl et al. (2016) showed that for mixed-phase clouds with cloud top temperatures >-10°C the reflectivity at heights were likely only ice crystals were present was usually below -40dBZ (see Fig. 7 in Bühl et al. (2016)). During PS106, the vast majority of the cloud radar reflectivity at the heights where liquid layers were observed in mixed-phase clouds was around -20dBZ. Therefore, we are confident that the reflectivity, which was used to derive the cloud droplet effective radius, was actually dominated by the liquid cloud droplets rather than ice crystals. To avoid an influence of the ice crystals on the reflectivity, which is used to derive the cloud droplet effective radius, a cloud radar Doppler peak separation, as e.g. proposed by Kalesse et al. (2019) and Radenz et al. (2019), would be necessary. These techniques are, however, not yet operational.

[Figure]

*Figure 1. 2D histogram depicting the frequency of occurrence of the radar reflectivity factor [dBZ]at different cloud top temperature [°C] for mixed-phase cloud cases during PS106.*

Additional information has been added to the manuscript regarding the derivation of liquid effective radius as seen below.

Once a cloudy pixel is identified, the cloud water content and effective radius are determined regardless if the cloud is detected as a single or mixed-phase cloud. The liquid water path is retrieved based on the HATPRO MWR measurements using the retrieval method developed in Löhnert and Crewell (2003). This method relies on a long-term radiosonde training data set, which in this case is based on Ny-Ålesund, NO (78.9°N, 11.85°E, WMO Code 6260). Once $Q_L$ is known, the liquid water content ($q_L$) and $r_{E,L}$ are determined. The $r_{E,L}$ considers the radar reflectivity factor and measurements of the integrated cloud liquid water based on the methodology described in Frisch et al. (2002). The retrieval of $q_L$ is obtained by distributing the observed the $Q_L$ from the HATPRO MWR adiabatically among the identified liquid and mixed-phase cloud pixels classified by the Cloudnet algorithm. This method assumes a log-normal cloud-droplet size distribution, which is constant with height. The uncertainties of $q_L$ are calculated by error propagation assuming a typical uncertainty of 20-25 $\mathrm{gm}^{-2}$ in $Q_L$ (Löhnert and Crewell, 2003). The adiabatic increase of liquid water is based on Brenguier (1991) and calculated by the following equation in .

$$\frac{dq_L}{dz} = -\left(1 - \left(\frac{C_p \cdot T}{L \cdot e}\right)\right) \cdot \left(\frac{1}{\left(\frac{C_p \cdot T}{L \cdot e}\right) + \left(\frac{L \cdot qs \cdot \rho a}{p - es}\right)}\right) \cdot (\rho a \cdot g \cdot e \cdot es) \cdot (p - es)^{-2}.$$

In Eq. ??, $T$ is the atmospheric temperature in , $p$ is atmospheric pressure in , $qs$ is the specific humidity mixing ratio in , $es$ is the saturated vapour pressure, $\rho a$ is the density of air in (see Eq. ??), $e$ is the ratio of the molecular weight of water vapour of dry air equal to 0.62198, $g$ is retrieval of $r_{E,L}$ considers the radar reflectivity factor and the profile of $Q_L$ based on the acceleration due to gravity -9.81 , $C_p$ is the heat capacity of air at a constant pressure 1005.0 , $L$ is methodology described in Frisch et al. (2002). The reflectivity used to derive the $r_{E,L}$ was likely not affected by the presence of ice crystals since most of the mixed-phase cloud cases observed during PS106 had cloud top temperatures larger than -10 °C and a radar reflectivity around -20 dBZ (not shown). The study of Bühl et al. (2016) showed that at these temperatures and radar reflectivities, the latent heat of evaporation $2.5 \times 10^6$ , $R_d$ is the specific gas constant for dry air 287.04 signal of the mixed-phase cloud is dominated by the presence of liquid droplets rather than the ice crystals. In cases where the radar reflectivity is lower (e.g., -40 dBZ) the ice crystals dominate more the signal of the mixed-phase cloud and other methods are necessary to derive the $r_{E,L}$ (Kalesse et al., 2019; Radenz et al., 2019).

**Comment 2**

new comment: Please expicitely mention in the manuscript what you did with the cloud properties, e.g. averaged 2 bins vertically and linear interpolation in time. You just mention which grid you use in the RTM but not how you adjust the cloud properties accordingly.

Response: The text has been edited as follows in Section 2.3:
* * *
**2.3 Radiative transfer simulations**

The TROPOS Cloud and Aerosol Radiative effect Simulator (henceforward T-CARS) is a Python-based framework to carry out radiative transfer simulations with a particular focus on the investigation of the radiative effects of aerosols and clouds. Parts of this framework have already been applied and described in Barlakas et al. (2020) and Witthuhn et al. (2021). T-CARS enables the use of various sources for input data such as atmospheric profiles of trace gases, temperature, humidity, properties of clouds, aerosols, and surface parameters. The present study employs the widely used rapid radiative transfer model (RRTM) for GCM applications (RRTMG; Mlawer et al. (1997); Barker et al. (2003); Clough et al. (2005)).

In this study, the daily T-CARS output files have a standard grid that consists of 197 atmospheric levels ranging from the surface up to 20 km height and with km height and 1-minute temporal resolution. The first 10 km km of the atmosphere is divided into 160 levels with a geometric layer thickness of 62.5 m. The m. The level thickness of each pixel for the first 10 km of the atmosphere corresponds to two vertical levels of Cloudnet's pixel which are averaged to the standard grid. The following 5 km km of the atmosphere have a layer thickness of 250 mm, while the last 5 km km of the atmosphere a layer thickness of 193.8 mm.
* * *
**Comment 3**

new comment: I am sorry. This question was misleading. What I was hinting at: single-layer clouds can be devided into liquid, ice, mixed-phase. Why is there another category precipitation for single-layer clouds? To me it is not clear why single-layer clouds which are precipitating can not be classified as well.
In the figure caption you mention "mixed-phase clouds of type 1 or 2" but only the first one is shown.

Response: The reason to have the precipitation category is to treat those cases with extra caution. The cloud products and flux observations can have larger uncertainties under precipitation events. This has been clarified in the paper with the following text:
* * *
The cloud phase flag was included to analyse the thermodynamic phase of clouds. Even though it is available for the entire PS106 time series, the focus is here directed to the thermodynamic phase of single-layer clouds , since these cases are the most frequent, and an analysis is less complex than for multilayer conditions. Fig. 5b Figure 5b shows on only single-layer cloud periods. This figure indicates an occurrence frequency of 32.6 36.7 % for single-layer mixed-phase clouds of type two (ice and super-cooled droplets), 16.8 19.4 % for mixed-phase clouds of type one (well-separated ice and liquid phase), and 15.6 21.5 % for single layer ice clouds. The remaining period is composed of single-layer clouds with precipitation (13.6 17.9 %), clear-sky periods (12.1 %), and single layer liquid clouds (2.7 4.5 %). We emphasise the need to consider precipitation periods since, during these events, there are larger uncertainties in observations (i.e., cloud radar, lidar, MWR, and radiometers).

We apologize since the wrong figure was accidentally chosen when compiling the manuscript. The label of Figure 5 has been corrected.

[Figure]

**Figure 5.** Daily and overall relative frequency of occurrence (RFO) of various cloud characteristics. Panel (a) shows the RFO of clear-sky, single-level clouds, and multilayer clouds. Panel (b) shows the RFO of the thermodynamic phase of single-layer clouds, differentiating periods of ice clouds, liquid clouds, mixed-phase clouds of type 1 or 2, and precipitation (PPT). Panel (c) is the RFO of various quality flags indicating optimum conditions (OC), low-level stratus (LLS), PPT and simultaneous occurrence of LLS and PPT.

**Comment 4**

new comment: I would have liked to seen a least a short critical discussion in the manuscript about how realistic these assumptions on the uncertainties in the input parameters are and which uncertainties might be there in addition which have not been considered. As I mentioned, there are other uncertainties (using hourly model data not capturing the full temporal variability of water vapor and temperature, the assumption of Ts=T10m,…) I think that it is really important to remind the reader of those ones as well.

Response: We agree that neglecting aerosols, extrapolating hourly data to 1 minute resolution, assuming near-surface temperature as skin temperature, and using model data for water vapour and temperature atmospheric profiles can lead to additional uncertainties. We emphasised these points even further with the following text in the manuscript (section 3.2).

'It is worth clarifying that additional uncertainties come from neglecting the presence of aerosols, the assumption of near-surface temperature as skin temperature, the extrapolation of hourly data into 1-min resolution, the assumption that the spatial interpolation from 0.25° latitude by 0.25° longitude (i.e., ERA5 data set) or 1° latitude by 1° longitude (i.e., CERES SYN1deg products) can capture the atmospheric and surface conditions and variability experienced during PS106. While it was attempted to quantify some of these uncertainties, a careful and more specific analysis should be extended in a different experimental setup.'

> It is worth clarifying that additional uncertainties come from neglecting the presence of aerosols, the assumption of near-surface temperature as skin temperature, the extrapolation of hourly data to 1-min resolution, the assumption that the coarse spatial grid from 0.25° latitude by 0.25° longitude (i.e., ERA5 data set) or 1° latitude by 1° longitude (i.e., CERES SYN1deg products) can capture the atmospheric and surface conditions and variability experienced during PS106. While it was attempted to quantify some of these uncertainties (e.g., the omission of aerosols in Fig. B1 and section 3.4.1), a careful and more specific analysis will be made by carrying out several sensitivity analyses to quantify these uncertainties considering the observations from the Multidisciplinary drifting Observatory for the Study of Arctic Climate (MOSAiC) expedition (Shupe et al., 2022). Our primary focus will be on the spatiotemporal differences among shipborne, reanalysis and satellite observations, which we believe are the largest source of uncertainties.
>
> **3.3 Case studies**

**Comment 5**

new comment: I think that this information, i.e. the further analyses you did to better understand the bias (i.e. varying input, testing with radiosonde,...) and the conclusion that the pyrgeometer measurements are likely influenced e.g. by the shipborne instrumentaion are of high relevance. This should definitely be included in more detail in the manuscript and in particular when you present and discuss the results of Fig.12. Just mentioning it as a aside in the conclusion points 2 and 3 is not sufficient.

Response: We agree with the comment. We have edited the following text in Section 3.4.1.

'Several simulations were conducted using only the Vaisala RS92-SGP radiosondes launched every 6 hours from *Polarstern* (Schmithüsen, 2017a, b), and a (Schmithüsen, 2017a, b), and also sensitivity analyses were made by varying the atmospheric temperature and humidity to try to match the observations of the LWD flux (not shown). However, the negative bias found for both T-CARS and CERES SYN1deg fluxes might also be caused by a positive bias of the ship-borne pyrgeometer observations, e.g., due to the influence of the exhaust plume of *Polarstern* or nearby instrumentation causing a fluctuation of temperature nearby the pyrgeometer. As there was only one pyrgeometer measurement aboard *Polarstern*, it is impossible to further investigate this hypothesis. However, for future campaigns, it is recommended here to operate two pyrgeometers installed in different locations of the research vessel to exclude such influences.'

> The  simulations were conducted using only the Vaisala RS92-SGP radiosondes launched every 6 hours from *Polarstern* (Schmithüsen, 2017a, b), and also sensitivity analyses were made by varying the atmospheric temperature and humidity to try to match the observations of the LWD flux (not shown). However, the negative bias found for both T-CARS and CERES SYN1deg fluxes might also be caused by a positive bias of the ship-borne pyrgeometer observations, e.g., due to the influence of the exhaust plume of *Polarstern* or other factors affecting the pyrgeometer. As there was only one pyrgeometer measurement aboard *Polarstern*, it is impossible to further investigate this bias. However, for future campaigns, it is recommended here to operate two pyrgeometers installed in different locations of the research vessel to exclude such influences.

**Comment 6**

new comment: The reason for this second peak should also be included in the manuscript, i.e. when discussing of Fig. 12.

Response: We agree with this point. Bellow is presented the additional description to the second peak. Additionally, is shown the difference between the previous version of the paper and the current one.

'The second peak centred around -50.0 $Wm^{-2}$ in Fig. 12d is most likely due to momentary obstructions on the observations that were not captured by the initial screening that affected about 5 data points of CERES SYN1deg simulations. It is also possible that part of the bias might be due to the presence of aerosols. However, since the behaviour for pristine and clear-sky conditions is similar, this cause is not the leading the bias for the second peak. In general, the biases of both T-CARS and CERES SYN1deg are both within the uncertainty limit of ±20 $Wm^{-2}$ indicating that radiative closure is determined for both data sets.'
* * *
The comparison for the SWD flux uses a stricter screening of data, which also excludes all periods when the pyranometer's field of view was obstructed by the superstructure of *Polarstern*. For T-CARS, a positive bias of 44.2 $W\,m^{-2}$ and a correlation coefficient of 0.85 were found initially without this screening. With screening, a bias of 9.5 $W\,m^{-2}$ and a correlation coefficient of 0.95 were obtained. In the case of CERES SYN1deg, the biases for all-sky, clear-sky, and pristine conditions were found to have values of -27.1 $W\,m^{-2}$, 3.6 $W\,m^{-2}$, and 12.0 $W\,m^{-2}$, respectively. These values confirm that the larger negative bias for all-sky conditions is due to the presence of clouds that were captured within the CERES SYN1deg footprint but did not pass over the shipborne remote sensing instrumentation. The second peak centred around -50.0 $W\,m^{-2}$ in Fig. 12d is most likely due to momentary obstructions on the observations that were not captured by the initial screening that contains about 5 data points of CERES SYN1deg simulations. In general, the biases of both T-CARS and CERES SYN1deg are both within the uncertainty limit of $\pm20$ $W\,m^{-2}$ indicating that radiative closure is achieved determined for both data sets.